# MOBODY: MODEL-BASED OFF-DYNAMICS OFFLINE REINFORCEMENT LEARNING

**Yihong Guo**[1]   **Yu Yang**[2]   **Pan Xu**[2]   **Anqi Liu**[1]

[1]Johns Hopkins University   [2]Duke University

{yguo80,aliu.cs}@jhu.edu   {yu.yang,pan.xu}@duke.edu

## ABSTRACT

We study off-dynamics offline reinforcement learning, where the goal is to learn a policy from offline source and limited target datasets with mismatched dynamics. Existing methods either penalize the reward or discard source transitions occurring in parts of the transition space with high dynamics shift. As a result, they optimize the policy using data from low-shift regions, limiting exploration of high-reward states in the target domain that do not fall within these regions. Consequently, such methods often fail when the dynamics shift is significant or the optimal trajectories lie outside the low-shift regions. To overcome this limitation, we propose MOBODY, a Model-Based Off-Dynamics Offline RL algorithm that optimizes a policy using learned target dynamics transitions to explore the target domain, rather than only being trained with the low dynamics-shift transitions. For the dynamics learning, built on the observation that achieving the same next state requires taking different actions in different domains, MOBODY employs separate action encoders for each domain to encode different actions to the shared latent space while sharing a unified representation of states and a common transition function. We further introduce a target Q-weighted behavior cloning loss in policy optimization to avoid out-of-distribution actions, which push the policy toward actions with high target-domain Q-values, rather than high source domain Q-values or uniformly imitating all actions in the offline dataset. We evaluate MOBODY on a wide range of MuJoCo and Adroit benchmarks, demonstrating that it outperforms state-of-the-art off-dynamics RL baselines as well as policy learning methods based on different dynamics learning baselines, with especially pronounced improvements in challenging scenarios where existing methods struggle.

## 1 INTRODUCTION

Reinforcement learning (RL) (Kaelbling et al., 1996; Li, 2017) aims to learn a policy that maximizes cumulative reward by interacting with an environment and collecting the corresponding rewards. While RL has led to impressive successes in many domains, such as autonomous driving (Kiran et al., 2021) and healthcare (Lee et al., 2023), it faces significant constraints on interaction with the environment due to safety or cost concerns. One solution is to learn a policy from a pre-collected offline dataset (Levine et al., 2020). Still, when the offline dataset is insufficient, data from another environment, such as a simulator with potentially mismatched dynamics, may be needed, but requires further domain adaptation. In our paper, we study a specific type of domain adaptation in RL, called off-dynamics offline RL (Liu et al., 2022; 2024a; Lyu et al., 2024b), where the simulator (source) and real/deployed (target) environments differ in their transitions. The agent is not allowed to interact with the environment but only has access to offline data that is pre-collected from the two domains with mismatched dynamics and trains a policy with the offline data.

Existing works on off-dynamics offline RL solve the problem by 1) reward regularization methods (Liu et al., 2022; Xue et al., 2023; Wang et al., 2026) through the state visitation frequency or estimation of the dynamics gap or 2) data filtering methods (Xu et al., 2023; Liu et al., 2024a; Wen et al., 2024) that penalize or filter out source transitions with high dynamics shift. As a result, the policy is mostly optimized with the transitions from the low shift regions, limiting exploration of high-reward states in the target domain that do not fall within these regions. Consequently, such

methods often fail when the dynamics shift is significant or the optimal/high-reward trajectories lie outside the low-shift regions. So we wonder, can we directly optimize the policy with the target transition, instead of only the low-shift regions to allow for more exploration of the high target reward and large shift region?

Motivated by this, we propose a Model-Based Off-Dynamics RL algorithm (MOBODY) that learns target domain dynamics through representation learning and optimizes the policy with *exploratory* rollout from the learned dynamics instead of only the low-shift region data. Existing dynamics learning methods, such as learning with limited target data, learning with combined source and target data, and pretraining on source and finetuning on the target domain, are infeasible in the off-dynamics RL setting due to the intrinsic dynamics difference in this problem. This is because 1) the dynamics learned from the combined dataset is not the accurate target dynamics, but the dynamics resemble the source one as the source transitions dominate the dataset, 2) the pretrain-finetune method still doesn't capture what is the difference between source and target dynamics using the same dynamics model, but only tries to learn the target domain based on the source transition.

To learn the target dynamics, we leverage shared structural knowledge across domains, such as the high-level robot motion and position in a robotics task, while employing separate modules to account for domain-specific dynamics differences. Specifically, we observe that to achieve the same next state starting from the same state, different actions are required in two domains. Based on this, we propose to learn separate action encoders for the two dynamics to encode actions into a unified action representation, and also learn a unified transition and state encoder to map the unified latent state and action representation to the next state. And such shared representation and transition functions can be learned with the auxiliary of the source data through representation learning. In this way, MOBODY learns separate transition functions for two domains but utilizes the source data to provide shared structure knowledge regarding the transitions. As shown in Figure 1, MOPO that di-

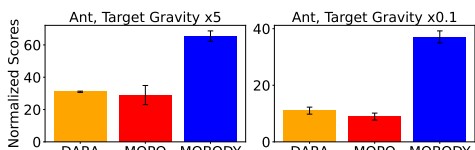

Figure 1: Comparison between DARA (Liu et al., 2022) (a SOTA model-free reward regularization method for offline off-dynamics RL), MOPO (Yu et al., 2020) (a vanilla model-based offline RL), and MOBODY on two MuJoCo tasks. We show that 1) the model-free method DARA receives low reward compared with model-based MOBODY due to a lack of exploration in the target domain, and 2) MOPO fails as it cannot learn a good transition for exploration with a combined source and target dataset.

rectly learns dynamics with combined source and target data significantly underperforms MOBODY, which is specifically optimized to learn the *target* dynamics.

We further propose a practical and useful target Q-weighted behavior cloning regularization in the policy learning to avoid out-of-distribution and high source Q value (but low target Q value) actions, inspired by the advantage-weighted regression (Peters & Schaal, 2007; Kostrikov et al., 2021a). The vanilla behavior cloning loss (Fujimoto & Gu, 2021) will push the policy to favor the action in the source data, but the action in the source data might not perform well in the target domain due to the dynamics shift. To overcome this issue, the target Q-weight behavior cloning loss regularization will up-weight action with the high *target* Q value. And we empirically validate the choice of the target-Q weight BC loss.

Our contribution can be summarized as follows:

- We propose a novel paradigm for off-dynamics offline RL, called model-based off-dynamics offline RL, that can explore the target domain with the learned target transitions instead of optimizing the policy only with the low-shift transitions.
- We propose a novel framework for learning the *target* dynamics with source data and limited target data by learning separate action encoders for the two domains while also learning a shared state and the transition in the latent space. We also incorporate a target Q-weighted behavior cloning loss for policy optimization that is simple, efficient, and empirically validated for off-dynamics offline RL settings.
- We evaluate our method on MuJoCo and Adroit environments in the offline setting with different types and levels of off-dynamics shifts and demonstrate the superiority of our model with an average $44\%$ improvement over baseline methods on the gravity and friction settings and $25\%$ on the kinematic and morphology shift settings.

## 2 BACKGROUND

**Off-dynamics offline reinforcement learning**. We consider two Markov Decision Processes (MDPs): the source domain $\mathcal{M}_{\text{src}} = (\mathcal{S}, \mathcal{A}, R, p_{\text{src}}, \gamma)$ and the target domain $\mathcal{M}_{\text{trg}} = (\mathcal{S}, \mathcal{A}, R, p_{\text{trg}}, \gamma)$. The difference between the two domains lies in the transition dynamics $p$, i.e., $p_{\text{src}} \neq p_{\text{trg}}$ or more specifically, $p_{\text{src}}(s' \mid s, a) \neq p_{\text{trg}}(s' \mid s, a)$. Following existing literature on off-dynamics RL (Eysenbach et al., 2020; Liu et al., 2022; Lyu et al., 2024a; Guo et al., 2024; Lyu et al., 2024b; Wen et al., 2024), we assume that reward functions are the same across the domains, which is modeled by the state, action, and next state, i.e., $r_{\text{src}}(s, a, s') = r_{\text{trg}}(s, a, s')$. The dependency of the reward on $s'$ is well-justified in many simulation environments and applications, such as the Ant environment in MuJoCo, where the reward is based on how far the Ant moves forward, measured by the change in its x-coordinate (i.e., the difference between the x-coordinate after and before taking action). The goal is to learn a policy $\pi$ with source domain data $(s, a, s', r)_{\text{src}}$ and limited target domain data $(s, a, s', r)_{\text{trg}}$ that maximize the cumulative reward in the target domain $\max_\pi \mathbb{E}_{\pi, p_{\text{trg}}} [\sum_t \gamma^t r_{\text{trg}}(s_t, a_t)]$. In the offline setting, we are provided with static datasets from a source and a target domain $\mathcal{D}_{\text{src}} = \{(s, a, s', r)_{\text{src}}\}$ and $\mathcal{D}_{\text{trg}} = \{(s, a, s', r)_{\text{trg}}\}$, which consist of the transitions/trajectories collected by some unknown behavior policy. Note that in the off-dynamics setting, the number of transitions from the target domain is significantly smaller than the source, i.e., $|D_{\text{trg}}| \ll |D_{\text{src}}|$, and normally the ratio $\frac{|D_{\text{src}}|}{|D_{\text{trg}}|}$ can vary from 10 to 200. In our paper, we follow the ODRL benchmark (Lyu et al., 2024b) in which the ratio is 200.

**Model-based offline reinforcement learning**. Model-based RL learns a transition function $\hat{T}(s', r|s, a)$ by maximizing the the likelihood $\hat{T} = \max_T \mathbb{E}_{D_{\text{offline}}}[\log \hat{T}(s', r|s, a)]$. Then, the algorithm rolls out new transition data to optimize the policy and take $u(s, a)$ as the uncertainty quantification to obtain a conservative transition, i.e., $(s, a, s', \hat{r} - \alpha u(s, a))$. The policy with offline data $\mathcal{D}_{\text{offline}}$ and online rollout $(s, a, s', \hat{r} - \alpha u(s, a))$. However, different from traditional model-based offline RL, we only have very limited target domain data and source data with dynamics shift. There is no existing model-based solution for off-dynamics RL, which calls for novel methodology development both in dynamics learning and policy learning.

Detailed discussions of related work are in Appendix A due to space limit.

## 3 MOBODY: MODEL-BASED OFF-DYNAMICS OFFLINE REINFORCEMENT LEARNING

In this section, we present our algorithm, MOBODY, for the off-dynamics offline RL problem setting. We first present how we learn the *target* dynamics with very limited target domain data $\mathcal{D}_{\text{trg}}$ and source domain data $\mathcal{D}_{\text{src}}$. Secondly, for policy learning, we incorporate a target Q-weighted behavior cloning loss to regularize the policy, where the target Q value is learned from *enhanced target data*, including reward regularized source data, target data, and rollout data from learned dynamics. The algorithm is summarized in Algorithm 2.

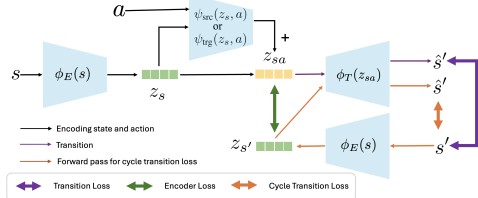

Figure 2: Architecture of the dynamics model. MOBODY encodes the state with $\phi_E$ and state action with $\psi$, outputs the next state through $\phi_T$, and learns the dynamics for both domains by transition loss shown in purple double arrow $\Leftrightarrow$. It learns the state action representation by matching the state action representation $z_{sa}$ with the next state representation $z_{s'}$ through encoder loss shown in the green double arrow $\Leftrightarrow$ and the state representation through cycle transition loss shown in orange double arrow $\Leftrightarrow$.

### 3.1 LEARNING THE TARGET DYNAMICS

**Decomposition of the dynamics**. In general, the dynamics can be modeled as $s' = \phi(s, a)$ or $s' = \phi^{\text{src}}(s, a)$ and $s' = \phi^{\text{trg}}(s, a)$ for the two domains. Although the dynamics are different, the transitions share some structured knowledge that we can utilize. Also, from another perspective, for the two domains, to achieve the same next state, different action is required, i.e., $(s, a_{\text{src}}, s')_{\text{src}}$ and $(s, a_{\text{trg}}, s')_{\text{trg}}$. Based on this, we propose using separate action encoders to encode actions from the two domains into the

shared latent space. So the source and target domains can share a unified representation of states and a common transition function with the latent action.

We define the $z_s$ as the state representation, $z_{sa}^{\mathrm{src}}$ and $z_{sa}^{\mathrm{trg}}$ as the state-action representations from the action encoder for source and target dynamics, respectively. Specifically, we model the state and state action representation through $z_s = \phi_E(s)$, $z_{sa}^{\mathrm{trg}} = \psi_{\mathrm{trg}}(z_s, a)$ and $z_{sa}^{\mathrm{src}} = \psi_{\mathrm{src}}(z_s, a)$, so that we can obtain a separate state action representation for two domains. So, with the learned representation $z_s$ and $z_{sa}$, we have the dynamics modeled as $s' = \phi_T(z_s, \psi(z_s, a))$, where $\phi_T$ is the transition function. For simplicity and to reduce the model parameters, we choose to directly add the state and state action representation together and feed into the transition function $s' = \phi_T(z_s + \psi(z_s, a))$. Note that this additive term $z_s + \psi(z_s, a)$ is also widely adopted in the implementation of the model-based RL, where the transition is modeled as $s' = s + f(s, a)$. We show the flow of the dynamics learning component in Figure 2. And our dynamics model is:

$$\text{source dynamics} : z_s = \phi_E(s), z_{sa} = z_s + \psi_{\mathrm{src}}(z_s, a), \hat{s}' = \phi_T(z_{sa}), \tag{1}$$

$$\text{target dynamics} : z_s = \phi_E(s), z_{sa} = z_s + \psi_{\mathrm{trg}}(z_s, a), \hat{s}' = \phi_T(z_{sa}), \tag{2}$$

where $\phi_E$ is the state encoder, $\phi_T$ is the transition, and $\psi_{\mathrm{src}}$ and $\psi_{\mathrm{trg}}$ are the state action encoders for source and target, respectively. Equation (1) and Equation (2) show that we use different modules (action encoder $\psi_{\mathrm{src}}$ and $\psi_{\mathrm{trg}}$) for the source and target domains, but with shared state representation from state encoder $\phi_E$ and unified transition function $\phi_T$. We now discuss how representation learning techniques, utilizing several loss functions, enable us to learn representation and dynamics.

**Transition Loss**. The transition loss minimizes the Mean Squared Error of the predicted next state and the ground truth next state as shown in a purple two-way arrow in Figure 2. The goal of the transition loss is to learn the shared transition knowledge $\phi_T$ using both source and target data.

$$L_{\mathrm{dyn}}^{\mathrm{src}} = \tfrac{1}{N} \sum_{i=1}^{N} \|s' - \phi_T(z_s + \psi_{\mathrm{src}}(z_s, a))\|^2; L_{\mathrm{dyn}}^{\mathrm{trg}} = \tfrac{1}{N} \sum_{i=1}^{N} \|s' - \phi_T(z_s + \psi_{\mathrm{trg}}(z_s, a))\|^2. \tag{3}$$

**Encoder Loss** (Learning separate action encoder $\psi_{\mathrm{src}}$ and $\psi_{\mathrm{trg}}$). The $\phi_T$ can map the latent state action representation to the next state for both domains, we use encoder loss to learn the separate action encoders for the two domains to map different actions to the unified latent space that served as the input to the $\phi_T$. Specifically, we adopt a general assumption in representation learning that the representation of the state action should be close to the next state (Ye et al., 2021; Hansen et al., 2022b), where the predicted representation of the current state-action pair $\psi(s, a)$ incorporates the transition information to be close to the next state representation $\phi_E(s')$. This encourages the action encoder to further encode the difference of the dynamics information for the two domains, thereby improving the efficiency of learning the dynamics model. The encoder loss is formulated as:

$$L_{\mathrm{rep}}^{\mathrm{src}} = \tfrac{1}{N} \sum_{i=1}^{N} \||z_{s'}|_\times - (z_s + \psi_{\mathrm{src}}(z_s, a))\|^2, L_{\mathrm{rep}}^{\mathrm{trg}} = \tfrac{1}{N} \sum_{i=1}^{N} \||z_{s'}|_\times - (z_s + \psi_{\mathrm{trg}}(z_s, a))\|^2, \tag{4}$$

$z_{s'} = \phi_E(s')$ is the next state representation encoded with $\phi_E$ and $|\cdot|_\times$ is the stopping gradient. Here, $N$ is the batch size. The encoder loss is shown in a green two-way arrow in Figure 2.

**Cycle Transition Loss** (Learning shared $\phi_E$ and $\phi_T$). To further improve the state representation quality and avoid mode collapse in the encoder loss, we include a "cycle transition loss" through VAE-style (Kingma et al., 2013) learning. The dynamics function maps the state action to the next state through the state action representation. Then, from one perspective, by setting $\psi$ to 0, the dynamics only input the state into the dynamics learning framework, and no action will be taken. The output of the dynamics will be the same state, i.e., $(s, 0, s)$, which is the same for two domains. So when the $\psi$ is set to 0, the state is predicted as: $\hat{s} = \phi_T(\phi_E(s) + 0)$. Then we can explicitly learn the state representation with the state in the offline dataset by minimizing: $\|\phi_T(\phi_E(s)) - s\|_2$. From this perspective, we can view $\phi_E$ as an encoder and $\phi_T$ as a decoder, and we propose using a Variational AutoEncoder (VAE) (Kingma et al., 2013) to learn the state representation.

Let $z_s$ be expressed as $z_s = \mu_{\phi_E}(s) + \sigma_{\phi_E}(s) \odot \epsilon$, with $\epsilon \sim \mathcal{N}(0, I)$ and $\mu_{\phi_E(s)}$ and $\sigma_{\phi_E}$ are the output of state encoder network $\phi_E$. Let $d_z$ be the dimension of the latent representation, the loss for learning the state representation is:

$$\mathcal{L}_{\mathrm{cycle}} = \tfrac{1}{2N} \sum_{i=1}^{N} \sum_{j=1}^{d_z} \left( \mu_{i,j}^2 + \sigma_{i,j}^2 - \log \sigma_{i,j}^2 - 1 \right) + \tfrac{1}{N} \sum_{i=1}^{N} \|s_i - \hat{s}_i\|_2. \tag{5}$$

The cycle transition loss is shown in an orange two-way arrow in Figure 2.

Unlike previous VAE-based dynamics learning methods, which are not tailored for off-dynamics RL, we introduce a cycle transition loss alongside the encoder loss to jointly learn state representations

and shared transition functions across domains, rather than just learning state representations. The VAE representation also mitigates mode collapse that arises when trained solely on the encoder loss. The decoder, serving as a shared transition function, maps the unified state-action representation from separate action encoders to the next state, providing additional supervision signals for learning cross-domain dynamics. In conclusion, our method learns the unified transition function $\phi_T$ for both domains, while using the $\psi_{\text{src}}$ and $\psi_{\text{trg}}$ to learn the distinct information of the two dynamics.

**Reward learning and uncertainty quantification (UQ) of the learned dynamics.** Given that the reward is modeled as a function of $(s, a, s')$ tuple and the reward function is the same across domains, we learn the reward function $\hat{r}(s, a, s')$ as a function of the $(s, a, s')$ tuple with the combined source and target dataset through the MSE loss $L_{\text{reward}} = \frac{1}{2}\mathbb{E}_{D_{\text{src}} \cup D_{\text{trg}}}\big[r(s, a, s') - \hat{r}(s, a, s')\big]^2 + \frac{1}{2}\mathbb{E}_{D_{\text{src}} \cup D_{\text{trg}}}\big[r(s, a, s') - \hat{r}(s, a, \hat{s}')\big]^2$, where $\hat{s}'$ is the predicted next state. Here, we use both the true next state and the predicted next state from the dynamics model to learn the reward model, as during inference, we do not have the true next state and only have a predicted next state. This is further described in the Appendix B. Also, we follow the standard model-based approach (Yu et al., 2020) for the UQ of the dynamics, by penalizing the estimated reward $\hat{r}$ with the uncertainty in predicting the next state: $\tilde{r}(s, a, s') = \hat{r}(s, a, s') - \beta u(s, a)$ where $u(s, a)$ the uncertainty of the next state and $\beta$ is the scale parameter. We refer to the details in Appendix B.

To summarize, the dynamics learning loss is:

$$\min \mathbb{E}_{D_{\text{src}}} L_{\text{dyn}}^{\text{src}} + \mathbb{E}_{D_{\text{trg}}} L_{\text{dyn}}^{\text{trg}} + \mathbb{E}_{D_{\text{src}} \cup D_{\text{trg}}}[L_{\text{reward}} + \lambda_{\text{rep}}(L_{\text{cycle}} + L_{\text{rep}})], \tag{6}$$

where $\lambda_{\text{rep}}$ is a scalar controlling the weight of the representation learning term and set to be 1 in the experiments, as we notice there is no significant performance difference with different $\lambda_{\text{rep}}$. And the representation loss is summing the source and target loss: $L_{\text{cycle}} = L_{\text{cycle}}^{\text{src}} + L_{\text{cycle}}^{\text{src}}$, and $L_{\text{rep}} = L_{\text{rep}}^{\text{src}} + L_{\text{rep}}^{\text{src}}$. We summarize the dynamics learning algorithm in Algorithm 1.

## 3.2 Policy Learning with the Target-Q-Weighted Behavior Cloning Loss

After we learn the target dynamics, we perform model-based offline RL training. During the policy optimization, we roll out new target data from the learned *target* dynamics with the current policy and state in the offline data and keep the rollout data in the $\mathcal{D}_{\text{fake}}$. Also, we want to utilize the source data to optimize the policy. We follow the previous work by DARA (Liu et al., 2022) on off-dynamics offline RL. This approach first performs reward regularization on the source data, which learns domain classifiers $p(\text{trg}|s, a, s')$ and $p(\text{trg}|s, a)$ to penalize the reward of large shift in the source data: $r_{\text{DARA}}(s, a) = r(s, a) - \eta \log \frac{p_{\text{src}}(s'|s,a)}{p_{\text{trg}}(s'|s,a)}$. Details of the DARA are referred to in Appendix C.1. Our *enhanced target data* is $\mathcal{D}_{\text{enhanced}} = \mathcal{D}_{\text{src\_aug}} \cup \mathcal{D}_{\text{trg}} \cup \mathcal{D}_{\text{fake}}$, a combination of regularized source data, target data, and model rollouts.

**Learning the Q function** We learn the Q functions following standard temporal difference learning with *enhanced target data*:

$$\min \mathcal{L}_{\text{Q}} = \min \mathbb{E}_{\mathcal{D}_{\text{src\_aug}} \cup \mathcal{D}_{\text{trg}} \cup \mathcal{D}_{\text{fake}}} \left[ \left( r + \gamma \max_{a'} Q_{\theta-}(s', a') - Q_{\theta}(s, a) \right)^2 \right]. \tag{7}$$

**Policy optimization with target Q-weighted behavior cloning.** In offline RL, a central challenge is exploration error, as out-of-distribution actions cannot be reliably evaluated—an issue exacerbated under off-dynamics settings. Behavior cloning (Fujimoto & Gu, 2021; Goecks et al., 2019) offers a simple and effective regularization by biasing the policy toward actions in the offline dataset, by pushing actions close to the actions in the offline dataset. However, in off-dynamics RL, naively cloning source-domain actions can harm performance: actions in the source dataset may perform poorly in the target domain due to the dynamics shift, so vanilla behavior cloning alone in TD3-BC (Fujimoto & Gu, 2021) is insufficient for policy regularization.

Instead, inspired by the advantage weighted regression and the IQL (Kostrikov et al., 2021a), i.e. $L_{\pi}(\phi) = \mathbb{E}_{(s,a) \sim \mathcal{D}} \left[ \exp\left( \beta \left( \hat{Q}_{\theta}(s, a) - V_{\psi}(s) \right) \right) \log \pi_{\phi}(a \mid s) \right]$, which re-weight the log likelihood of the offline data with the advantage, we can re-weight the behavior cloning loss with the *target* Q value, namely a Q weighted behavior cloning loss, where the target Q value is learned with *enhanced target data*, so that this Q value approximates the Q value in the target domain. Intuitively, the target

Q-weighted behavior cloning loss up-weights the policy's loss with higher target Q-values, guiding the policy toward actions expected to perform better under target dynamics. The policy loss with Q weighted behavior cloning loss is:

$$
\pi = \arg\min_{\pi} - \mathbb{E}_{(s,a) \in D_{\text{enhanced}}} \left[ \lambda Q(s, \pi(s)) \right]
$$
$$
+ \mathbb{E}_{(s,a) \in D_{\text{src\_aug}} \cup D_{\text{trg}}} \left[ \exp\left( \frac{Q(s, \pi(s))}{1/N \sum_i^N |Q(s_i, \pi(s_i))|} \right) (\pi(s) - a)^2 \right], \tag{8}
$$

where the $\lambda = \frac{\alpha}{1/N \sum_i^N |Q(s,a)|}$ is the scaler $\lambda$ that balance the behavior regularization error and $Q$ loss and $\alpha$ is a hyper parameters. We empirically validate the choice of target Q-weighted instead of AWR style loss in Section 4.3 and Appendix C.4.2. We summarize the MOBODY in Algorithm 2 in the Appendix B.

## 4 EXPERIMENTS

In this section, we empirically evaluate MOBODY in off-dynamics offline RL settings using four MuJoCo environments from the ODRL benchmark: HalfCheetah-v2, Ant-v2, Walker2d-v2, and Hopper-v2 and manipulation tasks in Adroit: Pen and Door. We also perform comprehensive ablation studies to justify the importance of each component of MOBODY.

### 4.1 EXPERIMENTAL SETUP

**Environments, Tasks, and Datasets.** We evaluate MOBODY on the MuJoCo and Adroit environments from the ODRL benchmark (Lyu et al., 2024b). For the MuJoCo environment, we set the source domain unchanged and consider several types of dynamics shifts for the target domain, 1) gravity and friction, each scaled at four levels: $\{0.1, 0.5, 2.0, 5.0\}$ by multiplying the original values in MuJoCo, and 2) kinematics and morphology shift, each is achieved by constraining the rotation angle ranges of certain joints or modifying the size of specific limbs or the torsos of the robot. We also consider the Adroit task with kinematics and morphology shift, scaled to medium and hard shift levels, to demonstrate that our method applies to a wide range of environments and shift types/levels. We use the medium-level offline datasets collected by the ODRL benchmark, which uses an SAC-trained behavior policy tuned to achieve about 50% of expert performance. The target dataset is then collected through rollout trajectories until 5,000 target transitions are reached. Also, the source data contains 1 million transitions.

We evaluate the performance with the **Normalized Score**, defined as: $normalized\_score = \frac{score - random\_score}{expert\_score - random\_score} \times 100$, where the $random\_score$ is achieved by the random policy and the $expert\_score$ is achieved by the SAC (Haarnoja et al., 2018) trained to the expert level in the target domain. We also conduct hyperparameter and computational cost analysis in the Appendix C.4 to demonstrate that our method is not overly sensitive to hyperparameters.

**Baselines.** We compare MOBODY against model-free, model-based, and off-dynamics offline RL baselines. For model-free methods, we use IQL (Kostrikov et al., 2021a) and TD3-BC (Fujimoto & Gu, 2021), trained directly on the combined offline dataset of source and target transitions, without any modification tailored to off-dynamics settings. For model-based offline RL, we adopt MOPO (Yu et al., 2020): instead of training dynamics only on the target domain (which performs poorly in our setting), we follow prior off-dynamics work (Eysenbach et al., 2020) and train MOPO's dynamics model and policy on the combined source+target dataset. We further include representative off-dynamics offline RL methods DARA (Liu et al., 2022), BOSA (Liu et al., 2024a), and SRPO (Xue et al., 2023) and REAG (Wang et al., 2026). Finally, we compare MOBODY with alternative dynamics learning strategies in Section 4.3.2.

### 4.2 MAIN RESULTS

**Results on MuJoco gravity/friction shift**. In Table 1, we show the detailed results and highlight the best and second-best scores of the MuJoco gravity and friction shift problems. In the last row of Table 1, we sum the normalized scores in total. Our proposed MOBODY receives **44%** improvement over the best performing baselines, REAG, and performs the best or second best in 28 out of 32

Table 1: Performance of MOBODY and baselines on four MuJoCo tasks under medium-level offline dataset with dynamics shifts in gravity and friction (levels 0.1, 0.5, 2.0, 5.0). Source domains remain unchanged; target domains are shifted. We report normalized target-domain scores (mean $\pm$ std over three seeds). Best and second-best scores are highlighted in cyan and light cyan, respectively. **MOBODY receives** $44\%$ **improvement over the second best baseline REAG.**

| Env | Level | BOSA | IQL | TD3-BC | MOPO | DARA | REAG | SRPO | MOBODY |
|---|---|---|---|---|---|---|---|---|---|
| HalfCheetah Gravity | 0.1 | $9.31 \pm 1.94$ | $9.62 \pm 4.27$ | $6.90 \pm 0.34$ | $6.28 \pm 0.22$ | $12.90 \pm 1.01$ | $16.14 \pm 0.66$ | $32.94 \pm 1.65$ | $14.18 \pm 1.06$ |
| | 0.5 | $43.96 \pm 5.68$ | $44.23 \pm 2.93$ | $6.38 \pm 3.91$ | $40.20 \pm 7.20$ | $46.11 \pm 1.93$ | $40.50 \pm 1.58$ | $41.99 \pm 1.63$ | $\mathbf{47.18 \pm 1.23}$ |
| | 2.0 | $27.86 \pm 0.94$ | $31.34 \pm 1.68$ | $29.29 \pm 3.62$ | $21.89 \pm 10.49$ | $31.85 \pm 1.31$ | $33.28 \pm 3.16$ | $32.24 \pm 1.97$ | $\mathbf{41.60 \pm 7.35}$ |
| | 5.0 | $17.95 \pm 11.97$ | $44.00 \pm 23.13$ | $73.75 \pm 14.11$ | $57.75 \pm 18.92$ | $27.67 \pm 17.01$ | $71.31 \pm 2.80$ | $-2.33 \pm 0.69$ | $\mathbf{83.05 \pm 1.21}$ |
| HalfCheetah Friction | 0.1 | $12.53 \pm 3.61$ | $26.39 \pm 11.35$ | $8.95 \pm 0.71$ | $28.32 \pm 9.23$ | $23.69 \pm 16.46$ | $9.74 \pm 0.46$ | $17.36 \pm 0.73$ | $\mathbf{57.53 \pm 2.49}$ |
| | 0.5 | $68.93 \pm 0.35$ | $69.80 \pm 0.64$ | $49.43 \pm 9.91$ | $54.98 \pm 5.91$ | $64.89 \pm 3.04$ | $66.50 \pm 0.99$ | $109.18 \pm 2.15$ | $69.54 \pm 0.48$ |
| | 2.0 | $46.53 \pm 0.37$ | $46.04 \pm 2.04$ | $43.51 \pm 0.74$ | $42.33 \pm 3.89$ | $46.25 \pm 2.36$ | $37.74 \pm 2.35$ | $75.19 \pm 1.54$ | $50.02 \pm 3.26$ |
| | 5.0 | $44.07 \pm 9.07$ | $44.96 \pm 6.78$ | $35.83 \pm 6.65$ | $42.39 \pm 10.22$ | $40.06 \pm 7.87$ | $25.74 \pm 3.24$ | $5.10 \pm 1.96$ | $\mathbf{59.20 \pm 4.91}$ |
| Ant Gravity | 0.1 | $25.58 \pm 2.21$ | $12.53 \pm 1.11$ | $13.23 \pm 2.61$ | $8.93 \pm 1.23$ | $11.03 \pm 1.24$ | $15.75 \pm 1.17$ | $13.78 \pm 1.81$ | $\mathbf{37.09 \pm 2.12}$ |
| | 0.5 | $19.03 \pm 4.41$ | $10.09 \pm 2.00$ | $12.91 \pm 2.85$ | $12.28 \pm 3.88$ | $9.04 \pm 1.35$ | $13.25 \pm 0.86$ | $7.02 \pm 2.73$ | $\mathbf{37.44 \pm 2.79}$ |
| | 2.0 | $41.77 \pm 1.52$ | $37.17 \pm 0.96$ | $34.04 \pm 4.12$ | $35.43 \pm 3.22$ | $36.64 \pm 0.82$ | $43.25 \pm 1.72$ | $4.17 \pm 2.03$ | $\mathbf{45.83 \pm 1.71}$ |
| | 5.0 | $31.94 \pm 0.69$ | $31.59 \pm 0.35$ | $6.37 \pm 0.45$ | $28.97 \pm 5.93$ | $31.01 \pm 0.39$ | $49.36 \pm 2.61$ | $8.45 \pm 1.24$ | $\mathbf{65.45 \pm 3.23}$ |
| Ant Friction | 0.1 | $58.95 \pm 0.71$ | $55.56 \pm 0.46$ | $49.20 \pm 2.55$ | $49.86 \pm 5.99$ | $55.12 \pm 0.24$ | $54.13 \pm 0.56$ | $2.55 \pm 3.45$ | $58.79 \pm 0.11$ |
| | 0.5 | $59.72 \pm 3.57$ | $59.28 \pm 0.80$ | $25.21 \pm 7.17$ | $32.28 \pm 3.25$ | $58.92 \pm 0.80$ | $57.46 \pm 0.65$ | $6.57 \pm 1.76$ | $\mathbf{62.41 \pm 4.10}$ |
| | 2.0 | $20.18 \pm 3.79$ | $19.84 \pm 3.20$ | $22.69 \pm 8.10$ | $15.93 \pm 0.87$ | $17.54 \pm 2.47$ | $21.28 \pm 0.72$ | $10.81 \pm 2.09$ | $\mathbf{47.41 \pm 4.40}$ |
| | 5.0 | $9.07 \pm 0.88$ | $7.75 \pm 0.25$ | $10.06 \pm 4.16$ | $13.89 \pm 3.20$ | $7.80 \pm 0.12$ | $9.53 \pm 0.65$ | $11.72 \pm 1.86$ | $\mathbf{31.17 \pm 5.57}$ |
| Walker2d Gravity | 0.1 | $18.75 \pm 12.02$ | $16.04 \pm 7.60$ | $36.48 \pm 0.95$ | $41.98 \pm 10.13$ | $20.12 \pm 5.74$ | $26.56 \pm 2.62$ | $13.67 \pm 3.19$ | $\mathbf{65.85 \pm 5.08}$ |
| | 0.5 | $40.09 \pm 20.37$ | $42.05 \pm 10.52$ | $27.43 \pm 3.92$ | $40.32 \pm 8.78$ | $29.72 \pm 16.02$ | $55.20 \pm 2.18$ | $56.28 \pm 2.34$ | $43.57 \pm 2.32$ |
| | 2.0 | $8.91 \pm 2.28$ | $25.69 \pm 10.70$ | $11.88 \pm 9.38$ | $28.79 \pm 3.07$ | $32.20 \pm 1.05$ | $13.50 \pm 2.38$ | $8.52 \pm 0.82$ | $\mathbf{44.32 \pm 4.58}$ |
| | 5.0 | $5.25 \pm 0.50$ | $5.42 \pm 0.29$ | $5.12 \pm 0.18$ | $5.65 \pm 0.99$ | $5.44 \pm 0.08$ | $4.61 \pm 1.13$ | $5.12 \pm 0.46$ | $\mathbf{46.05 \pm 20.73}$ |
| Walker2d Friction | 0.1 | $7.88 \pm 1.88$ | $5.72 \pm 0.23$ | $29.60 \pm 24.90$ | $27.99 \pm 2.11$ | $5.65 \pm 0.06$ | $10.58 \pm 0.71$ | $9.02 \pm 0.81$ | $28.23 \pm 9.13$ |
| | 0.5 | $63.94 \pm 20.40$ | $66.26 \pm 3.03$ | $45.01 \pm 18.98$ | $60.81 \pm 3.04$ | $68.81 \pm 1.12$ | $78.58 \pm 1.08$ | $-0.23 \pm 0.45$ | $76.96 \pm 1.99$ |
| | 2.0 | $39.06 \pm 17.36$ | $65.40 \pm 7.13$ | $67.89 \pm 1.66$ | $68.38 \pm 1.09$ | $72.91 \pm 0.37$ | $42.18 \pm 3.85$ | $15.51 \pm 2.73$ | $\mathbf{73.74 \pm 0.49}$ |
| | 5.0 | $10.07 \pm 4.91$ | $5.39 \pm 0.03$ | $5.76 \pm 0.84$ | $5.34 \pm 1.61$ | $5.36 \pm 0.28$ | $8.36 \pm 1.91$ | $4.94 \pm 0.66$ | $\mathbf{27.38 \pm 3.87}$ |
| Hopper Gravity | 0.1 | $27.82 \pm 13.41$ | $13.10 \pm 0.98$ | $15.59 \pm 6.09$ | $22.49 \pm 3.71$ | $23.40 \pm 11.62$ | $31.11 \pm 1.80$ | $17.62 \pm 1.66$ | $\mathbf{36.25 \pm 1.50}$ |
| | 0.5 | $28.54 \pm 12.77$ | $23.00 \pm 14.87$ | $23.92 \pm 1.91$ | $12.86 \pm 0.18$ | | $36.37 \pm 2.06$ | $67.06 \pm 3.60$ | $33.57 \pm 6.71$ |
| | 2.0 | $11.84 \pm 2.37$ | $16.10 \pm 1.64$ | $18.62 \pm 6.88$ | $11.76 \pm 0.32$ | $14.65 \pm 2.47$ | $16.44 \pm 1.60$ | $12.09 \pm 0.71$ | $\mathbf{23.79 \pm 2.09}$ |
| | 5.0 | $7.36 \pm 0.13$ | $8.12 \pm 0.16$ | $9.08 \pm 1.15$ | $7.77 \pm 0.31$ | $7.90 \pm 1.27$ | $8.11 \pm 0.97$ | $7.48 \pm 0.51$ | $8.06 \pm 0.03$ |
| Hopper Friction | 0.1 | $25.55 \pm 2.69$ | $24.16 \pm 4.50$ | $18.64 \pm 3.37$ | $34.32 \pm 6.79$ | $26.13 \pm 4.24$ | $33.08 \pm 2.53$ | $18.21 \pm 0.85$ | $\mathbf{51.19 \pm 2.56}$ |
| | 0.5 | $25.22 \pm 4.48$ | $23.56 \pm 1.68$ | $19.60 \pm 15.45$ | $12.32 \pm 3.96$ | $26.94 \pm 2.86$ | $38.10 \pm 3.32$ | $18.41 \pm 1.31$ | $\mathbf{41.34 \pm 0.49}$ |
| | 2.0 | $10.32 \pm 0.06$ | $10.15 \pm 0.06$ | $9.89 \pm 0.20$ | $10.99 \pm 0.76$ | $10.15 \pm 0.03$ | $10.20 \pm 0.30$ | $9.71 \pm 0.37$ | $\mathbf{11.00 \pm 0.14}$ |
| | 5.0 | $7.90 \pm 0.06$ | $7.93 \pm 0.01$ | $7.80 \pm 1.04$ | $7.68 \pm 0.19$ | $7.86 \pm 0.05$ | $8.20 \pm 0.36$ | $7.76 \pm 0.26$ | $8.07 \pm 0.04$ |
| Total | | $875.88$ | $901.52$ | $779.14$ | $893.22$ | $890.62$ | $986.13$ | $647.91$ | $\mathbf{1427.26}$ |

tasks. Also, MOBODY outperforms the baseline more in the large-shift setting, demonstrating the effectiveness of our method for exploration and the dynamics learning and the suboptimality of conservative reward regularization or data filtering methods.

**MOBODY improves more when the dynamics shift is larger.** Additionally, in larger shift scenarios, such as HalfCheetah-Friction-0.1, Ant-Friction-5.0, and Walker2d-Friction-5.0, MOBODY achieves significant improvement over baseline methods, which receive very low rewards in the target domain. We also summarize the performance comparison under different shift levels in Figure 5 in Appendix C.3. Existing methods, DARA, BOSA, SRPO and REAG, fail in large shift settings as the reward regularization methods are mainly trained with source data with regularization, resulting in optimizing the policy with the low dynamics-shift transitions and cannot adapt to the large shift target domain, as we mentioned earlier. Thus, such methods lack exploration of high-reward states in the target domain that do not fall within these low dynamics-shift regions, which is more frequent when shift is large.

**Results on MuJoco kinematics/morphology shift**. We also conduct experiments on MuJoco and Adroit with kinematics and morphology shift. Due to page limit, we summarize the results in Figure 3 by summing the normalized score across different tasks. We observe that MOBODY receives a higher overall score. We do not include error bars because the results are aggregated across many tasks, and a single standard deviation is not well-defined at this level of aggregation. We also present all the experimental results for each task in Table 4 and Table 5 in Appendix C.3, showing that our method performs the best in 29 out of 40 tasks and achieves an overall $25\%$ improvement in all tasks.

**DARA and BOSA do not have significant improvements compared with IQL.** The DARA reward augmentation term, based on a KL divergence between source and target dynamics, can become ill-defined when their supports barely overlap, destabilizing training and sometimes making DARA worse than IQL. For BOSA, relying on a target dynamics model trained only on 5,000 target transitions makes accurate dynamics learning difficult, thereby degrading performance.

In a few settings, MOBODY slightly underperforms SRPO or other baselines. SRPO assumes optimal policies across dynamics often induce similar stationary state distributions. When this holds

in some tasks, it yields strong performance there, but bad performance when the assumption is violated. However, MOBODY outperforms SRPO on most tasks, indicating better robustness under broader dynamics shifts. In the remaining MOBODY-underperforming cases, the gap to the best baseline is small (less than $1.7\%$), typically either because all methods fail and achieve very low rewards (e.g., Hopper-Gravity-5.0), or because baselines already perform very well under small shifts (e.g., HalfCheetah-Friction-0.1, Walker2d-Friction-0.1), and additional exploration benefits from MOBODY is less pronounced is these settings.

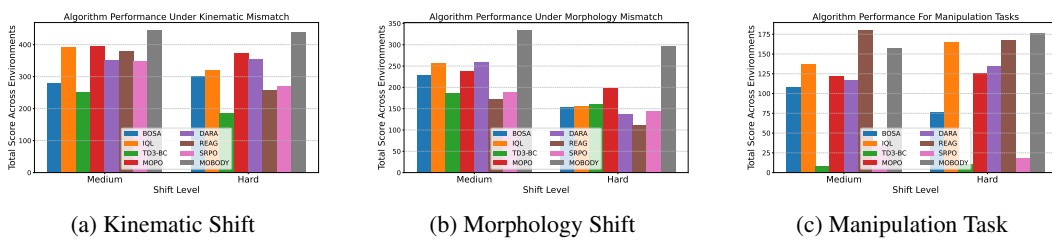

(a) Kinematic Shift  (b) Morphology Shift  (c) Manipulation Task

Figure 3: Aggregation experimental results on MuJoco kinematic and morphology shift task, and Manipulation tasks. Our method significantly outperforms the baselines besides Manipulation medium task. Detailed results of each environment, shift type, and shift level are referred to Table 4 and Table 5 in the Appendix C.3.

## 4.3 ABLATION STUDY

In Section 4.3.1, we first conduct ablation studies on two main components of MOBODY: dynamics learning and policy learning, showing the necessity of each component. In Section 4.3.2, we compare our dynamics learning with others to demonstrate the effectiveness and better generalization ability. In Section 4.3.3, we then conduct ablation studies to analyze how each component contributes to the overall performance and show that the model-based rollout from our learned dynamics is the main contributor, while the target-Q weighted BC loss is an auxiliary but essential regularizer that further improves AWR BC loss. In Section 4.3.4, we also conduct other ablation studies to validate the choice of target-Q weighted BC loss over the standard IQL weight.

### 4.3.1 ANALYSIS OF THE EFFECTIVENESS OF EACH COMPONENT

We evaluate the overall effectiveness of each component, then analyze specific design choices. For dynamic learning, we assess the impact of the cycle transition loss and representation learning. For policy learning, we examine the effectiveness of the Q-weighted loss.

We first evaluate the performance of our proposed dynamics learning and policy learning by replacing the dynamics learning with the existing dynamics learning model or the policy learning with the existing offline RL algorithm. We denote the two ablation studies as follows:

**A1: Replace dynamics learning** We compare our MOBODY with a variant replacing the dynamics learning with the existing model-based method. We use a black-box dynamics model trained on **target data only**, while the policy learning follows the same method as in MOBODY. Table 2 demonstrates that the **A1** variant is significantly degraded compared with our proposed MOBODY algorithm in Walker2d. This indicates that only using the existing dynamics models trained on the target data is insufficient to rollout trajectories in the target domain. This motivates us to propose a novel dynamics model learning method.

**A2: Replace policy learning** Similar to the **A1**, we replace the policy learning with the existing offline RL algorithm. We adopt the same dynamics learning approach as in MOBODY and use Conservative Q-Learning (CQL) (Kumar et al., 2020) for policy learning. Table 2 shows that our proposed MOBODY outperforms the **A2** variant in Walker2d. This demonstrates that the policy learning part of our proposed MOBODY with Q-weight behavior cloning can better utilize the dynamics model compared with the existing method.

Then we delve into the details of the dynamics learning and policy learning part, especially our designs of the loss function and Q-weighting. We have the following ablation studies:

**A3: No Cycle Transition Loss** Here, the dynamics model follows the dynamics learning of the proposed MOBODY, but without the cycle transition loss. We hope to evaluate the effectiveness of our proposed cycle transition loss. Table 2 illustrates that the **A3** suffers degradations compared with our MOBODY in most of the settings. This indicates that the cycle transition loss helps learn a better state representation in our proposed MOBODY method.

**A4: No Q-weighted** Similar to the **A3**, we compare our MOBODY with a variant without the Q-weighted behavior cloning loss. We keep the same dynamics learning method as our proposed MOBODY and replace the Q-weighted behavior cloning loss with the vanilla behavior cloning loss. In Table 2, our method outperforms the method without the Q-weighted behavior cloning in Walker2d. The **A4** underperforms MOBODY in most of the settings except the Walker2d 2.0 level, where all settings have similar performance. This suggests that our proposed Q-weighted approach can help regularize the policy learning in the off-dynamics offline RL scenarios.

Table 2: Performance of the ablation study of our proposed MOBODY method. A1-A4 represent four different ablation studies detailed in Section 4.3. The experiments are conducted on the Walker2d environments under the medium-level with dynamics shifts in gravity and friction in $\{0.1, 0.5, 2.0, 5.0\}$ shift levels. The source domains are the original environments, and the target domains are the environments with dynamic shifts. We report the normalized scores in the target domain with the mean and standard deviation across three random seeds. The higher scores indicate better performance. More experimental results on Hopper are in Table 6 in Appendix C.4.

| Env | Level | Algorithm Ablation | | Loss Ablation | | MOBODY |
|-----|-------|------|------|------|------|--------|
| | | A1 | A2 | A3 | A4 | |
| Walker2d Gravity | 0.1 | $55.23 \pm 10.22$ | $55.43 \pm 5.31$ | $35.34 \pm 10.97$ | $19.53 \pm 4.68$ | $65.85 \pm 5.08$ |
| | 0.5 | $35.66 \pm 3.11$ | $39.98 \pm 1.32$ | $30.63 \pm 2.92$ | $24.44 \pm 1.91$ | $43.57 \pm 2.32$ |
| | 2.0 | $31.94 \pm 5.32$ | $28.58 \pm 5.59$ | $34.42 \pm 3.60$ | $47.13 \pm 2.44$ | $44.32 \pm 4.58$ |
| | 5.0 | $3.56 \pm 0.79$ | $11.37 \pm 3.91$ | $4.42 \pm 1.20$ | $6.43 \pm 0.32$ | $46.05 \pm 20.73$ |
| Walker2d Friction | 0.1 | $24.34 \pm 10.33$ | $25.73 \pm 2.43$ | $21.42 \pm 3.85$ | $19.48 \pm 4.32$ | $28.23 \pm 9.13$ |
| | 0.5 | $56.31 \pm 7.17$ | $73.23 \pm 3.73$ | $68.53 \pm 4.14$ | $61.38 \pm 6.84$ | $76.96 \pm 1.99$ |
| | 2.0 | $60.52 \pm 5.82$ | $71.14 \pm 2.59$ | $67.98 \pm 6.96$ | $76.44 \pm 6.43$ | $73.74 \pm 0.49$ |
| | 5.0 | $4.32 \pm 0.85$ | $18.32 \pm 2.18$ | $5.42 \pm 0.82$ | $7.89 \pm 1.33$ | $27.38 \pm 3.87$ |

### 4.3.2 COMPARISON AMONG DIFFERENT DYNAMICS LEARNING APPROACHES

As a model-based method, MOBODY learns target dynamics that generate higher-quality transitions and yield lower estimation error. We compare against three dynamics-learning baselines: (1) target-only training, (2) combined source+target training, and (3) pretrain-finetune (pretraining on source, then finetuning on target). For a fair comparison, all methods share MOBODY's architecture but use a single action encoder and omit the cycle-transition loss. We evaluate the learned dynamics by the rollout MSE under the MOBODY policy at 1M training steps.

Table 3: Performance comparison using different dynamics learning models. First row: evaluation MSE of the rollout trajectories using different MOBODY policy, second row: normalized score of the policy. We see that MOBODY outperforms the baseline dynamics learning methods in both dynamics learning and overall performance.

| Metric | Task | Trained only on target data | Combined data | Pretrained-finetune | MOBODY |
|--------|------|------|------|------|--------|
| MSE | Walker2d-friction-0.5 | $2.23 \pm 0.26$ | $1.96 \pm 0.68$ | $2.21 \pm 0.19$ | $\mathbf{1.25 \pm 0.39}$ |
| | Walker2d-gravity-0.5 | $2.11 \pm 0.48$ | $1.87 \pm 0.32$ | $2.32 \pm 0.23$ | $\mathbf{1.93 \pm 0.34}$ |
| | Ant-friction-0.5 | $2.99 \pm 0.51$ | $2.01 \pm 0.24$ | $2.14 \pm 0.19$ | $\mathbf{1.88 \pm 0.18}$ |
| | Ant-gravity-0.5 | $1.57 \pm 0.39$ | $1.53 \pm 0.39$ | $1.73 \pm 0.43$ | $\mathbf{1.46 \pm 0.26}$ |
| Normalized Score | Walker2d-friction-0.5 | $56.31 \pm 7.17$ | $41.38 \pm 5.12$ | $62.93 \pm 5.43$ | $\mathbf{76.96 \pm 1.99}$ |
| | Walker2d-gravity-0.5 | $39.71 \pm 3.29$ | $42.13 \pm 3.98$ | $38.13 \pm 3.12$ | $\mathbf{43.57 \pm 2.32}$ |
| | Ant-friction-0.5 | $48.13 \pm 4.43$ | $46.23 \pm 6.85$ | $51.09 \pm 1.93$ | $\mathbf{62.41 \pm 4.10}$ |
| | Ant-gravity-0.5 | $28.32 \pm 3.87$ | $31.39 \pm 3.80$ | $29.69 \pm 7.23$ | $\mathbf{37.44 \pm 2.79}$ |

Table 3 reports both policy performance and rollout MSE for different dynamics-learning strategies, and shows that MOBODY consistently outperforms all baselines in both MSE and reward. This is

mainly because: (1) the target-only dataset is too small to learn accurate dynamics, (2) training on combined source+target data yields a model whose dynamics lie between the two domains rather than matching the target, and (3) the pretrain–finetune paradigm, while effective in supervised domain adaptation, is less suitable for off-dynamics RL, where the conditional next state $s' \mid (s, a)$ fundamentally differs across domains. In contrast, MOBODY explicitly learns shared structure while using separate action encoders to capture the dynamics differences between source and target.

### 4.3.3 MODEL-BASED ROLLOUT IS THE MAIN DRIVER OF IMPROVEMENT, AND TARGET-$Q$ WEIGHTED BC IS ESSENTIAL

To demonstrate that the main performance gains come from dynamics learning, model-based rollouts, and exploration, we compare AWR/IQL-style policy learning with and without MOBODY's rollouts. In Figure 4, we report results for four variants: (B1) IQL-style policy learning without model-based rollouts, (B2) AWR-weighted BC with MOBODY rollouts, (B3) Target-Q–weighted BC without rollouts, and (B4) the full MOBODY method.

By comparing (B1) vs. (B2) and (B3) vs.(B4)—which differ only in whether model-based rollouts are used—we observe substantial gains from incorporating MOBODY's rollouts, highlighting the effectiveness of our learned dynamics. In contrast, comparing (B1) vs. (B3), which differ only in the BC weighting (IQL vs. target-Q), shows no substantial improvement from using target-Q–weighted BC alone, indicating that this auxiliary term is not the main source of performance gains.

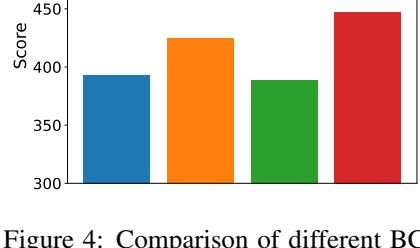

Figure 4: Comparison of different BC losses with and without dynamics learning shows that MOBODY's performance gains primarily stem from its novel dynamics learning and model-based rollouts.

However, we still want to highlight that the improvements arise from the full algorithm working in concert rather than from any single component in isolation, and each component is important. As further supported by Section 4.3.1, where removing the BC weight for the MOBODY will lead to a significant drop.

### 4.3.4 EMPIRICAL VALIDATION OF THE TARGET-Q WEIGHTED BC LOSS

In Figure 4, comparing (B2) and (B4), which differ only in the BC weighting, shows a clear performance gain for the target-Q-weighted BC used in MOBODY. AWR is well-suited for policy improvement (as in IQL), but might be less effective as a regularizer in off-dynamics RL: it is more important to bias the policy toward actions with high target-domain Q-values than high advantages, and when advantages shrink toward zero, it provides little supervision for the BC loss. In addition, the target-Q-weighted BC loss has a simpler form, whereas AWR/IQL requires training an extra value network, complicating optimization in the model-based setting. These results empirically support our choice of target-Q-weighted BC, and we provide a more detailed comparison with AWR/IQL in Appendix C.4.2.

## 5 CONCLUSION

We study the off-dynamics offline reinforcement learning problem through a model-based offline RL method. We introduce MOBODY, a model-based offline RL algorithm that enables policy exploration in the target domain via learned dynamics models. By leveraging shared latent representations across domains, MOBODY effectively learns target dynamics using both source and limited target data. Additionally, we propose a Q-weighted behavior cloning strategy that favors actions with high target Q value, further improving policy learning. Experimental results on MuJoCo and Adroit benchmarks demonstrate that MOBODY consistently outperforms prior methods, particularly in scenarios with significant dynamic mismatches, highlighting its robustness and generalization capabilities. Our method shows the potential of data augmentation in policy learning with a carefully learned dynamics model. Future work includes further investigation on improving the dynamics learning as well as investigation on sparse reward and goal-conditional RL settings.

## REPRODUCIBILITY STATEMENT

Our codes are available at: https://github.com/guoyihonggyh/MOBODY-Model-Based-Off-Dynamics-Offline-Reinforcement-Learning. The implementation of the method is based on the ODRL benchmark repository (Lyu et al., 2024b), which provides the comprehensive dataset and baseline method for evaluation. For our algorithm, we provide detailed information on the training loss for the dynamics learning and the policy optimization in the main text as well as the Algorithm 1 for dynamics learning and Algorithm 2 for policy optimization in Appendix B. We also provide hyperparameter analysis and rule-of-thumb hyperparameters in Appendix C.4, as well as the hyperparameters and model architecture that we used for tuning in Table 11.

## ACKNOWLEDGMENTS

This work is partially supported by a grant from the JHU Center for Digital Health and Artificial Intelligence and a grant from Open Philanthropy. YY and PX are supported in part by the National Science Foundation (DMS-2323112) and the Whitehead Scholars Program at the Duke University School of Medicine. We thank anonymous reviewers for their constructive feedback on this document.

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

## A  RELATED WORK

**Off-dynamics RL.** Off-dynamics RL aims to transfer the policy learned in the source domain to the target domain. One line of work is to regularize the reward of the source data with the target data using the domain classifier. Following this idea, DARC (Eysenbach et al., 2020) and DARAIL (Guo et al., 2024) solve the off-dynamics RL problem in the online paradigm, while DARA (Liu et al., 2022) and REAG (Wang et al., 2026) use the reward regularization techniques in the offline RL setting. Similarly, BOSA (Liu et al., 2024a) regularizes the policy by two support-constrained objectives. SRPO (Xue et al., 2023) regularize the policy through the state visitation frequency on source and target domain. PAR (Lyu et al., 2024a) learns the representation to measure the deviation of dynamic mismatch via the state and state-action encoder to modify the reward. Another line of work is utilizing the data filter method, including the VGDF (Xu et al., 2023) and IGDF (Wen et al., 2024), which filter out the trajectories similar to the target domain and train the RL policies on filtered data. These data filtering or reward regularization methods in off-dynamics offline RL settings cannot explore the target domain substantially, while we propose a novel model-based method that can explore the target domain with the learned *target* dynamics. When no target data are available during training, an orthogonally related line of work studies robust Markov decision processes (MDPs), which aim to achieve reliable performance in an unknown target environment by optimizing policies against worst-case transition dynamics within a predefined uncertainty set (Nilim & Ghaoui, 2003; El Ghaoui & Nilim, 2005; Iyengar, 2005; Xu & Mannor, 2006; Zhou et al., 2021; Liu & Xu, 2024a; Lu et al., 2024; Liu & Xu, 2024b; Zhang et al., 2024; Clavier et al., 2024; Liu et al., 2024b; He et al., 2025; Liu & Xu, 2025); see Liu & Xu (2025) for a recent overview. However, these approaches typically require carefully specified uncertainty sets and can be overly conservative, as they are built on the assumption that no target data are available during training—an assumption that may not hold in some practical scenarios.

**Model-based Offline RL.** Model-based offline RL leverages the strengths of model-based methods in the offline RL paradigm. MOReL (Kidambi et al., 2020) and MOPO (Yu et al., 2020) modify reward functions based on uncertainty estimations derived from ensembles of models. VI-LCB (Rashidinejad et al., 2021) leverages pessimistic value iteration, incorporating penalty functions into value estimation to discourage poorly-covered state-action pairs. COMBO (Yu et al., 2021) provides a conservative estimation without explicitly computing uncertainty, using adversarial training to optimize conservative value estimates. RAMBO (Rigter et al., 2022) further builds upon adversarial techniques by directly training models adversarially with conservatively modified dynamics to reduce distributional shifts. These methods are designed for one domain instead of an off-dynamics RL setting. In this paper, we propose a novel dynamics learning and policy optimization method for an off-dynamics RL setting.

**Representation Learning in RL.** Representation learning (Botteghi et al., 2025) is actively explored in image-based reinforcement learning tasks (Kostrikov et al., 2021b; Yarats et al., 2022; Liu et al., 2021; Zhu et al., 2020) to learn the representation of the image. For model-based RL, to improve sample efficiency, representation has been widely applied to learn the latent dynamics modeling (Karl et al., 2017; Hansen et al., 2022a), latent state representation learning (Barreto et al., 2017; Fujimoto et al., 2021), or latent state-action representation learning (Ota et al., 2020; Ye et al., 2021; Hansen et al., 2022b; Fujimoto et al., 2023). While previous works on representation learning seek to boost the performance through learning the state/state-action representation with representation constraint, such methods might not be suitable or cannot be directly applied to the off-dynamics RL settings as many of them learn the representation without considering transitions or only learns single domain transitions. Thus, in our paper, we learn the shared representation of the state and transition to aid the *target* dynamics learning with source domain data.

## B  ALGORITHM DETAILS

**Reward learning** Note that the reward is modeled as a function of $(s, a, s')$ tuple, as in many tasks, the reward is also related to the next state as mentioned in the Section 2. Also, recall that the reward function in the source and target domain remains the same. Thus, we can learn the reward function with source and target domain data together via the following loss function:

$$L_{\text{reward}} = \frac{1}{2}\mathbb{E}_{D_{\text{src}} \cup D_{\text{trg}}}\left[r(s, a, s') - \hat{r}(s, a, s')\right]^2 + \frac{1}{2}\mathbb{E}_{D_{\text{src}} \cup D_{\text{trg}}}\left[r(s, a, s') - \hat{r}(s, a, \hat{s}')\right]^2, \quad (9)$$

where $\hat{s}'$ is the predicted next state. Here, we use both the true next state and the predicted next state from the dynamics model to learn the reward model, as during inference, we do not have the true next state and only have a predicted next state.

**Uncertainty quantification (UQ) of the transition** To capture the uncertainty of the model, we learn $N = 7$ ensemble transition models, with each model trained independently via Eq.equation 6. We design the UQ of the reward estimation as $u(s,a) := \max_i \text{Std}(\hat{s}'_j) = \max_i \sqrt{1/N \sum_{j=1}^N (\hat{s}'_j - \mathbb{E}(\hat{s}'))^2}$, which is the largest standard deviation among all the state dimensions. This simple and intuitive uncertainty quantification using the ensemble model has been proven simple and effective in many machine learning literature (Parker, 2013) and also model-based RL algorithms (Yu et al., 2020). We find it sufficient to achieve good performance in our experiments by employing the penalized reward $\tilde{r}$ for the downstream policy learning: $\tilde{r}(s,a,s') = \hat{r}(s,a,s') - \beta u(s,a)$.

---

**Algorithm 1** Dynamics Learning via separate action encoders and the representation learning.

1: **Input:** Offline datasets $\mathcal{D}_{\text{src}} = \{(s,a,r,s')\}$, $\mathcal{D}_{\text{trg}} = \{(s,a,r,s')\}$, number of model learning steps $N_{\text{model}}$, target training frequency $K$.
2: **Initialize:** State encoder model $\phi_E$, transition model $\phi_T$, source state action encoder $\psi_{\text{src}}$, target state action encoder $\psi_{\text{trg}}$, reward model $\hat{r}$.
3: **for** $i = 1$ to $N_{\text{model}}$ **do**
4:      **Sample mini-batch:**
5:      **if** $i\% K = 0$ **then**
6:          Sample mini-batch $\{(s,a,r,s')\}$ from $\mathcal{D}_{\text{trg}}$
7:      **else**
8:          Sample mini-batch $\{(s,a,r,s')\}$ from $\mathcal{D}_{\text{src}}$
9:      **end if**
10:      Predict the next state with Eq. equation 1, and equation 2 with mini-batch data.
11:      Optimize the dynamics with the transition loss in Eq.equation 3, encoder loss in Eq.equation 4, cycle transition loss in Eq.equation 5 and reward loss in Eq.equation 9 with mini-batch data.
12: **end for**

---

**Algorithm 2** MOBODY: Model-Based Off-dynamics Offline Reinforcement Learning

1: **Input:** Offline dataset $\mathcal{D}_{\text{src}} = \{(s,a,r,s')\}$ and $\mathcal{D}_{\text{trg}} = \{(s,a,r,s')\}$, $D_{\text{fake}} = \{\}$, number of model learning steps $N_{\text{model}}$, policy training steps $N_{\text{policy}}$.
2: **Initialize:** Dynamics model, policy $\pi_\theta$, rollout length $L_{\text{rollout}}$.
     **Dynamics Training**
3: Learn target dynamics and reward estimation: $\hat{T}_{\text{trg}}, \hat{r}_{\text{trg}} \leftarrow$ Call Algorithm 1
     **Offline Policy Learning**
4: Regularize source data $\mathcal{D}_{\text{src\_aug}} = \{(s,a,r+\eta\Delta r,s')\}$ with DARA.
5: **for** $j = 1$ to $N_{\text{policy}}$ **do**
6:      Collect rollout data from $\hat{T}$ and $\hat{r}_{\text{trg}}$ starting from state in $D_{\text{src\_aug}}$ and $D_{\text{trg}}$. Add batch data to replay buffer $D_{\text{fake}}$.
7:      Sample batch $(s,a,s',r)_{\text{fake}}$ from $\mathcal{D}_{\text{fake}}$, $(s,a,s',r)_{\text{trg}}$ from $\mathcal{D}_{\text{trg}}$ and $(s,a,s',r)_{\text{trg\_aug}}$ from $\mathcal{D}_{\text{src\_aug}}$. Concatenate them as $(s,a,s',r)_{\text{train}}$.
8:      Learn the Q value function with Eq.equation 7
9:      Update policy $\pi_\theta$ with Eq. equation 8
10: **end for**
11: **Return:** Learned policy $\pi_\theta$

---

# C   EXPERIMENTAL DETAILS

## C.1   THE DARA REGULARIZATION FOR SOURCE DATA USED IN MOBODY

Note that in MOBODY, we use DARA to regularize the reward in the source data. In this section, we introduce the details of DARA.

DARA (Liu et al., 2022), the offline version of DARC (Eysenbach et al., 2020), trains the domain classifiers to calculate the reward penalty term $\Delta r(s, a, s')$ and regularize the rewards in the source domain dataset via:

$$\hat{r}_{\text{DARA}}(s, a, s') = r(s, a, s') + \eta \Delta r(s, a, s'),$$

where $\eta$ is the penalty coefficient, where we set to $0.1$ following the ODRL benchmark (Lyu et al., 2024b).

**Estimation of the $\Delta r$.** Following the DARC (Eysenbach et al., 2020; Liu et al., 2022), the reward regularization $\Delta r$ can be estimated with the following two binary classifiers $p(\text{trg}|s_t, a_t)$ and $p(\text{trg}|s_t, a_t, s_{t+1})$ with Bayes' rules:

$$p(\text{trg}|s_t, a_t, s_{t+1}) = p_{\text{trg}}(s_{t+1}|s_t, a_t)p(s_t, a_t|\text{trg})p(\text{trg})/p(s_t, a_t, s_{t+1}), \tag{10}$$

$$p(s_t, a_t|\text{trg}) = p(\text{trg}|s_t, a_t)p(s_t, a_t)/p(\text{trg}). \tag{11}$$

Replacing the $p(s_t, a_t|\text{trg})$ in Eq. equation 10 with Eq. equation 11, we obtain:

$$p_{\text{trg}}(s_{t+1}|s_t, a_t) = \frac{p(\text{trg}|s_t, a_t, s_{t+1})p(s_t, a_t, s_{t+1})}{p(\text{trg}|s_t, a_t)p(s_t, a_t)}.$$

Similarly, we can obtain the $p_{\text{src}}(s_{t+1}|s_t, a_t) = \frac{p(\text{src}|s_t, a_t, s_{t+1})p(s_t, a_t, s_{t+1})}{p(\text{src}|s_t, a_t)p(s_t, a_t)}$.

We can calculate the $\Delta r(s_t, a_t, s_{t+1})$ following:

$$\begin{aligned}
\Delta r(s_t, a_t, s_{t+1}) &= \log \left( \frac{p_{\text{trg}}(s_{t+1}|s_t, a_t)}{p_{\text{src}}(s_{t+1}|s_t, a_t)} \right) \\
&= \log p(\text{trg}|s_t, a_t, s_{t+1}) - \log p(\text{trg}|s_t, a_t) + \log p(\text{src}|s_t, a_t, s_{t+1}) - \log p(\text{src}|s_t, a_t).
\end{aligned}$$

**Training the Classifier $p(\text{trg}|s_t, a_t)$ and $p(\text{trg}|s_t, a_t, s_{t+1})$.** The two classifiers are parameterized bu $\theta_{\text{SA}}$ and $\theta_{\text{SAS}}$. To update the two classifiers, we sample one mini-batch of data from the source replay buffer $D_{\text{src}}$ and the target replay buffer $D_{\text{src}}$ respectively. Imbalanced data is considered here as each time we sample the same amount of data from the source and target domain buffer. Then, the parameters are learned by minimizing the standard cross-entropy loss:

$$\begin{aligned}
\mathcal{L}_{\text{SAS}} &= -\mathbb{E}_{\mathcal{D}_{\text{src}}} \left[ \log p_{\theta_{\text{SAS}}}(\text{trg}|s_t, a_t, s_{t+1}) \right] - \mathbb{E}_{\mathcal{D}_{\text{trg}}} \left[ \log p_{\theta_{\text{SAS}}}(\text{trg}|s_t, a_t, s_{t+1}) \right], \\
\mathcal{L}_{\text{SA}} &= -\mathbb{E}_{\mathcal{D}_{\text{src}}} \left[ \log p_{\theta_{\text{SA}}}(\text{trg}|s_t, a_t, s_{t+1}) \right] - \mathbb{E}_{\mathcal{D}_{\text{trg}}} \left[ \log p_{\theta_{\text{SA}}}(\text{trg}|s_t, a_t, s_{t+1}) \right].
\end{aligned}$$

Thus, $\theta = (\theta_{\text{SAS}}, \theta_{\text{SA}})$ is obtained from:

$$\begin{aligned}
\theta &= \arg\min_{\theta} \mathcal{L}_{CE}(\mathcal{D}_{\text{src}}, \mathcal{D}_{\text{trg}}) \\
&= \arg\min_{\theta} [\mathcal{L}_{\text{SAS}} + \mathcal{L}_{\text{SA}}].
\end{aligned}$$

## C.2 TECHNICAL DETAILS ABOUT BASELINE ALGORITHMS

In this section, we introduce the baselines in detail and the implementation follows the ODRL benchmark (Lyu et al., 2024b).

**BOSA** (Liu et al., 2024a). BOSA shows a distribution shift issue might exist when learning policies from the two domain offline data under dynamics mismatch. It handles the out-of-distribution (OOD) state actions pair through a supported policy optimization and addresses the OOD dynamics issue through a supported value optimization by data filtering. Specifically, the policy is updated with:

$$\mathcal{L}_{\text{actor}} = \mathbb{E}_{s \sim \mathcal{D}_{\text{src}} \cup \mathcal{D}_{\text{trg}}, a \sim \pi_\phi(s)} \left[ Q(s, a) \right], \quad \text{s.t.} \quad \mathbb{E}_{s \sim \mathcal{D}_{\text{src}} \cup \mathcal{D}_{\text{trg}}} \left[ \hat{\pi}_{\theta_{\text{offline}}}(\pi_\theta(s) \mid s) \right] > \epsilon.$$

Here, the $\epsilon$ is the threshold, $\hat{\pi}_{\theta_{\text{offline}}}$ is the learned policy for the combined offline dataset. The value function is updated with:

$$\begin{aligned}
\mathcal{L}_{\text{critic}} &= \mathbb{E}_{(s,a) \sim \mathcal{D}_{\text{src}}} \left[ Q(s, a) \right] \\
&\quad + \mathbb{E}_{(s,a,r,s') \sim \mathcal{D}_{\text{src}} \cup \mathcal{D}_{\text{trg}}, a' \sim \pi_\phi(\cdot|s)} \left[ I\left( \hat{p}_{\text{trg}}(s'|s, a) > \epsilon' \right) \left( Q_{\theta_i}(s, a) - y \right)^2 \right],
\end{aligned}$$

where $I(\cdot)$ is the indicator function, $\hat{p}_{\text{trg}}(s'|s,a) = \arg\max E_{(s,a,s')\sim D_{\text{trg}}}[\log\hat{p}_{\text{trg}}(s'|s,a)]$ is the estimated target domain dynamics, $\epsilon'$ is the threshold.

**IQL** (Kostrikov et al., 2021a). IQL learns the state value function and state-action value function simultaneously by expectile regression:

$$\mathcal{L}_V = \mathbb{E}_{(s,a)\sim\mathcal{D}_{\text{src}}\cup\mathcal{D}_{\text{trg}}}\left[L_2^\tau(Q_\theta(s,a) - V_\psi(s))\right],$$

where $L_2^\tau(u) = |\tau - I(u < 0)|\|u\|^2$, $I(\cdot)$ is the indicator function, and $\theta$ is the target network parameter. The state-action value function is then updated by:

$$\mathcal{L}_Q = \mathbb{E}_{(s,a,r,s')\sim\mathcal{D}_{\text{src}}\cup\mathcal{D}_{\text{trg}}}\left[\left(r(s,a) + \gamma V_\psi(s') - Q_\theta(s,a)\right)^2\right].$$

The advantage function is $A(s,a) = Q(s,a) - V(s)$. The policy is optimized by the advantage-weighted behavior cloning:

$$\mathcal{L}_{\text{actor}} = \mathbb{E}_{(s,a)\sim\mathcal{D}_{\text{src}}\cup\mathcal{D}_{\text{trg}}}\left[\exp(\beta\cdot A(s,a))\log\pi_\phi(a|s)\right],$$

where $\beta$ is the inverse temperature coefficient.

**TD3-BC** (Fujimoto & Gu, 2021). TD3-BC is an effective model-free offline RL approach that incorporates a behavior cloning regularization term to the objective function of the vanilla TD3, which gives:

$$\mathcal{L}_{\text{actor}} = \lambda\cdot\mathbb{E}_{s\sim\mathcal{D}_{\text{src}}\cup\mathcal{D}_{\text{trg}}}\left[Q(s,\pi_\theta(s))\right] + \mathbb{E}_{(s,a)\sim\mathcal{D}_{\text{src}}\cup\mathcal{D}_{\text{trg}}}\left[(a - \pi_\theta(s))^2\right],$$

where

$$\lambda = \frac{\nu}{\frac{1}{N}\sum_{(s_j,a_j)}Q(s_j,a_j)} \quad\text{and}\quad \nu\in\mathbb{R}^+$$

is the normalization coefficient.

**MOPO** (Yu et al., 2020). MOPO is a standard model-based offline policy optimization method, which learns dynamics first and penalizes rewards by the uncertainty of the dynamics. Lastly, it optimizes a policy with the SAC (Haarnoja et al., 2018). Specifically, following previous off-dynamics work (Eysenbach et al., 2020) in the online setting that applies MBPO as a baseline, we learn the dynamics with the combined offline source and target data. We follow the implementation in OfflineRL-kit.

**DARA** (Liu et al., 2022). We refer to the Appendix C.1 for the details. We follow the implementation in ODRL (Lyu et al., 2024b).

**SRPO** (Xue et al., 2023). SRPO proposes state-level regularization by leveraging the observation that optimal policies across related dynamics often induce similar stationary state distributions. We follow the practical implementation in SRPO (Xue et al., 2023) to train a discriminator to distinguish high-value states from low-value states and then augment the reward. A simplified core objective of SRPO can be written as:

$$\max_\pi \ \mathbb{E}_{\tau\sim\pi}\left[\sum_{t=0}^\infty \gamma^t r(s_t,a_t)\right] \quad\text{s.t.}\quad D_{\text{KL}}\big(d_\pi(\cdot)\,\|\,\zeta(\cdot)\big)\leq\varepsilon,$$

which, via Lagrangian relaxation, leads to a reward-shaped objective:

$$\mathcal{L}(\pi) = \mathbb{E}_{\tau\sim\pi}\left[\sum_{t=0}^\infty \gamma^t\left(r(s_t,a_t) + \lambda\log\frac{\zeta(s_t)}{d_\pi(s_t)}\right)\right].$$

In practice, the density ratio $\frac{\zeta(s)}{d_\pi(s)}$ is estimated with a discriminator $D(s)$ trained in a GAN, yielding:

$$\frac{\zeta(s)}{d_\pi(s)} \approx \frac{D(s)}{1 - D(s)}.$$

**REAG** (Wang et al., 2026). REAG proposes that reward augmentation methods can not be directly applied to return-conditioned supervised learning methods like DT (Chen et al., 2021). It introduces

a return matching method to address this problem. We follow the description in REAG to implement the methods:

$$\pi_S = \arg\min_\pi \hat{L}(\pi) := -\sum_{\tau \in \mathcal{D}_S} \sum_{t=1}^{H} \log \pi\big(a_t \mid s_t, \psi(g(\tau))\big),$$

where $g(\tau) = \sum_{t=1}^{H} r_t$ is the original cumulative return of trajectory $\tau$, and $\psi(\cdot)$ is the return augmentation function chosen so that $\pi_S$ approximates the optimal policy in the target domain. Using dynamics-aware reward augmentation (DARA), $\text{REAG}^*_{\text{Dara}}$ defines a per-state transformed return-to-go at step t as:

$$\psi\big(g_t(\tau)\big) := \sum_{h=t}^{H} r_h \; + \; \eta \sum_{h=t}^{H} \Delta r(s_h, a_h, s_{h+1}).$$

By directly matching return distributions, $\text{REAG}^*_{\text{MV}}$ leverages the Q-values trained by CQL (Kumar et al., 2020) to augment the source dataset. In our experiments, we employ this variant and denote it as **REAG**.

## C.3 Additional Experimental Results

In this section, we present additional results on various types of dynamic shifts, including Kinematic shift (kin) and Morphology shift (morph), on Mujoco and Adroit, following the ODRL benchmark. We present the results in Table 4 and Table 5, which are the detailed results of the Figure 3. We observe that our method outperforms the baseline methods in most cases, indicating that it is applicable to various types of dynamic shifts and environments.

Table 4: Performance comparison on HalfCheetah, Ant, Walker2d, and Hopper environments with kinematic and morphology shift. Our method performs best in 25 out of 32 total tasks and receives an overall 25% improvement over baselines. We use **M** and **H** to represent the medium and hard levels of the dynamics shift.

| Env | Type | Level | BOSA | IQL | TD3-BC | DARA | MOPO | REAG | SRPO | MOBODY |
|-----|------|-------|------|-----|--------|------|------|------|------|--------|
| HalfCheetah | morph-thigh | M | 22.83 ± 0.03 | 20.49 ± 0.50 | 19.49 ± 0.50 | 10.90 ± 0.43 | 17.32 ± 1.80 | 13.86±0.79 | 9.19 ± 1.91 | **27.18 ± 6.80** |
| | | H | 20.77 ± 0.66 | 21.69 ± 0.58 | 22.19 ± 1.08 | 10.35 ± 2.10 | 25.33 ± 2.23 | 9.95±0.93 | 21.65 ± 2.91 | **28.51 ± 9.20** |
| | morph-torso | M | 1.67 ± 0.87 | 1.87 ± 0.80 | 5.86 ± 0.21 | 2.91 ± 0.08 | 10.65 ± 4.86 | 2.62±0.14 | -0.16 ± 0.05 | **23.92 ± 12.24** |
| | | H | 17.09 ± 15.71 | 27.81 ± 3.14 | 2.73 ± 1.25 | 29.41 ± 7.88 | 32.78 ± 4.19 | 5.60±6.17 | 29.42 ± 5.02 | **40.45 ± 1.26** |
| | kin-footjnt | M | 36.79 ± 0.92 | 34.71 ± 0.72 | 30.19 ± 3.73 | 33.48 ± 0.34 | 32.49 ± 4.02 | 35.65±0.66 | **39.24 ± 1.21** | 31.88 ± 3.70 |
| | | H | 14.70 ± 0.92 | 31.68 ± 2.35 | 14.05 ± 2.96 | 31.19 ± 4.08 | 33.47 ± 5.61 | 31.08±0.43 | **35.64 ± 4.57** | 18.51 ± 7.30 |
| | kin-thighjnt | M | 14.92 ± 0.01 | 41.27 ± 3.16 | 41.77 ± 2.66 | 15.47 ± 0.62 | 38.33 ± 8.68 | 29.22±3.38 | 54.28 ± 2.26 | **59.17 ± 0.85** |
| | | H | 31.72 ± 0.17 | 31.60 ± 9.36 | 31.10 ± 9.86 | 31.46 ± 2.31 | 30.35 ± 2.93 | 22.16±0.60 | 7.48 ± 1.20 | **56.72 ± 0.08** |
| Ant | morph-halflegs | M | 49.94 ± 5.98 | 73.65 ± 2.70 | 46.60 ± 6.24 | 70.66 ± 3.36 | 66.32 ± 5.29 | 58.92±2.02 | 49.39 ± 3.61 | **79.25 ± 0.61** |
| | | H | 58.40 ± 3.41 | 57.51 ± 1.25 | 45.07 ± 2.82 | 58.46 ± 4.45 | 39.44 ± 8.57 | 47.96±10.59 | 59.42 ± 3.99 | **63.76 ± 3.27** |
| | morph-alllegs | M | 72.02 ± 3.57 | 61.12 ± 9.73 | 47.18 ± 6.89 | 64.83 ± 4.49 | 49.19 ± 5.32 | 30.77±0.37 | 53.58 ± 6.94 | **75.24 ± 7.85** |
| | | H | 18.50 ± 4.33 | 10.44 ± 0.51 | 14.53 ± 3.74 | 4.47 ± 6.18 | 12.71 ± 1.66 | 11.71±1.24 | 2.31 ± 0.17 | **24.13 ± 0.10** |
| | kin-anklejnt | M | 72.06 ± 4.63 | **77.60 ± 3.35** | 44.72 ± 15.96 | 75.43 ± 2.03 | 74.31 ± 1.92 | 73.62±0.34 | 70.34 ± 5.83 | 74.92 ± 6.46 |
| | | H | 63.78 ± 7.97 | 62.95 ± 7.88 | 66.22 ± 26.98 | 61.06 ± 4.92 | 63.28 ± 11.01 | 65.66±9.17 | 59.87 ± 7.02 | **76.97 ± 8.36** |
| | kin-hipjnt | M | 38.52 ± 5.88 | **60.97 ± 1.72** | 26.85 ± 4.26 | 55.73 ± 1.93 | 48.91 ± 12.65 | 45.66±3.91 | 23.50 ± 1.03 | 54.75 ± 4.58 |
| | | H | 50.57 ± 4.89 | 59.31 ± 2.92 | 33.85 ± 5.59 | 58.47 ± 3.42 | 52.87 ± 2.99 | 41.88±3.16 | 33.88 ± 5.16 | **59.61 ± 3.11** |
| Walker | morph-torso | M | 8.26 ± 4.83 | 12.35 ± 1.45 | 18.93 ± 9.36 | 15.79 ± 1.33 | 22.81 ± 13.78 | 8.07±0.63 | 11.89 ± 2.09 | **38.67 ± 2.05** |
| | | H | 1.61 ± 0.12 | 2.30 ± 0.58 | 1.54 ± 0.44 | 3.32 ± 1.13 | 9.92 ± 3.36 | 1.66±0.14 | 2.55 ± 1.31 | **11.96 ± 5.41** |
| | morph-leg | M | 46.70 ± 8.39 | 41.12 ± 13.58 | 22.24 ± 9.95 | 39.71 ± 13.67 | 44.33 ± 6.66 | 35.96±15.75 | 40.23 ± 6.03 | **57.57 ± 2.00** |
| | | H | 14.37 ± 3.34 | 16.15 ± 3.70 | 49.07 ± 2.38 | 13.13 ± 1.24 | 19.62 ± 0.71 | 14.86±4.02 | 11.17 ± 0.84 | **49.12 ± 0.52** |
| | kin-footjnt | M | 17.99 ± 1.15 | 56.62 ± 12.10 | 43.31 ± 20.48 | 55.81 ± 1.36 | 57.92 ± 5.95 | 41.63±15.27 | 51.73 ± 4.14 | **67.56 ± 3.05** |
| | | H | 25.76 ± 15.99 | 6.52 ± 1.61 | 26.34 ± 13.24 | 9.63 ± 0.91 | 37.21 ± 20.52 | 12.17±6.76 | 50.39 ± 2.92 | **57.93 ± 0.37** |
| | kin-thighjnt | M | 47.63 ± 27.26 | 61.28 ± 14.24 | 35.64 ± 11.74 | 56.28 ± 13.79 | 68.11 ± 3.60 | 68.31±4.15 | 47.72 ± 7.65 | **69.48 ± 4.22** |
| | | H | 48.66 ± 14.73 | 51.66 ± 2.05 | 43.88 ± 11.54 | 63.76 ± 2.06 | 73.52 ± 7.92 | 64.44±10.75 | 32.42 ± 2.08 | **78.14 ± 2.50** |
| Hopper | morph-foot | M | 12.67 ± 0.00 | 32.99 ± 0.16 | 12.69 ± 0.43 | **40.61 ± 1.64** | 12.96 ± 0.14 | 12.76±0.05 | 11.81 ± 3.02 | 13.05 ± 0.48 |
| | | H | 10.13 ± 0.62 | 11.78 ± 0.09 | 14.15 ± 4.30 | 13.32 ± 1.48 | 47.19 ± 12.77 | 11.33±0.05 | 9.17 ± 0.87 | **65.02 ± 11.98** |
| | morph-torso | M | 15.88 ± 1.18 | 13.38 ± 0.05 | 13.94 ± 0.75 | 13.29 ± 0.19 | 14.04 ± 0.35 | 10.01±4.78 | 12.12 ± 1.96 | **20.23 ± 1.29** |
| | | H | 11.73 ± 0.33 | 7.77 ± 3.73 | 11.54 ± 0.81 | 4.15 ± 0.05 | 11.83 ± 0.28 | 7.51±3.47 | 9.56 ± 1.79 | **12.34 ± 0.20** |
| | kin-legjnt | M | 36.51 ± 1.51 | 42.28 ± 0.08 | 11.76 ± 4.60 | 44.67 ± 0.58 | 43.57 ± 0.80 | **69.87±11.61** | 47.94 ± 6.71 | 54.89 ± 0.26 |
| | | H | 36.13 ± 1.70 | 45.02 ± 4.08 | 18.87 ± 1.46 | **65.44 ± 4.10** | 50.38 ± 3.74 | 55.42±4.66 | 24.75 ± 1.36 | 56.88 ± 3.68 |
| | kin-footjnt | M | 14.92 ± 0.01 | 15.58 ± 0.11 | 17.09 ± 0.04 | 15.47 ± 0.62 | 31.33 ± 16.25 | 14.85±0.63 | 12.87 ± 0.82 | **33.94 ± 14.81** |
| | | H | 31.72 ± 0.17 | 32.41 ± 0.16 | 32.21 ± 0.00 | 32.99 ± 0.78 | 33.21 ± 0.07 | 16.51±12.05 | 25.66 ± 4.54 | **33.35 ± 0.89** |
| Total | | | 964.95 | 1123.88 | 865.60 | 1101.65 | 1205.70 | 917.67 | 951.01 | **1515.10** |

Figure 5 summarizes the normalized scores across all environments under different shift levels on MuJoco gravity and friction shift settings. In Figure 5a, MOBODY consistently outperforms baselines under gravity shifts, with especially large gains at the more challenging and larger shift levels on 0.1 and 5.0, as MOBODY can explore more of the environment with the learned dynamics. A similar trend is observed in Figure 5b, where MOBODY again outperforms all baselines, with greater improvements in the larger shift (0.1 and 5.0) compared to the smaller ones (0.5 and 2.0). Existing methods, DARA and BOSA, fail in large shift settings as the reward regularization methods

Table 5: Performance comparison on Pen and Door tasks. Our method is the best or second best in 6 out of 8 tasks. We use **M** and **H** to represent the medium and hard levels of the dynamics shift.

| Env | Type | Level | BOSA | IQL | TD3-BC | DARA | MOPO | REAG | SRPO | MOBODY |
|---|---|---|---|---|---|---|---|---|---|---|
| Pen | kin-broken-jnt | M | 30.63 ± 9.01 | 24.34 ± 15.49 | 6.86 ± 6.63 | 38.60 ± 3.44 | 37.99 ± 7.46 | **62.36±2.60** | 23.41±2.39 | 37.67 ± 4.54 |
| | | H | 7.18 ± 2.02 | 7.74 ± 3.48 | 1.31 ± 1.29 | 9.41 ± 6.06 | 8.14 ± 2.92 | **26.30±6.49** | 8.10±5.75 | 13.73 ± 6.32 |
| | morph-shrink-finger | M | 10.72 ± 6.65 | 13.75 ± 4.91 | 2.20 ± 1.71 | 8.72 ± 3.12 | 3.48 ± 0.85 | **18.06±3.52** | 7.60±3.73 | 16.48 ± 10.46 |
| | | H | 11.78 ± 6.57 | 32.16 ± 1.14 | 9.12 ± 9.03 | 22.17 ± 3.90 | 28.89 ± 2.48 | 18.38±4.59 | 8.76±6.82 | **37.80 ± 1.18** |
| Door | kin-broken-joint | M | 25.42 ± 22.04 | 37.43 ± 12.76 | -0.23 ± 0.01 | 20.18 ± 5.29 | 27.90 ± 7.92 | 38.86±7.63 | 0.67±1.25 | **39.26 ± 3.72** |
| | | H | 30.64 ± 26.87 | 56.02 ± 7.74 | -0.12 ± 0.02 | 58.22 ± 9.91 | 57.45 ± 9.58 | 60.22±5.23 | 0.56±1.25 | **61.61 ± 9.84** |
| | morph-shrink-finger | M | 41.59 ± 5.95 | 60.74 ± 12.83 | -0.19 ± 0.01 | 50.32 ± 4.78 | 52.02 ± 1.74 | 61.45±0.68 | 1.44±0.98 | **63.67 ± 9.52** |
| | | H | 26.97 ± 8.62 | **68.64 ± 8.34** | -0.20 ± 0.02 | 44.22 ± 7.19 | 67.06 ± 1.96 | 62.66±1.60 | 0.76±0.62 | 62.88 ± 5.25 |
| Total | | | 184.93 | 300.82 | 18.75 | 251.84 | 282.93 | **348.28** | 51.30 | 333.10 |

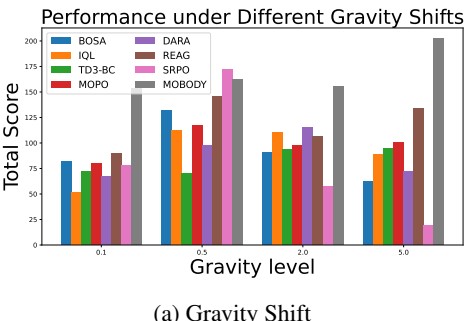

(a) Gravity Shift

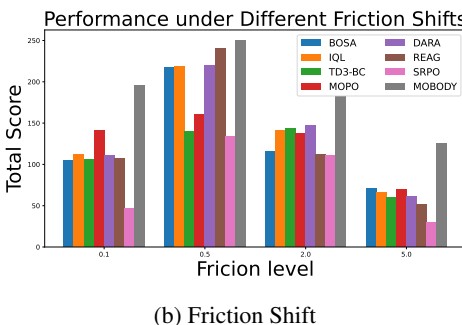

(b) Friction Shift

Figure 5: Performance of our MOBODY and baselines in different dynamics shift with various shift levels $\{0.1, 0.5, 2.0, 5.0\}$. The scores are summed over all the environments (HalfCheetah, Ant, Walker2D, and Hopper) in the target domain. We directly compare the algorithms in the same dynamics shift levels. The higher scores indicate better performance. We can observe a larger improvement for larger shift cases (0.1 and 5.0).

cannot account for the large shift, as they are mainly trained with source data with regularization, thus usually receive high rewards in the source domain, but don't really adapt to the target domain, especially in large shifts, as the policy gets different rewards. Also, they lack the exploration of the target domain.

## C.4 ADDITIONAL ABLATION STUDY RESULTS

In this section, we provide additional ablation studies to empirically justify the design of each component and the effectiveness of MOBODY.

### C.4.1 ADDITIONAL RESULTS ON SECTION 4.3

First, we present additional ablation studies results on Hopper as we mentioned in Section 4.3. Same as Table 2, we evaluate the overall effectiveness of each component (A1 and A2) and then analyze specific design choices (A3 and A4). For dynamics learning, we assess the impact of the cycle transition loss and representation learning. For policy learning, we examine the effectiveness of the Q-weighted loss. We draw the same conclusion as Section 4.3.

### C.4.2 COMPARISON OF USING IQL WEIGHT AND TARGET Q WEIGHTED FOR BC LOSS.

We empirically justify our target Q-weighted BC loss and conduct additional experiments showing that target-Q weighted BC performs better than IQL weight.

The target Q-weighted BC loss is inspired by the advantage-weighted regression (AWR) (Kostrikov et al., 2021a) and the reward-weighted regression (RWR) (Peters et al., 2010). While the AWR is built on the RWR and we follow the idea of RWR by replacing the Monte-Carlo estimation on the reward with the target Q value.

Table 6: Performance of ablation study of our proposed MOBODY method. The experiments are conducted on the Hopper environments under the medium-level with dynamics shifts in gravity and friction in $\{0.1, 0.5, 2.0, 5.0\}$ shift levels. The source domains are the original environments and the target domains are the environments with dynamics shifts. We report the normalized scores in the target domain with the mean and standard deviation across three random seeds. The higher scores indicate better performance.

| Env | Level | Algorithm Ablation | | Loss Ablation | | MOBODY |
|---|---|---|---|---|---|---|
| | | A1 | A2 | A3 | A4 | |
| Hopper Gravity | 0.1 | $14.53 \pm 2.81$ | $28.07 \pm 4.12$ | $33.65 \pm 3.21$ | $11.54 \pm 1.12$ | $36.25 \pm 1.50$ |
| | 0.5 | $28.83 \pm 3.32$ | $25.70 \pm 1.86$ | $23.52 \pm 3.33$ | $20.11 \pm 1.26$ | $33.57 \pm 6.71$ |
| | 2.0 | $10.64 \pm 1.92$ | $12.32 \pm 5.21$ | $10.90 \pm 1.29$ | $16.40 \pm 4.12$ | $23.79 \pm 2.09$ |
| | 5.0 | $8.12 \pm 0.69$ | $8.23 \pm 1.92$ | $8.79 \pm 0.94$ | $8.89 \pm 1.01$ | $8.06 \pm 0.03$ |
| Hopper Friction | 0.1 | $26.09 \pm 4.75$ | $35.14 \pm 7.97$ | $24.42 \pm 2.86$ | $20.07 \pm 10.32$ | $51.19 \pm 2.56$ |
| | 0.5 | $22.42 \pm 3.32$ | $31.31 \pm 5.08$ | $29.26 \pm 6.02$ | $27.07 \pm 3.73$ | $41.34 \pm 0.49$ |
| | 2.0 | $10.64 \pm 0.32$ | $9.41 \pm 1.03$ | $10.31 \pm 0.12$ | $8.47 \pm 1.13$ | $11.00 \pm 0.14$ |
| | 5.0 | $8.43 \pm 1.32$ | $7.52 \pm 0.29$ | $8.14 \pm 0.91$ | $7.55 \pm 1.02$ | $8.07 \pm 0.04$ |

Table 7: Comparison of using IQL-style weights for the BC loss versus the target-Q-weighted BC loss (MOBODY).

| Env | Shift | IQL weight for BC | MOBODY |
|---|---|---|---|
| Halfcheetah | gravity-0.5 | $45.13 \pm 2.21$ | $\mathbf{47.18 \pm 1.23}$ |
| Halfcheetah | gravity-2.0 | $40.42 \pm 3.63$ | $\mathbf{41.46 \pm 7.35}$ |
| Halfcheetah | friction-0.5 | $64.82 \pm 1.53$ | $\mathbf{69.54 \pm 0.48}$ |
| Halfcheetah | friction-2.0 | $\mathbf{51.30 \pm 2.53}$ | $50.02 \pm 3.26$ |
| Walker2d | gravity-0.5 | $\mathbf{48.45 \pm 2.33}$ | $43.57 \pm 2.32$ |
| Walker2d | gravity-2.0 | $39.67 \pm 3.21$ | $\mathbf{44.32 \pm 4.58}$ |
| Walker2d | friction-0.5 | $63.47 \pm 5.01$ | $\mathbf{76.96 \pm 1.99}$ |
| Walker2d | friction-2.0 | $71.39 \pm 1.81$ | $\mathbf{73.74 \pm 0.49}$ |

Here, we test a more standard IQL-style weighting, which is the advantage-weighted regression built on reward-weighted regression, i.e., using IQL weights $\exp(\beta(Q(s,a) - V(s)))$, and report the results in the following table. We see that our Q-weighted BC loss slightly improves the performance. The reason might be 1) training an extra value network here using model-based rollout introduces more complexity and noise, leading to worse performance.

Since the IQL-style weighting (AWR) is worse empirically while requiring training an additional value network (thus increasing complexity), we adopt a simpler exponential weighting directly on the Q-value, inspired by IQL and reward-weighted regression (RWR). We further normalize Q by its average absolute value, following TD3-BC, because the trade-off between policy optimization and BC is highly sensitive to the scale of rewards and Q-values; this normalization yields a more stable and comparable weighting across tasks.

### C.4.3 COMPARISON OF DIFFERENT TARGET DATA SIZE

In Table 8, we further provide additional experimental results with varying amounts of target data, including 500, 1,000, 2,000, and 5,000 transitions. We observe performance degradation in MOBODY as the target dataset size decreases. But this decrease is not that significant, showing our method also works well when the target data size is small.

### C.5 HYPERPARAMETERS ANALYSIS AND COMPUTATIONAL COST

**Hyperparameter Analysis**. We conducted two hyperparameter analyses: the BC loss weight and uncertainty penalty of the model-based method in the policy learning part, as detailed in Table 9. We can see that these parameters are important in the performance of BC loss and need to be tuned across

Table 8: Performance of MOBODY under different numbers of target transitions in HalfCheetah. "5,000" corresponds to the original setting in the paper and the data size used in the ODRL benchmark.

| Shift-Level/data size | 5,000 | 2,000 | 1,000 | 500 |
|---|---|---|---|---|
| gravity-0.5 | **47.18 ± 1.23** | 42.72 ± 2.05 | 39.29 ± 2.52 | 40.65 ± 3.09 |
| gravity-2.0 | **41.46 ± 7.35** | 28.31 ± 5.47 | 28.69 ± 5.95 | 26.78 ± 6.31 |
| friction-0.5 | **69.54 ± 0.48** | 69.62 ± 1.08 | 67.56 ± 1.52 | 64.14 ± 1.90 |
| friction-2.0 | **50.02 ± 3.26** | 46.66 ± 2.92 | 44.26 ± 3.31 | 44.70 ± 3.53 |

Table 9: Hyperparameters of the policy learning. Our method is not very sensitive to the hyperparameters.

| Task | BC-Weight / Uncertainty Penalty | 1 | 5 | 10 |
|---|---|---|---|---|
| Walker2d-Friction-0.5 | 0.05 | 76.96 ± 1.99 | 55.67 ± 21.18 | 51.43 ± 19.37 |
| | 0.1 | 75.64 ± 11.05 | 70.49 ± 2.81 | 62.54 ± 5.49 |
| | 1 | 75.63 ± 2.57 | 82.91 ± 4.43 | 62.50 ± 6.83 |
| Walker2d-kin-footjnt-medium | 0.05 | 67.56±3.05 | 56.88±1.47 | 65.14±2.57 |
| | 0.1 | 62.19±5.27 | 64.17±3.62 | 66.30±0.01 |
| | 1 | 59.33±5.15 | 62.69±5.00 | 62.56±1.09 |
| Walker2d-kin-footjnt-hard | 0.05 | 40.96±1.58 | 57.75±0.34 | 57.59±0.47 |
| | 0.1 | 57.93±0.37 | 56.27±1.12 | 43.13±15.51 |
| | 1 | 34.31±21.12 | 53.92±3.74 | 43.74±14.06 |
| Walker2d-kin-thighjnt-medium | 0.05 | 69.48±4.22 | 60.40±4.27 | 66.85±6.90 |
| | 0.1 | 65.13±3.72 | 65.24±2.10 | 64.19±1.22 |
| | 1 | 64.17±1.83 | 62.10±5.88 | 70.39±0.28 |
| Walker2d-kin-thighjnt-hard | 0.05 | 78.14±2.50 | 59.21±4.79 | 70.20±2.71 |
| | 0.1 | 76.50±1.49 | 61.96±6.71 | 55.95±16.29 |
| | 1 | 69.45±1.78 | 66.92±0.04 | 71.38±5.42 |

different tasks and environments. It is interesting to note that even the suboptimal parameters (0.05) in Table 9 outperform the baseline algorithms.

**Rule of thumb hyperparameters**. We notice that there is no universal set of hyperparameters that works well across all tasks with different environments, shift types, and levels. Even without the dynamics shift, model-based RL methods typically require different hyperparameters for different environments. But empirically, we could have a set of hyperparameters that generally receives a relatively good performance for most tasks, i.e., weight of BC = 0.1 and MOPO penalty = 5. From there, we primarily tune the BC loss weight based on the convergence behavior of the policy. In most cases, using a MOPO penalty of 5 and a BC loss weight selected from the range 0.05, 0.1, 1, 2 yields strong performance. Overall, the number of hyperparameters is modest compared to those commonly required in offline model-based RL methods.

**Computational Resources** We run all experiments on a single GPU (NVIDIA RTX A5000, 24,564 MiB) paired with 8 CPUs (AMD Ryzen Threadripper 3960X, 24-Core). Each experiment requires approximately 12 GB of RAM and 20 GB of available disk space for data storage.

**Computational Cost** We provide an estimated running time of MOPO, DARA, BOSA and our method in Appendix C.5. The running time of MOBODY requires approximately 25% more time to run 1 million steps compared to model-free DARA and is faster than the BOSA. The extra running time is due to the dynamic learning and generation of rollouts. On the other hand, MOPO and MOBODY have similar running times. This demonstrates that we have a similar computational cost and running time compared to the existing model-based method, as the additional loss calculation doesn't significantly increase the computation time.

Table 10: Running time comparison on A5000, AMD Ryzen Threadripper 3960X 24-Core Processor.
.

|  | Walker2d-Gravity-0.5 | HalfCheetah-Gravity-0.5 |
|---|---|---|
| BOSA | ∼3 hours | ∼3.5 hours |
| DARA | ∼2 hours | ∼2.5 hours |
| MOPO | ∼2.5 hours | ∼3 hours |
| MOBODY | ∼2.5 hours | ∼3 hours |

### C.6 ENVIRONMENT SETTING

**Gravity Shift**. Following the ODRL benchmark (Lyu et al., 2024b), we modify the gravity of the environment by editing the gravity attribute. For example, the gravity of the HalfCheetah in the target is modified to 0.5 times the gravity in the source domain with the following code.

```
# gravity
<option gravity="0 0 -4.905" timestep="0.01"/>
```

**Friction Shift** The friction shift is generated by modifying the friction attribute in the geom elements. The frictional components are adjusted to $\{0.1, 0.5, 2.0, 5.0\}$ times the frictional components in the source domain, respectively.

**Kinematic Shift** The kinematics shift is simulated through broken joints by limiting the rotation ranges of some hand joints. We consider the broken ankle joint, hip joint, foot joint, etc, for Mujoco and Adroit environments.

**Morphology Shift** The morphology shift is achieved by modifying the size of specific limbs or torsos of the simulated robot in Mujoco and shrink the finger size in the manipulation task, without altering the state space and action space.

## D LIMITATION AND FUTURE WORK

MOBODY relies on the assumption that the source and target domains share a common state representation $\phi_E$ and transition $\phi_T$ that map the unified latent state action representation to the next state. We believe this assumption is reasonable in our setting: the source and target domains share underlying structure, and MOBODY is designed to exploit this while allowing domain-specific differences via separate action encoders for the two domains. Empirically, we evaluate MOBODY under various types and levels of dynamics shift, and observe that it outperforms or matches recent ODRL baselines across almost all benchmark settings. In particular, in large-shift cases where existing methods fail, MOBODY performs significantly better than the baselines. This suggests that our shared-mapping assumption is not overly restrictive and remains applicable in many off-dynamics RL scenarios.

In extreme cases where the shift is so large that this shared-structure assumption breaks down, the key assumptions of existing baselines (e.g., DARA's low-shift-region assumption) would also fail. In such regimes, more advanced techniques such as stronger domain adaptation or zero-shot transfer may be needed, which we see as an interesting direction for future work.

Also, similar to baselines, our method also struggles with the sparse-reward settings like Antmaze. We believe that tackling sparse-reward off-dynamics RL is an important and challenging future research direction that requires substantially different methods from those in the current literature.

### USAGE OF LLM

All ideas and research are conducted by the author, and the paper itself is written by the author. The LLM is used as a tool for polishing the written content of the paper and checking the grammar errors.

Table 11: Hyperparameter of the MOBODY and baselines.

| Hyperparameter | Value |
| --- | --- |
| **Shared** | |
| Actor network | (256, 256) |
| Critic network | (256, 256) |
| Learning rate | $3 \times 10^{-4}$ |
| Optimizer | Adam |
| Discount factor | 0.99 |
| Replay buffer size | $10^6$ |
| Nonlinearity | ReLU |
| Target update rate | $5 \times 10^{-3}$ |
| Source domain Batch size | 128 |
| Target domain Batch size | 128 |
| **MOBODY** | |
| Latent dimensions | 16 |
| State encoder | (256, 256) |
| State action encoder | (32) |
| Transition | (256, 256) |
| Representation penalty $\lambda_{\text{rep}}$ | 1 |
| Rollout length | 1, 2 or 3 |
| MOPO-Style Reward Penalty $\beta$ | 1,5 or 10 |
| Q-weighted behavior cloning | 0.05, 0.1 or 1 |
| Classifier Network | (256, 256) |
| Reward penalty coefficient $\lambda$ | 0.1 |
| **DARA** | |
| Temperature coefficient | 0.2 |
| Maximum log std | 2 |
| Minimum log std | $-20$ |
| Classifier Network | (256, 256) |
| Reward penalty coefficient $\lambda$ | 0.1 |
| **BOSA** | |
| Temperature coefficient | 0.2 |
| Maximum log std | 2 |
| Minimum log std | $-20$ |
| Policy regularization coefficient $\lambda_{\text{policy}}$ | 0.1 |
| Transition coefficient $\lambda_{\text{transition}}$ | 0.1 |
| Threshold parameter $\epsilon, \epsilon'$ | $\log(0.01)$ |
| Value weight $\omega$ | 0.1 |
| CVAE ensemble size | 1 for the behavior policy, 5 for the dynamics model |
| **IQL** | |
| Temperature coefficient | 0.2 |
| Maximum log std | 2 |
| Minimum log std | $-20$ |
| Inverse temperature parameter $\beta$ | 3.0 |
| Expectile parameter $\tau$ | 0.7 |
| **TD3_BC** | |
| Normalization coefficient $\nu$ | 2.5 |
| BC regularization loss | 0.05, 0.1 or 1 |
| **MOPO** | |
| Transition | (256,256,256) |
| Maximum log std | 2 |
| Minimum log std | $-20$ |
| Reward penalty $\tau$ | 1, 5 or 10 |
| Rollout Length | 1, 2 or 3 |

