# OpenReview forum: "MOBODY: Model-Based Off-Dynamics Offline Reinforcement Learning"
_ICLR.cc/2026/Conference — ICLR 2026 Poster_

### Official Review · Reviewer_MfFn · 2025-10-27

**Soundness:** 3
**Presentation:** 2
**Contribution:** 2
**Rating:** 6
**Confidence:** 3

**Summary:**

**Summary.** This work studies how to tackle the problem of off-dynamics offline RL. More precisely, off-dynamics offline RL aims to learn a policy using offline data from a source domain and a much smaller amount of data from a target domain. Main challenge limits exploration and adaptation to the target's optimal trajectories when the dynamics shift is significant. To alleviate this issue, this paper introduces MOBODY that explicitly learns an unified state representation and separate action encoders for source and target domains. They further introduce a Q-weighted behavior cloning technique to bias the policy toward high-value actions instead of naive action imitation. The method achieves consistent improvements over strong baselines on MuJoCo and Adroit benchmarks with different types and severities of dynamics shift.

---
**Review summary.** Overall, the reviewer thinks that the paper is clearly written, well-motivated, and timely, especially in the context of sim-to-real transfer. The idea itself is conceptually simple, intuitively reasonable, and appears to be practical to implement. Although the work is not theoretically deep, it is coherent and sufficiently justified as a framework-oriented contribution. However, the experiments section leaves several open questions and limitations that need further clarification. Therefore, the reviewer assigns an initial score of 6 and plan to revisit this rating after the authors address the concerns and questions raised in this review.

**Strengths:**

**Writing**
- The paper is very readable and logical from motivation to formulation to algorithm and experiments.
- Figures and tables are generally informative; the method and notation are consistent.

---
**Methodology**
- The central idea, learning shared latent state and transition functions with domain-specific action encoders, is simple and well-motivated.
- The architecture (shared $\phi_E$, $\phi_T$; separate $\psi_{\text{src}}$, $\psi_{\text{trg}}$) directly targets the off-dynamics challenge, that is, achieving the same actions but different dynamic transtion.
- The introduction of a Q-weighted BC is reasonable, encouraging the policy to favor actions with high target-domain value while mitigating overfitting to source data.

---
**Experiments**
- The experiments are extensive, covering MuJoCo with diverse transition (e.g., gravity/friction/kinematic/morphology shifts) and Adroit tasks with multiple difficulty levels.
- Ablation studies demonstrate the impact of both the cycle transition loss and Q-weighting.
- Computational cost, hyperparameter sensitivity, and reproducibility are reported in detail.

**Weaknesses:**

**Writing**
- Figures 3–4 summarize results but are under-analyzed. differences and error ranges are not interpreted in the text. Please discuss more why and how come.
- Algorithm 1 and 2 include many moving parts, such as DARA regularization and reward modeling, that are not clearly integrated into the paper.

**Methodology**
- It is not fully clear how the encoder losses (Eq. 4) interact within the overall objective (Eq. 6). Are $L^{\text{src}}{\text{rep}}$ and $L^{\text{trg}}{\text{rep}}$ averaged or weighted differently?
- The paper should clarify whether $p(s'|s,a)$ is assumed deterministic. This strongly affects the validity of the VAE-style cycle transition loss and the interpretation of uncertainty.
- The cycle transition loss (Eq. 5) is creative, but its formal grounding is weak: the gradient-stopping mechanism and KL structure are under-explained, and it is unclear how it avoids mode collapse or degenerate representations.
- The target-domain encoder $\psi_{\text{trg}}$ is trained with very limited data. Overfitting and instability risks should be analyzed or mitigated via regularization or weight sharing.
- Target Q-weighted BC is empirically validated but lacks theoretical analysis; no formal bound or convergence argument is provided, and the technique is a relatively mild variation of AWR.

**Experiments**
- The reviewer concerns baseline selection and fairness.
    - Most baselines (IQL, TD3-BC, DARA, BOSA, MOPO) are strong but somewhat outdated.
    - More recent or structurally related methods (e.g., BPRL, LEO, Koopman Q-learning, or model-based domain adaptation approaches) are missing.
- In several tasks, simpler methods, IQL, outperform off-dynamics-specific methods, i.e., DARA, BOSA. This discrepancy deserves explicit discussion.
- Only two dynamics domains are considered. It is unclear how MOBODY generalizes to multiple or continuous domains.
- High variance in results, for example, Walker2d-Gravity 5.0: $46.05 \pm 20.73$, suggests instability that is not analyzed.
- There is no qualitative evidence supports cross-domain transfer.

**Limitations**
- MOBODY assumes that a shared transition mapping ($\phi_E$, $\phi_T$) exists across domains. This is a strong assumption; when dynamics are structurally divergent, the approach may fail.
- The approach relies on uncertainty regularization from model ensembles, but this is treated superficially.
- There is no analysis of behavior under extreme dynamics mismatch, where shared representations might be misleading.
- Need to more discuss and survey about prior connections.

**Miscellaneous**
- Eq. (3): inconsistent summation notation and formatting.
- Eq. (6): unclear aggregation of $L_{\text{rep}}$ terms.
- Line 165: justification for adding $z_s$ to $\psi(z_s,a)$ is weak (“for simplicity” is insufficient).
- Line 126–127: references needed for conservative reward regularization (e.g., MOPO, COMBO).
- In several places, “off-dynamics RL” should be replaced with “off-dynamics offline RL” for accuracy.

**References**
- M. Weissenbacher et al. Koopman Q-learning: Offline Reinforcement Learning via Symmetries of Dynamics. ICML 2022.
- J. Eo, et al. The impact of dataset on offline reinforcement learning performance in UAV-based emergency network recovery tasks. IEEE Comm. Let. 2023.
- M. Uehara, et al. Pessimistic model-based offline reinforcement learning under partial coverage. ICLR 2022.
- B. Hao, et al. Bootstrapping fitted q-evaluation for off-policy inference. ICML 2021.
- G. An, et al. Uncertainty-based offline reinforcement learning with diversified q-ensemble. NeurIPS 2021.
- D. Lee, et al. Temporal Distance-aware Transition Augmentation for Offline Model-based Reinforcement Learning. ICML 2025.

**Questions:**

- In Table 1, IQL often outperforms specialized methods such as DARA and BOSA. Could the authors explain why this occurs?
    - Have the authors tried using AWR style policy extraction for DARA or BOSA to see whether the observed gap stems from the extraction mechanism rather than the dynamics modeling?
- For Ablation 1, would it be possible to combine alternative dynamics models, like COMBO or RAMBO, with the proposed Q-weighted BC to isolate the contribution of the policy regularizer?
- For Ablation 2, could the authors replace the Q-weighted cloning loss with an AWR to directly compare different policy extraction approaches?
- BOSA appears conceptually compatible with MOBODY’s dynamics learning. Have the authors tested whether integrating the two (BOSA + MOBODY dynamics) yields complementary effects or insights into the efficacy of Q-weighted behavioral cloning?
- Given the extremely small target dataset, have the authors observed any instability or collapse in the target encoder $\psi_{\text{trg}}$?
- How does MOBODY perform when the amount of target-domain data decreases further, e.g., below 1 K samples?
- How many ensemble models are used for uncertainty quantification?
    - Is the performance sensitive to ensemble size or the uncertainty penalty $\beta$?
- The reviewer wonders what the aurhos predict will happen if we consider a representation based on the TD information of each domain.
    - A. Zhang, et al. Learning Invariant Representations for Reinforcement Learning without Reconstruction. ICLR 2021.
    - C. Zheng, et al. Contrastive Difference Predictive Coding. ICLR 2024.
    - D. Lee, et al. Temporal Distance-aware Transition Augmentation for Offline Model-based Reinforcement Learning. ICML 2025.

---

> ### Author Response · Authors · 2025-11-21
>
> We thank the reviewer for the constructive review to help us improve the paper. We now address each concern of the reviewer.
>
> ---
> **Q1** Figures 3–4 summarize results but are under-analyzed.
>
> **A1:** TLDR: We further discuss the results in Figures 3–4.
>
> Figure 3 summarizes the results on the MuJoCo kinematic shift, morphology shift, and Adroit manipulation tasks. We report the performance by summing the returns across all tasks for each method at each shift level. The per-task results are provided in Tables 4 and 5. In Figures 3 and 4, we do not include error bars because the results are aggregated across many tasks, and a single standard deviation is not well-defined at this level of aggregation.
>
> Figure 3 shows that our method outperforms the baselines across different shift levels and achieves the best performance on 32 out of 40 tasks.
>
> Figure 4 further reports the aggregated returns of all methods under different gravity/friction shift levels. We observe that in the larger shifts (5.0 and 0.1) setting, our method achieves larger improvements over the baselines than at moderate shifts (0.5 and 2.0), demonstrating its effectiveness in the large-shift regime. We attribute this to the improved exploration enabled by the learned target dynamics.
>
> ---
> **Q2** DARA regularization and reward modeling are not integrated in the main text.
>
> **A2:** Due to the page limit, and these two modules are not the main contribution of our paper, we put the detailed discussion in Appendix B. In the revision, we will briefly summarize them in the main text to make the overall presentation easier to follow.
>
> ---
> **Q3** It is not fully clear how the encoder losses interact within the overall objective. Are $L^{trg}_{rep}$ and $L^{src}_{rep}$ averaged or weighted differently?
>
> **A3:** We directly sum $L_{rep}^{trg}$ and $L_{rep}^{src}$. In our experiments, we sum all the losses together, and the weights are all set to be 1 as stated in Appendix C. We will make it clear in the revision.
>
> ---
> **Q4** Whether the transition is assumed deterministic.
>
> **A4:** TLDR: We do not assume the transition to be deterministic.
>
> In our experiments, the underlying environments have deterministic transitions, but our method does not rely on a deterministic assumption. The transition model and VAE-style cycle transition loss are defined in a probabilistic framework: even if the true dynamics are deterministic, we can still treat $p(s' \mid s,a)$ as a distribution. In this case, the VAE-style loss and uncertainty estimates remain valid, as they operate on the learned distribution rather than on a single next state. Similar Gaussian transition models and uncertainty estimates have been widely used in prior model-based offline RL work, such as MOPO, COMBO, and RAMBO, where probabilistic dynamics are explicitly modeled.
>
>
> ---
> **Q5** The gradient-stopping mechanism and KL structure are under-explained, and it is unclear how they avoid mode collapse or degenerate representations.
>
> **A5:** The gradient stopping is not introduced to avoid mode collapse or degenerate representations; rather, it is used to prevent gradients from flowing through the target in the loss, and this is a widely adopted technique in prior work [1,2]. In addition, the KL regularization term comes from the VAE formulation, which is known to better mitigate mode collapse than standard autoencoders or purely deterministic representation learning methods.
>
>
>
> ---
> **Q6** The target-domain encoder is trained with very limited data. Overfitting and instability risks should be analyzed or mitigated via regularization or weight sharing.
>
> **A6:** We empirically observe overfitting when the VAE-style cycle transition loss is removed: the learned state–action representations could overfit the training data but generalize poorly. To mitigate this, we introduce a VAE-style cycle transition loss, whose KL term regularizes the latent representation toward a normal distribution, a standard property of VAEs. In addition, we employ weight sharing between the transition model and the encoder, which are jointly trained on both source and target data to further reduce overfitting. By contrast, the action encoders are domain-specific and are trained separately on source and target data, respectively. With the separate action encoder and shared module, we can mitigate the overfitting issue induced by the small target data.

---

> ### Author Response · Authors · 2025-11-21
>
> **Q7** Target Q-weighted BC is empirically validated but lacks theoretical analysis.
>
>
> **A7:** TLDR: The target-Q weighted BC loss is inspired by the advantage-weighted regression (AWR) and reward-weighted regression. ALso, the weighted BC loss is to guide the policy toward actions that are expected to perform better under the target dynamics, instead of uniformly weighting the action in the source.
>
> We first clarify that our main contribution regarding the target-Q-weighted BC loss is not the specific functional form in Eq. (8). Instead, our motivation has two key aspects:
> (1) introducing a BC loss specifically to mitigate out-of-distribution issues in offline off-dynamics RL, and
> (2) using target-Q weighting to explicitly push the policy toward actions with high Q-values under the target dynamics. The first motivation (using BC to improve robustness) is intuitive and follows prior work, while we view the second, leveraging target-Q information to reweight the BC loss, as the key conceptual contribution and motivation in the off-dynamics RL problem rather than the exact formula.
>
> More concretely, in off-dynamics RL, naively cloning source-domain actions can harm performance: actions that perform well in the source domain may perform poorly in the target domain due to dynamics shift, so a vanilla TD3-BC–style loss is insufficient. Our target-Q-weighted BC loss is therefore designed to guide the policy toward actions that are expected to perform better under the target dynamics, directly mitigating this mismatch, and the target Q-value is learned from reward-augmented data.
>
> Also, the target Q-weighted BC loss is inspired by the advantage-weighted regression (AWR) in IQL and the reward-weighted regression (RWR) [7]. While the AWR is built on the RWR and we follow the idea of RWR by replacing the Monte-Carlo estimation on the reward with the target Q value.
>
>
>
> ---
> **Q8** The baseline selection fairness.
>
> **A8:** In our paper, we primarily compare against the baselines introduced in the ODRL benchmark [4] (published in 2025), including the most recent offline off-dynamics RL methods DARA [5] and BOSA [6]. In addition, classic methods such as IQL and TD3-BC are widely used as baselines in previous off-dynamics offline RL works, so we also include them for completeness.
>
> We appreciate the additional references suggested by the reviewer; however, these methods are designed for the standard offline RL setting rather than the off-dynamics RL setting, and thus are not directly comparable under our evaluation protocol. We will add a discussion of these offline RL methods in the related work section of the revised version.
>
>
> ---
> **Q9** Why IQL outperforms off-dynamics-specific methods, i.e., DARA, BOSA.
>
> **A9:** In some settings, the DARA reward augmentation term, derived from a KL divergence between source and target dynamics, can become ill-defined when the two dynamics have little or disjoint support. This can destabilize training and cause DARA to perform worse than IQL. For BOSA, the target dynamics model trained purely on target data could also be problematic in our setting, where only about 5,000 target transitions are available, making it difficult to accurately learn the target dynamics and potentially hurting performance.
>
>
>
> ---
> **Q10** There is no qualitative evidence that supports cross-domain transfer.
>
> **A10:** We empirically validate that our method can support cross-domain transfer. We would be grateful if the reviewer could clarify what specific qualitative evidence they have in mind.
>
> ---
> **Q11** The model ensemble uncertainty quantification is treated superficially.
>
> **A11:**
> Our uncertainty quantification (UQ) follows the standard model-ensemble approach widely adopted in prior model-based RL works such as MOPO, RAMBO, and COMBO. We do not claim this component as a novel contribution. Concretely, we train an ensemble of 7 dynamics models and, for each transition, compute the standard deviation of the predicted next state across the ensemble. We then take the maximum standard deviation across state dimensions as a scalar UQ measure for that transition. Note that open-source implementations typically use 5–10 ensemble members as a standard hyperparameter choice. The full procedure and implementation details are provided in Appendix B.

---

> ### Author Response · Authors · 2025-11-21
>
> **Q12** MOBODY assumes that a shared transition mapping $(\phi_E, \phi_T )$ exists across domains, which could fail when assumption fail.
>
> **A12:** TLDR: we clarify that the shared-structure assumption is reasonable in our setting, and show that MOBODY can be applied to various off-dynamics RL problems, outperforming baselines, especially for large dynamics shifts.
>
> We believe this assumption is reasonable in our setting: the source and target domains share underlying structural knowledge, and MOBODY is explicitly designed to exploit this shared structure. At the same time, we allow for domain-specific differences by using separate action encoders for the two domains, enabling the model to capture distinct properties of the source and target dynamics.
>
> Empirically, we evaluate MOBODY under various types and levels of dynamics shift, and observe that it outperforms or matches recent ODRL baselines across almost all benchmark settings. In particular, in large-shift cases where existing methods fail, MOBODY performs significantly better than the baselines. This suggests that our shared-mapping assumption is not overly restrictive and remains applicable in many off-dynamics RL scenarios.
>
> In extreme cases where the shift is so large that this shared-structure assumption fails, we argue that the core assumptions of existing baselines (e.g., DARA’s assumption that the optimal policy lies in a low dynamics shift region) would also fail. In such settings, more sophisticated approaches, such as stronger domain adaptation techniques or zero-shot transfer methods, may be required, which we view as an interesting direction for future work.
>
> ---
> **Q13**  No analysis of behavior under extreme dynamics mismatch.
>
> **A13:**
> We also evaluate our method under extreme dynamics mismatch, such as gravity/friction factors of 0.1 and 5.0, the largest shift the existing benchmark has, where baseline methods almost always fail. In many of these settings, MOBODY achieves substantial performance gains over the baselines, demonstrating that our approach remains effective and applicable even under large dynamics shifts.
>
>
>
> ---
> **Q14** Need to discuss and survey more about prior connections.
>
> **A14:** We include a discussion of both recent and classic model-based RL and off-dynamics RL in the appendix. We will add further discussion of related work, as well as how they are connected with our method on offline RL, model-based RL, and representation learning, in the revision.
>
>
> ---
> **Q15**  Justification for adding $z_s$ to $\psi(z_s, a)$ .
>
> **A15:** In principle, the aggregation of the state representation and the state–action representation could be implemented via an additional aggregation function $f$, i.e., $f(z_s, \psi(z_s,a))$. However, this would further increase model complexity. Inspired by common model-based RL implementations that parameterize transitions as $s' = s + f(s,a)$, where the next state is represented as the current state plus a residual, we instead adopt a similar additive form in the latent space: $z_s + \psi(z_s, a)$. Our empirical results support this design choice.
>
>
> ---
> **Q16** Have the authors tried using AWR style policy extraction for DARA or BOSA to see whether the observed gap stems from the extraction mechanism rather than the dynamics modeling?
>
> **A16:** Our implementation of DARA already adopts the AWR-style policy learning used in IQL, with DARA’s reward regularization applied on top. In our experiments, this AWR-style policy extraction alone does not close the performance gap to MOBODY; we do not observe it reaching MOBODY’s performance without explicit dynamics modeling.
>
> For BOSA, we follow the algorithm presented in the paper, using support policy optimization to regularize the policy instead of any BC policy regularization.
>
> ---
> **Q17** What is the performance of COMBO, with the proposed Q-weighted BC to isolate the contribution of the policy regularizer?
>
> **A17:** TLDR: we have such an ablation in A2, using COMBO in the model-based RL training.
>
> We clarify that COMBO does not introduce a stronger dynamics model. Instead, it builds on the CQL idea to conservatively learn the value function for OOD state–action pairs. In our paper, we include an ablation A2 in Section 4.3, where we replace our policy learning component with CQL (the main idea behind COMBO). The resulting performance is worse than that of our full method.
>
> CQL is specifically designed to conservatively estimate the value of OOD state–action pairs. However, in the off-dynamics RL setting, we argue that the OOD issue is even more severe because (1) the amount of target data is limited, and (2) actions that are good in the source domain may perform poorly in the target domain. Thus, stronger regularization on the policy, such as behavior cloning or our Q-weighted BC loss, is preferable and empirically more effective than purely value-based conservatism as in COMBO/CQL.

---

> ### Author Response · Authors · 2025-11-21
>
> **Q18** Replace the Target-Q weighted BC loss with the AWR-style policy regularization.
>
> **A18:** TLDR: We provide additional results showing that target-Q-weighted behavior cloning (BC) loss improves AWR-style policy regularization when used as a regularizer in off-dynamics RL.
>
> We evaluate a standard IQL-style weighting scheme, which is $\exp(\beta (Q(s,a) - V(s)))$. We report the results in the table below. We see that our Q-weighted BC loss yields slightly better performance. A possible reason is that training an additional value network from model-based rollouts introduces extra complexity and noise, which can hurt performance.
>
> #### Table: Comparison of using IQL weight for BC loss and target Q weighted BC loss.
> | Env| Shift| IQL weight for BC| MOBODY|
> |------------|--------------|-------------------|-------------------|
> | Halfcheetah| gravity-0.5| 45.13 ± 2.21| **47.18 ± 1.23**|
> | Halfcheetah| gravity-2.0| 40.42 ± 3.63| **41.46 ± 7.35**|
> | Halfcheetah| friction-0.5| 64.82 ± 1.53| **69.54 ± 0.48**|
> | Halfcheetah| friction-2.0| **51.30 ± 2.53**| 50.02 ± 3.26|
> | Walker2d| gravity-0.5| **48.45 ± 2.33**| 43.57 ± 2.32|
> | Walker2d| gravity-2.0| 39.67 ± 3.21| **44.32 ± 4.58**|
> | Walker2d| friction-0.5| 63.47 ± 5.01| **76.96 ± 1.99**|
> | Walker2d| friction-2.0| 71.39 ± 1.81| **73.74 ± 0.49**|
>
> ---
> **Q19** What is the performance of integrating the two (BOSA + MOBODY dynamics), as they are conceptually compatible, and what is the efficacy of Q-weighted behavioral cloning?
>
> **A19:** In the following table, we provide additional results where we implement BOSA using MOBODY’s dynamics. Plugging MOBODY’s dynamics into BOSA improves performance over vanilla BOSA in 6 out of 8 tasks, demonstrating the effectiveness of our dynamics-learning component. However, the performance is not as good as MOBODY itself, further highlighting the benefit of our target-Q-weighted behavior cloning compared to the support policy optimization used in BOSA.
>
> #### Table: Performance of implementing the BOSA with the MOBODY’s dynamics.
> | Env| Shift| BOSA| MOBODY's dynamics + BOSA| MOBODY|
> |-----------|--------------|-------------------|--------------------|-------------------|
> | HalfCheetah | gravity-0.5  | 43.96 ± 5.68| 45.07 ± 7.49| **47.18 ± 1.23**|
> | HalfCheetah | gravity-2.0  | 27.86 ± 0.54| 32.99 ± 1.29| **41.60 ± 7.35**|
> | HalfCheetah | friction-0.5 | 68.93 ± 0.35| 69.11 ± 0.28| **69.54 ± 0.48**|
> | HalfCheetah | friction-2.0 | 46.53 ± 0.37| 45.14 ± 0.91| **50.02 ± 3.26**|
> | Walker2d| gravity-0.5  | 40.09 ± 20.37| 39.81 ± 11.01| **43.57 ± 2.32**|
> | Walker2d| gravity-2.0  | 8.91 ± 2.28| 11.75 ± 4.01| **44.32 ± 4.58**|
> | Walker2d| friction-0.5 | 63.94 ± 20.40| 71.32 ± 8.41| **76.96 ± 1.99**|
> | Walker2d| friction-2.0 | 39.06 ± 17.53| 44.97 ± 12.86| **73.74 ± 0.49**|
>
> ---
> **Q20** How does MOBODY perform when the amount of target-domain data decreases further?
>
> **A20:** TLDR: 5k target transitions are a very small amount of target data, as also noted in prior work [1,2,3]. We additionally conduct ablations with fewer target transitions. Moreover, our dynamics are deterministic, so the induced off-dynamics shifts are inherently non-smooth or even discontinuous.
>
> We first clarify that, following the ODRL benchmark [1], 5,000 transitions already constitute a very small amount of target data. Prior online and offline off-dynamics RL work typically assumes substantially more target data, often using a ratio such as 1/10 of the source transitions [2,3] (e.g., 100,000 target transitions for 1 million source transitions or training steps). In the table below, we further report results with 500, 1k, 2k, and 5k target transitions. MOBODY’s performance degrades as the target dataset shrinks, but the drop is not dramatic, indicating that our method remains effective even with very limited target data.
>
> Furthermore, the off-dynamics shifts from friction/gravity or kinematics/morphology changes in our experiments are non-smooth and contain discontinuous changes in the dynamics. With deterministic transitions, any dynamics mismatch produces different next states, and larger shifts lead to larger differences between next states, making it more non-smooth and discontinuous. Empirically, MOBODY remains effective even under extreme mismatch (e.g., 0.1/5.0 shift levels), as shown in Table 1.
>
> #### Table: Performance of MOBODY under different target transition sizes.
> | Env| Shift| 5k (original)| 2k| 1k| 500|
> |------------|--------------|------------------|------------------|------------------|------------------|
> | Halfcheetah| gravity-0.5| **47.18 ± 1.23**| 42.72 ± 2.05| 39.29 ± 2.52| 40.65 ± 3.09|
> | Halfcheetah| gravity-2.0| **41.46 ± 7.35**| 28.31 ± 5.47| 28.69 ± 5.95| 26.78 ± 6.31|
> | Halfcheetah| friction-0.5 | **69.54 ± 0.48**| 69.62 ± 1.08| 67.56 ± 1.52| 64.14 ± 1.90|
> | Halfcheetah| friction-2.0 | **50.02 ± 3.26**| 46.66 ± 2.92| 44.26 ± 3.31| 44.70 ± 3.53|

---

> ### Author Response · Authors · 2025-11-21
>
> **Q21** How many ensemble models are used for uncertainty quantification, and is the performance sensitive to ensemble size or the uncertainty penalty?
>
> **A21:**
> In our experiments, we use seven ensemble models, following the open-source implementation OfflineRL-Kit [8]. We also evaluate our algorithm with different ensemble sizes. For ensembles of 4 and 10 models, we observe that increasing the ensemble size generally reduces the variance while keeping the mean performance similar, indicating that larger ensembles lead to more stable policy learning due to improved uncertainty quantification.
>
> | Env        | Shift        | Ensemble 4        | Ensemble 7        | Ensemble 10       |
> |-----------|--------------|-------------------|-------------------|-------------------|
> | HalfCheetah | gravity-0.5  | 48.34 ± 5.49      | 47.18 ± 1.23      | 48.11 ± 1.08      |
> | HalfCheetah | gravity-2.0  | 42.83 ± 7.13      | 41.60 ± 7.35      | 39.78 ± 6.41      |
> | HalfCheetah | friction-0.5 | 70.98 ± 3.24      | 69.54 ± 0.48      | 68.12 ± 0.44      |
> | HalfCheetah | friction-2.0 | 51.14 ± 5.41      | 50.02 ± 3.26      | 50.46 ± 1.84      |
>
>
> ---
> **Q22** What will happen if we consider a representation based on the TD information of each domain?
>
> **A22:**
> TLDR: We discuss below how these related works connect to our method and how they may inspire future directions.
>
> We thank the reviewer for pointing out these references, which provide valuable perspectives for future work.
> For the first paper [9], we believe it is less directly related to our setting, as it focuses on learning invariant representations for image observations without explicitly modeling transition dynamics, whereas our method requires learning the dynamics given a shared state representation.
>
> The second paper [10], on the other hand, studies goal-conditioned RL and offers interesting insights that could potentially benefit off-dynamics RL. As reported in the ODRL benchmark [4], existing off-dynamics RL methods fail on AntMaze, a goal-conditioned off-dynamics RL task where the map layouts differ across domains. The contrastive representation learning of goals and states in [10] could be adapted to off-dynamics RL by aligning state representations with goal representations, which may improve performance on AntMaze and other goal-conditioned off-dynamics RL tasks.
>
> For the third paper [11], the representation learning component itself is not new, which uses an autoencoder. However, the representation is further learned and constrained by trajectory-level and transition-level temporal distance losses. It is unclear whether such temporal distance information in the representation would directly help dynamics learning, since their main goal is to stitch states and avoid implausible trajectories. While this may not substantially improve model rollouts, it could be beneficial for methods that perform data augmentation via trajectory generation in off-dynamics RL.
>
> Overall, we appreciate the reviewer bringing up these interesting works. Although they are not directly aligned with our dynamics-learning objective, we agree that representation learning constrained by additional side information could be useful for generating synthetic yet realistic target data in off-dynamics data augmentation algorithms, and we see this as a promising direction for future work.
>
>
>
> ---
> References:
>
> [1] Can Increasing Input Dimensionality Improve Deep Reinforcement Learning?
>
> [2] For sale: State-action representation learning for deep reinforcement learning.
>
> [3] Cross-Domain Policy Adaptation by Capturing Representation Mismatch.
>
> [4] ODRL: A Benchmark for Off-Dynamics Reinforcement Learning
>
> [5] ​​DARA: Dynamics-Aware Reward Augmentation in Offline Reinforcement Learning.
>
> [6] Beyond OOD State Actions: Supported Cross-Domain Offline Reinforcement Learning
>
> [7] Relative entropy policy search.
>
> [8] OfflineRL-Kit: https://github.com/yihaosun1124/OfflineRL-Kit
>
> [9] Learning Invariant Representations for Reinforcement Learning without Reconstruction. ICLR 2021.
>
> [10] Contrastive Difference Predictive Coding. ICLR 2024.
>
> [11] Temporal Distance-aware Transition Augmentation for Offline Model-based Reinforcement Learning. ICML 2025.

---

> > ### Comment · Reviewer_MfFn · 2025-11-26
> >
> > Thank you for detailed response. The reviewer cannot consider raising the score until the main claim of the paper is stated in a technically accurate and satisfactory manner. As the current phrasing remains unclear and potentially misleading, the reviewer may need to reassess the score unless this point is resolved.
> >
> > ---
> > `Related to Q4` While it is possible to use a probabilistic transition model, this does not automatically imply that the method truly adapts to stochastic environments. In practice, next-state prediction typically relies on the mean or mode of the learned distribution, which does not capture multi-modal or genuinely stochastic transitions. Handling stochasticity is therefore a separate and substantially different challenge.
> >
> > The reviewer would appreciate it if the authors could more clearly articulate the precise scope of the problem addressed in this work, without overstating the generality of the setting.
> >
> > ---
> > `Related to Q5` Actual question is how the cycle transition loss itself avoids collapsing to trivial representations. Sorry for the confusion.
> >
> > ---
> > `Related to Q6` The response relies on generic statements (e.g., KL regularizes, weight sharing helps) without explaining how $\psi_{trg}$ avoids collapse under extremely small target data. The reviewer think that the answer does not resolve the overfitting/instability risk the question was asking about due to no analysis, no mechanism, and no empirical evidence.
> >
> > ---
> > `Related to Q8` The authors already use IQL and similar methods as baselines, yet they provide no meaningful discussion or comparison regarding the additional relevant approaches raised by the reviewer.
> >
> > ---
> > `Related to Q9` The explanation for why IQL performs well in off-dynamics settings lacks any structural insight or supporting evidence, making the argument unconvincing.
> >
> > ---
> > `Related Q 12 and 13` The core concern remains unresolved. The authors provide no substantive analysis of when the shared-transition assumption breaks or how the method behaves under extreme or structurally divergent dynamics, leaving the method’s robustness and failure modes insufficiently addressed.
> >
> > ---
> > `Related to Q16` The reviewer cannot distinguish whether the advantage MOBODY has over other algorithms comes from AWR or from other configurations.
> >
> > ---
> > `Related to Q18 and 19` Thank you for the additional experiments. From a reviewer’s perspective, the central claimed contribution of MOBODY is its dynamics modeling. However, the new results suggest that the impact of the proposed dynamics module is in fact limited (Q19). Moreover, AWR/IQL-style policy extraction appears to provide performance gains that are at least comparable, and in some cases stronger, than the proposed Q-weighted BC formulation (Q18 vs. Q19 tables). This raises concerns that the core contribution of the paper may not be the main driver of the reported improvements, instead, auxiliary design choices seem to play a disproportionately large role.

---

> ### Author Response · Authors · 2025-12-02
> **Further Response**
>
> **Q1:** Using a probabilistic transition model doesn’t automatically mean the method truly handles stochastic or multi-modal environments, since predictions often just use the mean or mode. They ask the authors to clearly state the actual scope of their setting and avoid overstating its generality.
>
> **A1:** In our experiments, the environments are deterministic. In our response to Q4, our intention was to clarify that the VAE-style mapping and uncertainty quantification remain meaningful and valid even in deterministic settings. In general, our method is mainly designed for deterministic environments. We will explicitly state it in the paper that the experimental environments are deterministic.
>
> ---
> **Q2:** How does the cycle transition loss itself avoid collapsing to trivial representations? And how $\psi_\text{trg}$ avoids collapse under extremely small target data
>
> **A2:** First, the cycle-transition objective is a VAE-style loss, which helps mitigate degenerate representation without VAE or deterministic autoencoder representation.
>
> Regarding $\psi_{\text{trg}}$, we would like to clarify that we do not rely on $\psi_{\text{trg}}$ to prevent mode collapse. At a high level, $\psi_{\text{src}}$ and $\psi_{\text{trg}}$ are introduced to allow the model to capture domain-specific information from the source and target environments, respectively. In practice, these modules are very small neural networks (a single hidden layer with 32 units). On the other hand, the shared encoders $\phi_E$ and $\phi_T$ are trained jointly on both source and target data using the VAE-style cycle-transition loss, capturing shared state representation and transition of the two domains, which will help mitigate the overfitting, and we believe that this is the main reason that we can mitigate the overfitting issue.
>
>
> Empirically, we assess the generalization of the learned dynamics after 1M training steps. We compare MOBODY’s dynamics model with several alternative dynamics-learning schemes, while all policies are trained using MOBODY’s policy optimization. All variants share MOBODY’s architecture but use a single action encoder.
>
> (1) Compared to training dynamics on target-only data (where encoder, decoder, and action encoder are all fit using target data alone), MOBODY attains lower MSE, indicating reduced overfitting and better generalization to the target environment, thanks to joint training of the encoder/decoder on both source and target data.
>
> (2) Simply merging source and target data into one training set without a shared-structure design also performs worse: this “mixed” model tends to approximate dynamics in between the source and target domains rather than the true target dynamics, leading to higher target MSE and degraded downstream policy performance.
>
>
>
>
> | Metric            | Task                  | Trained only on target data | Combined data        | MOBODY                |
> |-------------------|-----------------------|-----------------------------|----------------------|-----------------------|
> | MSE               | Walker2d-friction-0.5 | 2.23 ± 0.26                 | 1.96 ± 0.68          | **1.25 ± 0.39**       |
> | MSE               | Walker2d-gravity-0.5  | 2.11 ± 0.48                 | 1.87 ± 0.32          | **1.93 ± 0.34**       |
> | MSE               | Ant-friction-0.5      | 2.99 ± 0.51                 | 2.01 ± 0.24          | **1.88 ± 0.18**       |
> | MSE               | Ant-gravity-0.5       | 1.57 ± 0.39                 | 1.53 ± 0.39          | **1.46 ± 0.26**       |
> | Normalized Score  | Walker2d-friction-0.5 | 56.31 ± 7.17                | 41.38 ± 5.12         | **76.96 ± 1.99**      |
> | Normalized Score  | Walker2d-gravity-0.5  | 39.71 ± 3.29                | 42.13 ± 3.98         | **43.57 ± 2.32**      |
> | Normalized Score  | Ant-friction-0.5      | 48.13 ± 4.43                | 46.23 ± 6.85         | **62.41 ± 4.10**      |
> | Normalized Score  | Ant-gravity-0.5       | 28.32 ± 3.87                | 31.39 ± 3.80         | **37.44 ± 2.79**      |

---

> ### Author Response · Authors · 2025-12-02
> **Further Response 2**
>
> **Q3:** The authors already use IQL and similar methods as baselines, yet they provide no meaningful discussion or comparison regarding the additional relevant approaches raised by the reviewer.
>
> **A3:** We further provide discussion and analysis of the additional relevant approaches raised by the reviewer. We implement the Koopman Q-learning and show the results in the following table. We see that MOBODY outperforms the Koopman Q-learning. Though the Koopman Q-learning learns the symmetries of dynamics for data augmentation, it doesn't capture the difference in dynamics, and the learned dynamics are not the target dynamics compared with our method.
>
> Regarding model-based domain adaptation, to the best of our knowledge, there is no prior work specifically tailored to off-dynamics offline RL. The most related work we are aware of is [3], but its setting differs substantially from ours: they consider online adaptation with access to a source expert policy and interaction with the target domain, whereas we operate in a purely offline setting with no interaction in the target environment. Thus, it will be an inappropriate comparison.
>
> For LEO, we are not certain which paper the reviewer has in mind and would appreciate it if the reviewer could point it out. For BPRL, we suspect the reviewer may be referring to [4], an active ICLR 2026 submission that uses behavior programming for state representation learning. However, their experiments are conducted on MiniGrid with discrete state and action spaces, so we believe their method may not directly apply to our continuous-control, off-dynamics offline RL setting. In addition, we believe this method could provide interesting insight into how to design the latent representation and how to use the side information to constrain the latent representation in off-dynamics RL.
>
>
> #### Table: comparison of MOBODY with Koopman Q learning. Koopman Q learning receives a similar reward as IQL and DARA. MOBODY outperforms them.
> | Env| Koopman Q learning| IQL| DARA| MOBODY|
> |--------------------------|-------------------------|--------------------|--------------------|--------------------|
> | HalfCheetah-gravity-0.5  | 46.89 ± 4.31| 44.23 ± 2.93| 46.11 ± 1.93| 47.18 ± 1.23|
> | HalfCheetah-gravity-2.0  | 30.63 ± 3.62| 31.34 ± 1.68| 31.85 ± 1.31| 41.60 ± 7.35|
> | HalfCheetah-friction-0.5 | 68.06 ± 2.02| 69.80 ± 0.64| 64.89 ± 3.04| 69.54 ± 0.48|
> | HalfCheetah-friction-2.0 | 42.62 ± 1.39| 46.04 ± 2.04| 46.25 ± 2.36| 50.02 ± 3.26|
> | Walker2d-gravity-0.5| 39.52 ± 6.33| 42.05 ± 10.52| 29.72 ± 16.02| 43.57 ± 2.32|
> | Walker2d-gravity-2.0| 29.22 ± 4.19| 25.69 ± 10.70| 32.20 ± 1.05| 44.32 ± 4.58|
> | Walker2d-friction-0.5| 72.65 ± 0.97| 66.26 ± 3.03| 68.81 ± 1.12| 76.96 ± 1.99|
> | Walker2d-friction-2.0| 68.44 ± 1.42| 65.40 ± 7.13| 72.91 ± 0.37| 73.74 ± 0.49|
> | Total| 396.04| 390.81| 392.74| 446.93|
> ---
>
> **Q4:** The explanation for why IQL performs well in off-dynamics settings lacks any structural insight or supporting evidence, making the argument unconvincing.
>
> **A4:**  Similar phenomena have also been observed in [1] and [2], where IQL/BC-SAC outperforms recent off-dynamics RL methods in the “source offline + target online” setting. As mentioned in A9 and also in [2], the dynamics gap term $p_{\text{trg}}(s'\mid s,a)/p_{\text{src}}(s'\mid s,a)$ of DARA, defined via a KL-based regularizer, can be ill-defined for baseline methods when the two dynamics have disjoint or nearly disjoint support. For example, if $p_{\text{src}}(s'\mid s,a)$ is extremely small, then the ratio $p_{\text{trg}}(s'\mid s,a)/p_{\text{src}}(s'\mid s,a)$ becomes extremely large, making the regularized reward
> $r + \lambda \frac{p_{\text{trg}}(s'\mid s,a)}{p_{\text{src}}(s'\mid s,a)}$
> dominate the original reward r. In that case, the policy is effectively optimized to exploit the dynamics shift term $p_{\text{trg}}(s'\mid s,a)/p_{\text{src}}(s'\mid s,a)$ rather than the true reward signal. This issue becomes particularly severe under large dynamic shifts, leading to unstable training and, in practice, performance that can be worse than IQL.
>
>
> BOSA, on the other hand, learns the Q-function using “filtered” data filtered by a learned target dynamics model trained only on target data. However, the target dynamics may be bad when only limited target data are available, which introduces bias and noise into the filtered data used for Q learning. Moreover, under a large shift, the source data can be almost entirely filtered out, causing the algorithm to learn a policy purely from the limited target data, thus performing worse than IQL, which simply trains on the combined dataset.
>
> We thank the reviewer for raising this point. However, we believe that this phenomenon, while important, is not central to our experimental results, and a more detailed diagnosis of why each baseline can fail is somewhat orthogonal to the main contributions of our paper.

---

> ### Author Response · Authors · 2025-12-02
> **Further Response 3**
>
> **Q6:** The reviewer cannot distinguish whether the advantage MOBODY has over other algorithms comes from AWR or from other configurations.
>
> **A6:** As stated in A16, our implementation of DARA follows the ODRL benchmark and uses IQL as the backbone. Specifically, we apply DARA as a reward regularizer when training the Q and V functions, and then update the policy using an AWR-style objective. For BOSA, we follow the original implementation in the paper, first training a dynamics model and then performing supported policy optimization. **Thus, DARA can be viewed as IQL/AWR augmented with DARA-style reward regularization.**
>
> Since DARA also relies on AWR-style policy optimization but still does not match MOBODY’s performance, this suggests that MOBODY’s gains do not primarily come from the target-weighted BC loss alone, but also from the model-based rollout and exploration components.
>
> In the following A7, we will further explain whether the advantage MOBODY has over other algorithms comes from AWR or from other configurations.
>
> **Q7**  From a reviewer’s perspective, MOBODY’s main claimed contribution is its dynamics modeling. However, the new results suggest that this module has limited impact (Q19), while AWR/IQL-style policy extraction yields comparable or even stronger gains than the proposed Q-weighted BC (cf. Q18 vs. Q19). This raises concerns that the reported improvements may be driven more by auxiliary design choices than by the core contribution.
>
> **A7** We answer the questions step by step.
>
> (1) **The central claimed contribution of MOBODY is its dynamic modeling?**
>
>  We would first like to clarify that the central idea of MOBODY is to enable exploration in the target domain under dynamics shift. Existing regularization and data-filtering methods restrict training to low-shift regions, which can be severely suboptimal when the optimal policy lies outside these regions. In contrast, MOBODY uses a model-based RL framework that explicitly supports exploration under dynamics shift, with the dynamics module making this exploration feasible and reliable. Thus, the core contribution is the overall framework that couples model-based exploration with policy optimization, rather than the dynamics model or target-weighted BC loss in isolation.
>
>
> (2) **The new results suggest that the impact of the proposed dynamics module is, in fact, limited in Q(19)?**
>
> In Q(19), we showed that BOSA with MOBODY’s dynamics model improves performance by about 6% over vanilla BOSA, making it the strongest method after MOBODY itself. This already indicates that MOBODY’s learned dynamics are beneficial.
>
> However, MOBODY and BOSA use the dynamics model in fundamentally different ways. MOBODY rollouts synthetic transitions under the learned target dynamics and directly optimizes the policy on these rollouts, explicitly enabling model-based exploration in the target domain. In contrast, BOSA uses the target dynamics model only as a filter when training the Q-function, with an objective of the form
> $\mathbb{E}{(s,a) \sim D{\text{src+trg}}}\big[ \mathbf{1}(T_\text{trg}(s’ \mid s,a) > \epsilon),(Q(s,a) - r - Q(s’,a’))^2 \big]$,
> i.e., it trains $Q$ only on source and target transitions that “look like” target transitions. This design restricts the effective use of both source and target data; even with ground-truth target dynamics, the algorithm would still discard many transitions, optimize with target data and would not exploit model-based exploration. It would not fully exploit model-based exploration, so we do not expect dramatically improved performance.
>
> Moreover, in the policy optimization stage, BOSA uses supported policy optimization with a constraint
> $\log \hat{\pi}_{\text{src+trg}}(a \mid s) > \epsilon$,
> which explicitly encourages the learned policy to stay within the support of the combined source+target data. As we discuss in the paper, actions that appear good under source dynamics can perform poorly in the target domain under large dynamics shifts, so imposing a joint support constraint over source and target can be harmful when the mismatch is substantial.
>
> Therefore, we believe the modest gain of “BOSA + MOBODY dynamics” over vanilla BOSA is mainly due to BOSA’s intrinsic design: using the dynamics only for filtering and constraining the policy on source+target support, rather than any limitation of MOBODY’s dynamics model.

---

> ### Author Response · Authors · 2025-12-02
> **Further Response 4**
>
> (3) **Moreover, AWR/IQL-style policy extraction appears to provide performance gains that are at least comparable, and in some cases stronger, than the proposed Q-weighted BC formulation (Q18 vs. Q19 tables).**
>
> There may be a misunderstanding regarding the results in Q18. That experiment compares two variants of MOBODY that both use model-based rollouts: one with target Q-weighted BC and one with IQL-weighted BC. In other words, **Q18 does not compare plain AWR/IQL-style policy extraction against our full method, but only two different weighting schemes for the BC term within MOBODY.**
>
> Empirically, Q18 motivates our choice of the target Q-weighted BC loss as an auxiliary regularizer: it is simpler to implement and achieves better performance than the IQL-weighted BC loss in off-dynamics offline RL. To further separate the effect of the BC weighting from model-based rollouts, we also compare AWR (B3) and target Q-weighted BC (B5) without any model-based rollouts in the following table. In this setting, the target Q-weighted BC loss (B5) performs similarly to IQL (B3) and remains far below MOBODY. This shows that the major performance gains come from the model-based rollouts and exploration enabled by our dynamics model, while the target BC term serves as an important regularizer that stabilizes training and further boosts performance. But we still want to highlight that the target BC loss is essential in our algorithm that improves the performance as shown in A4 in Section 4.3.1 of the main paper.
>
>
>
>
> (4) **We further provide a more comprehensive comparison of our method with others, showing the benefit and contribution of each component. We report the summarized result in the last row for easier comparison.**
>
> - (a) **B1 vs. B2.** B1 corresponds to the original BOSA, and B2 to BOSA using MOBODY’s dynamics model. B2 consistently outperforms B1, showing that MOBODY learns a better dynamics model. As discussed in Q6(2), the improvement is limited mainly because BOSA only uses the dynamics for filtering, not for rollouts or exploration.
> -  (b) **B5 vs. B6.**
> B6 is exactly MOBODY, while B5 uses the target Q-weighted BC loss in policy optimization but does not use model-based rollouts. MOBODY (B6) significantly outperforms (B5), indicating that the auxiliary design choice (target Q-weighted BC) does not play a disproportionately large role. Comparing (B5) and (B3) (which differ in the BC weighting), we also do not observe substantial improvement from target Q-weighted BC alone. This further supports that the BC loss is a secondary regularization tool rather than the main source of gains.
>
> - \(c\) **B3 vs. B4.** Both B3 and B4 use IQL-style weighting for policy regularization, but only B4 uses model-based rollouts from MOBODY’s dynamics. We again observe performance improvements when model rollouts are used, demonstrating the effectiveness of our dynamics learning. This comparison also reinforces that the major performance gains do not come from the target Q-weighted BC loss, but from the model-based rollout and exploration enabled by MOBODY.
>
> Taken together, these results show that while MOBODY’s dynamics model can improve other methods (e.g., BOSA), the main performance advantage of MOBODY comes from its dynamics modeling and its model-based exploration and rollout scheme, with the BC loss serving as a **complementary and essential** regularizer rather than the primary driver of the gains.
>
>
> | Env| B1| B2| B3| B4| B5| B6|
> |-|-|-|-|-|-|-|
> || BOSA| BOSA + MOBODY dynamics | Policy learning with IQL weight (without Model-based rollout)| Policy learning with IQL weight (with Model-Based rollout)| Policy learning with target Q weight (without Model-Based rollout) | MOBODY, policy learning with target Q weight (with Model-Based rollout)|
> | HalfCheetah-gravity-0.5| 43.96 ± 5.68| 45.07 ± 7.49| 46.11 ± 1.93| 45.13 ± 2.21| 37.65 ± 2.13| 47.18 ± 1.23 |
> | HalfCheetah-gravity-2.0| 27.86 ± 0.94| 32.99 ± 1.29| 31.85 ± 1.31| 40.42 ± 3.63| 28.66 ± 3.21| 41.60 ± 7.35 |
> | HalfCheetah-fraiction-0.5 | 68.93 ± 0.35| 69.11 ± 0.28| 64.89 ± 3.04| 64.82 ± 1.53| 66.11 ± 2.09| 69.54 ± 0.48 |
> | HalfCheetah-friction-2.0| 46.53 ± 0.37| 45.14 ± 0.91| 46.25 ± 2.36| 51.30 ± 2.53| 43.97 ± 5.59| 50.02 ± 3.26 |
> | Walker2d-gravity-0.5| 40.09 ± 20.37| 39.81 ± 11.01| 29.72 ± 16.02| 48.45 ± 2.33| 38.72 ± 4.11| 43.57 ± 2.32 |
> | Walker2d-gravity-2.0| 8.91 ± 2.28| 11.75 ± 4.01| 32.20 ± 1.05| 39.67 ± 3.21| 37.34 ± 2.46| 44.32 ± 4.58|
> | Walker2d-fraiction-0.5| 63.94 ± 20.40| 71.32 ± 8.41| 68.81 ± 1.12| 63.47 ± 5.01| 64.77 ± 1.04| 76.96 ± 1.99 |
> | Walker2d-friction-2.0| 39.06 ± 17.36| 44.97 ± 12.86| 72.91 ± 0.37| 71.39 ± 1.81| 71.05 ± 0.93| 73.74 ± 0.49 |
> | Total| 339.28| 360.16| 392.74| 424.65| 388.27| 446.93|

---

> ### Author Response · Authors · 2025-12-02
> **Further Response 5**
>
> **Q5:** The authors provide no substantive analysis of when the shared-transition assumption breaks or how the method behaves under extreme or structurally divergent dynamics, leaving the method’s robustness and failure modes insufficiently addressed.
>
> **A5:** When the shared-transition assumption breaks, our method effectively reduces to training the dynamics model on target data only, since no useful shared information or structure remains. In this case, the performance should be similar to the “target-only” dynamics training baseline, as demonstrated in Ablation A1 (Section 4.3). But empirically, we demonstrate that our method still works in large shift cases that recent literature has considered, at least showing that **our method potentially can be more robust and applicable than baselines**
>
> Regarding behavior under extreme or structurally divergent dynamics, we explicitly evaluate our method under gravity and friction shift levels of **0.1 and 5.0**, where the target domain’s gravity/friction is scaled by 0.1 or 5.0 relative to the source. **These represent the largest shift levels considered in the current off-dynamics RL literature and in the ODRL benchmark, and these are highly divergent dynamics.** As shown in Table 1 of the main paper, baseline methods perform very poorly in these settings (e.g., HalfCheetah-Friction-5.0, Walker2d-Friction-5.0, Ant-Friction-5.0), whereas MOBODY achieves more than 30% improvement over the best baselines.
>
> Beyond environment-level shifts, our experiments in Tables 4 and 5 also include morphology and kinematics shifts, where joints/hips are disabled or legs/feet are shortened. For these tasks, we consider both medium and **hard shift levels**, corresponding to increasing degrees of robot damage as defined in the ODRL benchmark. **This shows that in the Mujoco or Adroit tasks, our method still works and performs better than baseline even when the finger of the hand is totally broken, or the leg/hip/joint is totally broken.**
>
> Overall, our experiments **cover most of the “hard” and large/extreme shift settings in the ODRL benchmark and recent off-dynamics RL research literature**, including both environment shifts (gravity/friction) and agent shifts (morphology/kinematics). Across all these scenarios, our method consistently outperforms existing off-dynamics RL approaches. If the reviewer believes that these settings still do not qualify as “extreme” or “structurally divergent” dynamics, we would appreciate it if the reviewer could point out what kinds of environments or shift patterns they have in mind.
>
>
>
> Reference:
>
> [1] ODRL: A Benchmark for Off-Dynamics Reinforcement Learning
>
> [2] Composite Flow Matching for Reinforcement Learning with Shifted-Dynamics Data
>
> [3] Provably Efficient Model-based Policy Adaptation
>
> [4] BPRL: A BEHAVIORAL APPROACH TO STATE REPRESENTATION IN REINFORCEMENT LEARNING

---

### Official Review · Reviewer_NauK · 2025-10-29

**Soundness:** 3
**Presentation:** 3
**Contribution:** 2
**Rating:** 4
**Confidence:** 4

**Summary:**

This paper tackles the "off-dynamics offline reinforcement learning" (ODRL) problem, where the goal is to learn a target-domain policy using a large source-domain dataset and a very small target-domain dataset, which have different dynamics. The authors propose MOBODY (Model-Based Off-Dynamics Offline RL), an algorithm that learns a model of the target dynamics to generate synthetic data, allowing the policy to explore high-reward regions that may be in high-shift areas, which existing methods often fail to do.

**Strengths:**

1. The paper correctly identifies a key weakness of current ODRL methods (e.g., DARA, BOSA): their conservatism, which stems from data filtering or reward penalization, prevents them from exploring high-reward states that may exist outside of low-dynamics-shift regions.
2. The central idea of using separate action encoders while sharing state and transition functions is a clever and intuitive way to leverage the large source dataset to learn shared structural knowledge while still capturing the domain-specific differences with the small target dataset.

**Weaknesses:**

1. The paper fails to cite or compare against a relevant class of transfer learning and offline RL methods. The related work discusses reward regularization (DARA) and data filtering (VGDF), but it omles a large body of work on policy regularization, specifically methods that use state-distribution matching or regularization. For example, State Regularized Policy Optimization (SRPO) and similar approaches that explicitly constrain the policy's state-visitation frequency to be close to a trusted distribution (like the target dataset) are highly relevant. Such methods provide an alternative and potentially more direct way to handle the dynamics shift than the complex model-based approach proposed here.
2. While the dynamics learning architecture is interesting, the policy learning side feels incremental and like a complex combination of existing ideas.
- The "enhanced target data" buffer is a simple-enough combination of target data, model rollouts (a standard model-based RL technique, e.g., MOPO), and augmented source data (using the existing DARA method).
- The "target Q-weighted behavior cloning loss" (Eq. 8) is presented as a novel contribution but is a minor variant of advantage-weighted regression, which the paper itself cites as an inspiration (IQL). The paper provides no theoretical or strong empirical justification (e.g., a direct comparison) for why its specific formulation ($\exp (Q / \operatorname{avg}|Q|) \times(\pi(s)-a)^2$) is superior to the more established and better-understood formulations used in methods like IQL.
3. The final MOBODY algorithm is highly complex. It stitches together a VAE-style dynamics learner with an ensemble for uncertainty, the full DARA framework for reward augmentation, and a custom Q-weighted policy optimizer. This high number of "moving parts" makes the method difficult to reproduce, tune, and analyze, and it obscures which components are providing the most benefit.
4. The motivation for the Q-weighted loss is purely intuitive. There is no analysis of its properties. Why use an exponential weight? Why normalize by the average absolute Q-value? How sensitive is this to the Q-value estimates, which are known to be noisy? A more principled derivation or analysis is needed to justify this specific design over simpler, existing alternatives.

**Questions:**

1. Could you elaborate on why methods that directly regularize the policy's state-visitation distribution (like SRPO) were not considered as baselines, given that they directly address the problem of distribution shift under new dynamics?

2. The Q-weighted BC loss (Eq. 8) is very similar to IQL's advantage-weighted regression. What is the performance of your method if you simply replace your loss with the IQL policy loss? This would clarify whether the specific formulation of Eq. 8 is a key part of your contribution.

3. How much of the performance gain comes from the model-based rollouts versus the superior dynamics-learning architecture? (i.e., what is the performance of MOPO if it uses your dynamics model?)

---

> ### Author Response · Authors · 2025-11-21
>
> We thank the reviewer for the constructive review of our paper. We answer the reviewer's question one by one and include a TLDR for each response if it is too long.
>
> ---
> **Q1.** Discussion of the existing baselines on the state-visitation frequency method SRPO.
>
> **A1:**
> TLDR: We discuss the missing baseline SRPO and provide additional experiment results on SRPO. Our method significantly outperforms SRPO in 12 out of 16 tasks, receiving $95.11$ \% overall improvement, and our model is more applicable and robust across different tasks.
>
> We thank the reviewer for pointing out this baseline; we will add further discussion and experimental results in the revision. SRPO regularizes the policy so that the induced state distributions are similar across the two domains, whereas reward-regularization methods such as DARA aim to match the trajectory distributions. Moreover, the success of SRPO relies on the assumption of homomorphous MDPs, i.e., if a state pair (s, s') can be reached in one step in the source domain, then it can also be reached in one step in the target domain, and vice versa. This is a relatively strong assumption and can be easily violated. For example, when the target domain has much larger gravity, a one-step transition from one state to another that is feasible in the source may be impossible in the target.
>
> We provide additional experimental results comparing SRPO and MOBODY. We observe that our method significantly outperforms SRPO in 12 out of 16 settings, while SRPO performs better in a few cases. Notably, in some settings where MOBODY performs better, SRPO can fail and learn very low returns, such as in Walker2d-Gravity-0.1/2.0/5.0. This further demonstrates that MOBODY is more robust and broadly applicable than SRPO.
>
> We hypothesize that SRPO works well when the homomorphous MDP assumption is approximately satisfied, but degrades substantially when this assumption is violated. Overall, the empirical results indicate that our method performs better across more tasks and is applicable in a wider range of settings. In addition, similar to DARA, the state-distribution constraint in SRPO may still overlook high-reward states that are rare in the source domain (e.g., high-reward states in high-shift regions), which can lead to a suboptimal policy.
>
> #### Table: Performance of SRPO and MOBODY, MOBODY outperforms RADT in 12 out of 16 tasks and receives much higher performance in these tasks. MOBODY achieves an overall performance improvement of $95.11$ \% over SRPO.
> | Env         | Shift        | SRPO              | MOBODY            | Improvement (%)   |
> |:------------|:-------------|:------------------|:------------------|:------------------|
> | Halfcheetah | gravity-0.1  | **32.94± 2.71** | 14.18 ± 1.06      | -56.95%           |
> | Halfcheetah | gravity-0.5  | 41.99± 2.67       | **47.18 ± 1.23** | +12.36%           |
> | Halfcheetah | gravity-2.0  | 32.24 ±3.89       | **41.46 ± 7.35** | +28.60%           |
> | Halfcheetah | gravity-5.0  | -2.33±0.48        | **83.05 ± 1.21** | -         |
> | Halfcheetah | friction-0.1 | 17.36 ±0.54       | **57.53 ± 2.49** | +231.39%          |
> | Halfcheetah | friction-0.5 | **109.18±  4.63** | 69.54 ± 0.48      | -36.31%           |
> | Halfcheetah | friction-2.0 | **75.19± 2.36** | 50.02 ± 3.26      | -33.48%           |
> | Halfcheetah | friction-5.0 | 5.10±3.84         | **59.20 ± 4.91** | +1060.78%         |
> | Walker2d    | gravity-0.1  | 13.67±10.17       | **65.85 ± 5.08** | +381.71%          |
> | Walker2d    | gravity-0.5  | **56.28± 5.47** | 43.57 ± 2.32      | -22.58%           |
> | Walker2d    | gravity-2.0  | 8.52±   0.68      | **44.32 ± 4.58** | +420.19%          |
> | Walker2d    | gravity-5.0  | 5.12±0.21         | **46.05 ± 20.73** | +799.41%          |
> | Walker2d    | friction-0.1 | 9.02± 0.66        | **28.23 ± 9.13** | +212.97%          |
> | Walker2d    | friction-0.5 | -0.23± 0.20       | **76.96 ± 1.99** | -        |
> | Walker2d    | friction-2.0 | 15.51±  7.46      | **73.74 ± 0.49** | +375.44%          |
> | Walker2d    | friction-5.0 | 4.94±0.44         | **27.38 ± 3.87** | +454.25%          |
>
> ---
>
> **Q2** Whether the "enhanced target data" buffer is a simple-enough combination of target data.
>
> **A2:** “Enhanced target data” is merely a notation we use in the policy learning section to compactly denote the combination of model rollouts, target data, and reward-augmented data. It is not a novel concept or technical contribution; we will further clarify it in our revision.

---

> ### Author Response · Authors · 2025-11-21
>
> **Q3** Any empirical justification of the target Q-weighted BC loss and why it is in this form?
>
> **A3:**
> TLDR: we empirically justify our target Q-weighted BC loss and conduct additional experiments showing that target-Q weighted BC performs better than IQL weight.
>
> The target Q-weighted BC loss is inspired by the advantage-weighted regression (AWR) [1] and the reward-weighted regression (RWR) [2]. While the AWR is built on the RWR and we follow the idea of RWR by replacing the Monte-Carlo estimation on the reward with the target Q value.
>
> Empirically, we justify the target Q-weighted BC loss in ablation study A4 in Section 4.3 of the main paper: replacing the Q-weighted BC loss with vanilla BC consistently leads to worse performance than MOBODY and other baselines. We also test a more standard IQL-style weighting, which is the advantage-weighted regression built on reward-weighted regression [1], i.e., using weights $\exp(\beta (Q(s,a) - V(s)))$, and report the results in the following table. We see that our Q-weighted BC loss slightly improves the performance. The reason might be 1) training an extra value network here using model-based rollout introduces more complexity and noise, leading to worse performance.
>
> Since the IQL-style weighting (AWR) is worse empirically while requiring training an additional value network (thus increasing complexity), we adopt a simpler exponential weighting directly on the Q-value, inspired by IQL and reward-weighted regression (RWR). We further normalize Q by its average absolute value, following TD3-BC, because the trade-off between policy optimization and BC is highly sensitive to the scale of rewards and Q-values; this normalization yields a more stable and comparable weighting across tasks.
>
> #### Table 2: Comparison of using IQL weight for BC loss and target Q weighted BC loss.
> | Env         | Shift        | IQL weight for BC               | MOBODY            |
> |------------|--------------|-------------------|-------------------|
> | Halfcheetah| gravity-0.5  | 45.13 ± 2.21| **47.18 ± 1.23**|
> | Halfcheetah| gravity-2.0  | 40.42 ± 3.63| **41.46 ± 7.35**|
> | Halfcheetah| friction-0.5 | 64.82 ± 1.53| **69.54 ± 0.48**|
> | Halfcheetah| friction-2.0 | **51.30 ± 2.53**  | 50.02 ± 3.26      |
> | Walker2d   | gravity-0.5  | **48.45 ± 2.33**  | 43.57 ± 2.32      |
> | Walker2d   | gravity-2.0  | 39.67 ± 3.21| **44.32 ± 4.58**|
> | Walker2d   | friction-0.5 | 63.47 ± 5.01| **76.96 ± 1.99**|
> | Walker2d   | friction-2.0 | 71.39 ± 1.81| **73.74 ± 0.49**|
>
>
> ---
> **Q4.** The final MOBODY algorithm is highly complex, which might be hard to tune and analyze.
>
> **A4:**
> TLDR: we provide detailed parameter tuning ablation studies and rule-of-thumb hyperparameters in the Appendix, demonstrating that despite the multiple components, our tuning components of our method are comparable to what is commonly seen in offline model-based RL methods.
>
> We argue that, despite having multiple components, MOBODY is not particularly difficult to tune. First, on the dynamics learning side, although we use multiple losses, all loss weights are simply set to 1, and adjusting them does not noticeably improve performance. This part requires no hyperparameter tuning. Second, on the policy learning side, the main components are the ensemble-based uncertainty quantification and the target Q-weighted BC loss. The ensemble uncertainty is the standard mechanism used in model-based RL, and BC loss is also widely used in offline RL to mitigate the OOD issue of the state-action pair. We further provide a hyperparameter analysis in Table 7 of the Appendix, where we vary the uncertainty penalty weight and the target Q-weighted BC weight. The results show that performance is quite stable across a range of settings, and even suboptimal hyperparameter combinations still outperform baselines in most cases. In addition, we include a rule-of-thumb hyperparameter tuning procedure in the Appendix (see L959), illustrating that with a small amount of tuning, our method already achieves strong performance. Altogether, these observations suggest that MOBODY is not overly sensitive to hyperparameters and is not being over-tuned.
>
> Beyond this, we emphasize that there is no single universal hyperparameter setting that works optimally across all tasks with different environments, shift types, and shift levels. Even in standard (no-shift) settings, model-based RL methods typically require environment-specific hyperparameters. In this context, the number of hyperparameters introduced by MOBODY is modest and comparable to what is commonly seen in offline model-based RL methods.
>
> ---

---

> ### Author Response · Authors · 2025-11-21
>
> **Q5.** It obscures which components are providing the most benefit.
>
> **A5:**
> TLDR: We provide ablation studies to demonstrate the benefit of each component and show that the performance gains come from the full algorithm working together.
>
> First, we emphasize that our performance gains do not come from a single component in isolation; rather, they arise from the full MOBODY algorithm working together: the learned dynamics enables the exploration of the high reward region target domain and the target Q weighted BC loss regularize the policy to avoid the exploration to the OOD region in model-based RL training and also guide the policy to generate high Q value action in the target domain instead of source domain. For each major component, we provide ablation studies in Section 4.3 to demonstrate both its effectiveness and necessity. Specifically, we conduct four ablations (e.g., removing the target Q-weighted BC loss, learning dynamics using only target data, etc.), where in each study we modify exactly one component of MOBODY while keeping all others fixed.
>
> Specifically,  we show that in the following Table 3, by removing the target Q weighted BC loss, our method only shows slight improvement over the MOPO, but when combined with target Q weighted BC, we receive a bigger performance improvement. Also, in the ablation A1 in the main paper, we also test that by training the dynamics on target data only, even with Q-weighted BC, the performance is not comparable with baselines in some cases. These demonstrate the necessity of each component and clarify the contribution of each part without obscuring the overall synergy of the full method.
>
>
> ---
> **Q6.** The motivation for the target Q-weighted loss is purely intuitive.
>
> **A6:** TLDR: the key motivation of using target Q-weighted loss is to guide the policy toward actions that are expected to perform better under the target dynamics, instead of uniformly weighting the action in the source.
>
> We first clarify that our main contribution and motivation regarding the target-Q-weighted BC loss is not the specific functional form in Eq. (8). Instead, our motivation has two components: (1) introducing a BC loss specifically to mitigate out-of-distribution issues in offline off-dynamics RL, and (2) using target-Q weighting to explicitly push the policy toward actions with high Q-values under the target dynamics.  The first motivation (using BC to improve robustness) is intuitive and follows prior work, while we view the second, leveraging target-Q information to reweight the BC loss, as the key conceptual contribution and motivation in the off-dynamics RL problem rather than the exact formula.
>
> More concretely, in off-dynamics RL, naively cloning source-domain actions can harm performance: actions that work well in the source domain may perform poorly in the target domain due to the dynamics shift, so a vanilla TD3-BC–style loss is inadequate. Our “target Q-weighted BC loss” is therefore designed to guide the policy toward actions that are expected to perform better under the target dynamics, directly mitigating this mismatch where the target Q value is learned from the reward augmented data.

---

> ### Author Response · Authors · 2025-11-21
>
> **Q7.** How much of the performance gain comes from the model-based rollouts versus the superior dynamics-learning architecture? (i.e., what is the performance of MOPO if it uses your dynamics model?)
>
> **A7:** We are not quite understanding the question regarding "performance gain comes from the model-based rollouts versus the superior dynamics-learning architecture". We are happy to provide more answers if the reviewer can clarify.
>
> We want to clarify that part of the performance gain does come from the model-based rollouts in the model-based RL, while the model-based rollouts rely on the superior dynamics-learning architecture, as the rollouts come from the learned dynamics.
>
> To answer your second question in the parentheses, we also provide additional experimental results using our dynamics learning approach with MOPO. Comparison between MOPO and MOBODY shows that model-based rollouts will only help if they come from a superior dynamics-learning architecture. And how we obtain rollouts from the MOPO model-based model is exactly the same as our MOBODY model-based model. This result shows that another important source of improvement comes from additional policy learning techniques in MOBODY. Thanks for suggesting this comparison. We will add them to our revision.
>
>
> #### Table 3: Performance of the MOPO uses MOBODY dynamics.
> | Env| Shift| MOBODY| MOBODY Dynamics in MOPO| MOPO|
> |------------|--------------|-------------------|--------------------|-------------------|
> | Halfcheetah| gravity-0.5| **47.18 ± 1.23**| 42.15 ± 2.31| 40.20 ± 7.20|
> | Halfcheetah| gravity-2.0| **41.46 ± 7.35**| 24.38 ± 5.21| 21.89 ± 10.49|
> | Halfcheetah| friction-0.5| **69.54 ± 0.48**| 53.76 ± 3.82| 54.98 ± 5.91|
> | Halfcheetah| friction-2.0| **50.02 ± 3.26**| 41.13 ± 4.14| 42.33 ± 3.89|
> | walker2d   | gravity-0.5| **43.57 ± 2.32**| 39.48 ± 2.99| 40.32 ± 8.78|
> | walker2d   | gravity-2.0| **44.32 ± 4.58**| 33.19 ± 4.72| 28.79 ± 3.07|
> | walker2d   | friction-0.5| **76.96 ± 1.99**| 65.41 ± 4.19| 60.31 ± 3.04|
> | walker2d   | friction-2.0|  **73.74 ± 0.49**| 69.74 ± 2.07| 68.38 ± 1.09|
>
>
> [1] Advantage-Weighted Regression: Simple and Scalable Off-Policy Reinforcement Learning.
>
> [2] Relative entropy policy search.

---

### Official Review · Reviewer_WLw7 · 2025-11-01

**Soundness:** 3
**Presentation:** 3
**Contribution:** 3
**Rating:** 6
**Confidence:** 3

**Summary:**

The paper proposes MOBODY, a model-based approach to off-dynamics offline reinforcement learning, addressing the setting where the source and target environments share rewards but differ in dynamics. Unlike prior methods that penalize or exclude high-shift samples, MOBODY leverages both limited target data and abundant source data through separate action encoders and a shared latent transition model. A cycle-transition loss is introduced to stabilize representation learning, while a target-Q-weighted behavior cloning loss guides the policy toward high-value target actions. Experimental results on MuJoCo and Adroit benchmarks show that MOBODY consistently outperforms existing off-dynamics and model-based baselines, particularly under large dynamics shifts.

**Strengths:**

- The paper introduces a clear and well-motivated model-based framework for off-dynamics offline reinforcement learning. Unlike prior work that relies on reward regularization or data filtering, it directly learns the target dynamics, offering a fresh and practical way to handle dynamics mismatch in offline settings.
- The paper introduces MOBODY, which shows solid and consistent improvements over existing baselines on several off-dynamics offline RL benchmarks.
- This paper provides a comprehensive empirical evaluation across both MuJoCo and Adroit environments, covering multiple types of dynamics shifts—gravity, friction, kinematics, and morphology—at varying levels of severity.
- The paper is well written, with clear organization and logical flow throughout. In particular, the results are effectively visualized—figures and tables are well designed, easy to interpret, and enhance the overall clarity and impact of the presentation.

**Weaknesses:**

- Lack of theoretical analysis supporting the proposed approach. The paper does not provide any convergence guarantees, error bounds, or stability analysis, which are important for understanding the conditions under which the algorithm is expected to perform reliably [1-2]. Given the complexity of the model and the presence of domain shifts, some theoretical insights—even in simplified settings—would significantly enhance the rigor and credibility of the work.
- While MOBODY generally outperforms existing baselines, some of the reported improvements appear marginal or potentially within the range of statistical noise—for example, in the Walker2d-Friction-2.0 setting, where the gains over baselines are minimal. Moreover, in a few scenarios, MOBODY underperforms relative to other methods. The paper would benefit from a more detailed analysis or discussion of these cases to help contextualize performance variance and identify potential limitations of the approach.
- All experiments are conducted in dense-reward environments, which may not fully capture the challenges faced in sparse-reward settings. It would strengthen the paper to include an analysis of MOBODY's performance in sparse environments, where exploration and generalization are typically more difficult [3-4].
- Although the method is described as operating with "limited target data," it still assumes access to 5,000 transitions, which may not reflect truly scarce or low-resource settings. It remains unclear how MOBODY would perform when target data is extremely limited, noisy, or when the dynamics mismatch is highly non-smooth or discontinuous. Evaluating the sensitivity of the model under such challenging conditions would help clarify the robustness and practical applicability of the approach.
- The proposed framework introduces multiple components—such as separate action encoders for the source and target domains (ψ_src, ψ_trg) and a VAE-based cycle transition module—which likely increase both training time and memory usage. However, the paper does not provide any runtime, training cost, or sample-efficiency comparisons against baseline methods, making it difficult to assess the practical overhead of the approach.
- While the paper compares MOBODY against several strong baselines, it overlooks a class of return-conditioned or return-based methods that have also demonstrated competitive performance in off-dynamics offline RL settings [5]. Including representative algorithms from this category—such as decision transformer variants or return-augmented models—would provide a more comprehensive evaluation and help clarify whether the proposed approach offers advantages beyond standard model-based and reward-regularized baselines.

**Questions:**

- Could the authors provide any theoretical guarantees (e.g., convergence, stability, or error bounds) for MOBODY, even under simplified assumptions? How does the model behave under known pathological dynamics mismatches?
- In settings where MOBODY's improvements over baselines are small (e.g., Walker2d-Friction-2.0), or where it underperforms, what factors contribute to this variance? Could the authors provide an analysis or hypothesis for these outcomes?
- How does MOBODY perform in sparse-reward environments, where exploration is more challenging and informative transitions are rare? Could the authors include results or discussion in such settings?
- How sensitive is MOBODY to the quantity of target data? Have the authors evaluated performance when fewer than 5,000 transitions are available?
- Several return-conditioned or return-based algorithms (e.g., Decision Transformers, RADT, or return-augmented models) have shown promising results in off-dynamics offline RL. How does MOBODY compare conceptually and empirically to such return-based approaches?

## Reference
- **[1]** Miyaguchi, Kohei. "Worst-case offline reinforcement learning with arbitrary data support." Advances in Neural Information Processing Systems 37 (2024): 61481-61512.

- **[2]** Brandfonbrener, David, et al. "When does return-conditioned supervised learning work for offline reinforcement learning?." Advances in Neural Information Processing Systems 35 (2022): 1542-1553.

- **[3]** Cen, Zhepeng, et al. "Learning from sparse offline datasets via conservative density estimation." arXiv preprint arXiv:2401.08819 (2024).

- **[4]** Rengarajan, Desik, et al. "Reinforcement learning with sparse rewards using guidance from offline demonstration." arXiv preprint arXiv:2202.04628 (2022).

- **[5]** Wang, Ruhan, et al. "Return augmented decision transformer for off-dynamics reinforcement learning." arXiv preprint arXiv:2410.23450 (2024).

---

> ### Author Response · Authors · 2025-11-21
>
> We thank the reviewer for the positive feedback and constructive review of our paper.
>
> ---
> **Q1.**
> The work lacks theoretical analysis: there are no convergence guarantees, error bounds, or stability results to characterize when the algorithm is reliable, especially under domain shifts. Even simplified theoretical insights would substantially improve the rigor and credibility of the paper.
>
> **A1:** Thank you for the suggestion. We note that the references mentioned by the reviewer are primarily theoretical works under different scenarios (offline RL without distribution shift and return-conditioned supervised learning), whereas our paper is empirical designed and our contribution is proposing a practical algorithm for off-dynamics offline RL. Our work is motivated and designed from principle ideas and achieves competitive performance in various complicated environments and settings. We see a more formal theoretical treatment of our method as an interesting and complementary direction, and we will incorporate a clearer discussion of this in the revision.
>
> We provide a high-level sketch of a potential theoretical understanding of our method, and we will refine this argument and present a more rigorous analysis in the revised version.
>
> The potential theoretical analysis is motivated by MOPO and PAR[4], utilizing the classic decompostiion that bound the performance error with dynamics learning error through latent space representation. Specifically, the performance error can be decomposed in terms of the dynamics estimation error. In our setting, the target dynamics error $\lVert \hat{T}\_{\text{trg}}(s,a) - T\_{\text{trg}}(s,a) \rVert$ can be further related to the discrepancy in the latent space, $\lVert \psi\_{\text{src}}(z_s,a) - \psi\_{\text{trg}}(z_s,a) \rVert$ and its estimated error, under the assumption of the Lipchitz continuous of the transition function and the relation with the source dynamics. This suggests that the error bound depends on the dynamics mismatch in the latent representation. Thus, the bound will depend on (i) the true dynamics mismatch, (ii) the estimated dynamics mismatch, and (iii) the generalization error of the learned source dynamics model. MOBODY is designed to accurately capture the target dynamics, particularly to learn $\lVert \psi\_{\text{src}}(z_s,a) - \psi\_{\text{trg}}(z_s,a) \rVert$ in the latent space, thereby reducing $\lVert \hat{T}\_{\text{trg}}(s,a) - T\_{\text{trg}}(s,a) \rVert$, which is also well motivated from the previous theory in PAR.
>
>
>
>
> ---
> **Q2.** Discussion on some underperforming cases of MOBODY and why the improvement of the MOBODY is small in some cases.
>
>
> **A2:**
> TLDR: Overall, MOBODY is always better or comparable when compared with baselines.
>
> There are a small number of cases (only 4 out of 32 friction/gravity-shift tasks) where MOBODY underperforms other baselines. Even in these settings, MOBODY remains within a small margin of the best-performing baseline with less 1.7% of the performance loss, while MOBODY receives 58% improvement in other cases. We believe this may be due to two factors: (1) in very challenging cases such as Hopper-Gravity-5.0, all methods achieve very low rewards (i.e., essentially fail to learn), and MOBODY obtain comparable performance in these near-failure regimes; and (2) in some small-shift cases, baselines based on policy regularization already achieve relatively high rewards, such as HalfCheetah-Friction-0.1 and Walker-Friction-0.1. In such settings, the additional exploration benefit from model-based methods is less pronounced, so MOBODY’s improvements can be modest, or the performance of MOBODY is comparable to these baselines. Overall, MOBODY is consistently better than or comparable to the baselines.
>
>
> ---
> **Q3:** Whether MOBODY can be extended to the sparse reward setting.
>
> **A3:** For our method, we do not observe significant improvements in sparse-reward settings such as AntMaze. We argue that sparse rewards are intrinsically challenging in the off-dynamics RL setting. As shown in the ODRL benchmark [1], for AntMaze, a sparse-reward benchmark, no existing method achieves a reward greater than 0, and all methods essentially fail. Therefore, we believe that tackling sparse-reward off-dynamics RL is an important and challenging future research direction that requires substantially different methods from those in the current literature. We will add a discussion of this limitation and direction in the paper.

---

> > ### Comment · Reviewer_WLw7 · 2025-11-22
> > **Response (1/3)**
> >
> > **For A1:** Thanks a lot for the explanation.
> >
> > **For A2:** Thanks for the analysis! I did notice one small but interesting pattern and wanted to ask about it.
> > In some of the small-shift cases, MOBODY actually gives very large improvements (for example, HalfCheetah-Friction-0.1 and Walker2d-Gravity-0.1). But in other small-shift settings like Ant-Friction or Walker2d-Friction, the gains are much smaller or just similar to the baselines.
> > I’m curious whether this difference might come from the environment mechanics themselves or if there might be another reason behind this variation. Would you be able to share a bit of intuition on this?
> >
> > **For A3:** I couldn’t find experiments on sparse-reward environments like AntMaze in the paper. If I missed them, please feel free to point me to the right place. I understand sparse-reward off-dynamics RL is extremely challenging, but I’m still curious how MOBODY performs in that kind of setting. Any brief insight or preliminary result would be great.

---

> > > ### Comment · Reviewer_WLw7 · 2025-11-22
> > > **Response (2/3)**
> > >
> > > **For A4:** I care a lot about how the target dataset is constructed, because different papers use different conventions. I’m not fully convinced by the statement that “5k target transitions are already a very small amount of target data, as also noted in prior work [1,2,3].” For example, [3] constructs the target data at the trajectory level, while methods like DARA build the target dataset at the step level, which leads to different data distributions and different levels of supervision.
> > > However, I appreciate the additional experiments you provided. I would suggest updating the PDF to include a clear explanation of how the 5k target transitions are generated
> > >
> > > **For A5:** Thank you for the analysis — the running time looks completely reasonable to me.

---

> > > > ### Comment · Reviewer_WLw7 · 2025-11-22
> > > > **Response (3/3)**
> > > >
> > > > **For A6:** Thank you so much for running the additional comparison. Could you share a few more details about the experimental setup for the RADT baseline? This would help me better understand the comparison.
> > > >
> > > > Also, I’d recommend adding the key points from your rebuttal directly into the revised PDF, and ***marking the changes in a different color***. That would make the paper feel more complete and easier for readers to follow the updates.

---

> ### Author Response · Authors · 2025-11-21
>
> **Q4.** How MOBODY would perform when the target data is extremely limited (less than 1k), noisy, or when the dynamics mismatch is highly non-smooth or discontinuous.
>
> **A4:**
> TLDR: 5k target transitions are already a very small amount of target data, as also noted in prior work [1,2,3]. We additionally conduct ablations with even fewer target transitions. Moreover, our dynamics are deterministic, so the induced off-dynamics shifts are inherently non-smooth or even discontinuous.
>
> We would first like to clarify that, following the ODRL benchmark [1], 5,000 transitions already are a very small number of target transitions. Prior work on online or offline off-dynamics RL typically assumes more access to target transitions, often using a ratio such as 1/10 of the source transitions [2,3], e.g., 100,000 target transitions for 1 million source transitions or training steps. In the table below, we further provide additional experimental results with varying amounts of target data, including 500, 1k, 2k, and 5k transitions. We observe performance degradation for MOBODY as the size of the target dataset decreases. But this decrease is not that significant, showing our method also works well when the target data size is small.
>
> Furthermore, we clarify that the off-dynamics shifts induced by friction/gravity or kinematics/morphology changes in our experiments constitute non-smooth or even discontinuous changes in the dynamics. In our setting, the transition is deterministic, and any mismatch in dynamics leads to different next states, implying a non-smooth shift in the transition map. As the magnitude of the shift increases, the difference in the next state typically becomes larger, making the dynamics shift more non-smooth and discontinuous. Empirically, we also evaluate our method under extreme dynamics mismatch, and observe that it remains effective under such large shifts as shown in Table 1 of the main paper with 0.1/5.0 shift level.
>
>
> #### Table: Performance of MOBODY under different target number transitions. 5k is the original result in the paper, and the data size in the ODRL benchmark.
> | Env         | Shift        | 5k MOBODY        | 2k MOBODY        | 1k MOBODY        | 500 MOBODY       |
> |------------|--------------|------------------|------------------|------------------|------------------|
> | Halfcheetah| gravity-0.5  | **47.18 ± 1.23**     | 42.72 ± 2.05     | 39.29 ± 2.52     | 40.65 ± 3.09     |
> | Halfcheetah| gravity-2.0  | **41.46 ± 7.35**     | 28.31 ± 5.47     | 28.69 ± 5.95     | 26.78 ± 6.31     |
> | Halfcheetah| friction-0.5 | **69.54 ± 0.48**     | 69.62 ± 1.08     | 67.56 ± 1.52     | 64.14 ± 1.90     |
> | Halfcheetah| friction-2.0 | **50.02 ± 3.26**     | 46.66 ± 2.92     | 44.26 ± 3.31     | 44.70 ± 3.53     |
>
> ---
> **Q5.** What are the runtime, training cost, or sample-efficiency comparisons against baseline methods?
>
> **A5:**
> TLDR: We provided the runtime analysis and training cost in Table 8 in the Appendix.
>
> We clarify that we have provided an estimated runtime analysis in Table 8 of the Appendix. Compared with the model-free method DARA, MOBODY incurs a higher runtime due to the additional transition model training and model rollouts. In contrast, MOBODY has a similar runtime to the model-based method MOPO. Although MOBODY includes more components than vanilla MOPO while using a transition model of similar size, the forward and backward computation costs remain comparable, and the additional losses in MOBODY are computed efficiently from intermediate layers, so they do not introduce substantial extra running time or significantly higher memory usage.

---

> ### Author Response · Authors · 2025-11-21
>
> **Q6.** How does MOBODY compare conceptually and empirically to such return-based approaches?
>
> **A6:** We provide additional experimental results on the RADT [3]. Our method outperforms the RADT in 13 out of 16 tasks, achieving an overall performance improvement of 53.23%. Though the RADT uses a different paradigm compared with ours, the intrinsic idea of it is still applying the reward regularization method like DARA to augment the return, leading to a similar problem as the DARA, optimizing the policy using data from low-shift regions and limiting exploration of high-reward states in the target domain that do not fall within these regions. RADT leverages the return-conditioned supervised learning paradigm, which is known to lack stitching ability and thus relies heavily on data quality. This might explain the performance drop in RADT.
>
> #### Table: Performance of RADT and MOBODY, MOBODY outperforms RADT in 13 out of 16 tasks and receives much higher performance in these tasks. MOBODY achieves an overall performance improvement of 53.23% over RADT.
> | Env         | Shift        | RADT-MV             | MOBODY            | Improvement (%)   |
> |:------------|:-------------|:-----------------|:------------------|:------------------|
> | Halfcheetah | gravity-0.1  | **16.14± 0.43** | 14.18 ± 1.06      | -12.14%          |
> | Halfcheetah | gravity-0.5  | 40.50± 2.48      | **47.18 ± 1.23** | +16.49%          |
> | Halfcheetah | gravity-2.0  | 33.28 ± 9.96     | **41.46 ± 7.35** | +24.58%          |
> | Halfcheetah | gravity-5.0  | 71.31±7.82       | **83.05 ± 1.21** | +16.46%          |
> | Halfcheetah | friction-0.1 | 9.74±0.21        | **57.53 ± 2.49** | +490.66%         |
> | Halfcheetah | friction-0.5 | 66.50 ± 0.98     | **69.54 ± 0.48** | +4.57%           |
> | Halfcheetah | friction-2.0 | 37.74 ± 5.52     | **50.02 ± 3.26** | +32.54%          |
> | Halfcheetah | friction-5.0 | 25.74±10.49      | **59.20 ± 4.91** | +129.99%         |
> | Walker2d    | gravity-0.1  | 26.56±6.84       | **65.85 ± 5.08** | +147.93%         |
> | Walker2d    | gravity-0.5  | **55.20 ± 4.75** | 43.57 ± 2.32      | -21.07%          |
> | Walker2d    | gravity-2.0  | 13.50 ± 5.68     | **44.32 ± 4.58** | +228.30%         |
> | Walker2d    | gravity-5.0  | 4.61±1.27        | **46.05 ± 20.73** | +898.92%         |
> | Walker2d    | friction-0.1 | 10.58± 0.50      | **28.23 ± 9.13** | +166.82%         |
> | Walker2d    | friction-0.5 | **78.58 ± 1.17** | 76.96 ± 1.99      | -2.06%           |
> | Walker2d    | friction-2.0 | 42.18 ± 14.80    | **73.74 ± 0.49** | +74.82%          |
> | Walker2d    | friction-5.0 | 8.36±3.63        | **27.38 ± 3.87** | +227.51%         |
>
>
>
>
> [1] ODRL: A Benchmark for Off-Dynamics Reinforcement Learning
>
> [2] Off-Dynamics Reinforcement Learning: Training for Transfer with Domain Classifiers
>
> [3] Return Augmented Decision Transformer for Off-Dynamics Reinforcement Learning
>
> [4]Cross-Domain Policy Adaptation by Capturing Representation Mismatch

---

> ### Author Response · Authors · 2025-11-25
>
> We thank the reviewer for the quick response. We now further address the questions and at the meantime, we uploaded the new PDF.
>
> ---
> **Q1.** For A2: In some of the small-shift cases, MOBODY actually gives very large improvements (for example, HalfCheetah-Friction-0.1 and Walker2d-Gravity-0.1). But in other small-shift settings like Ant-Friction or Walker2d-Friction, the gains are much smaller or just similar to the baselines.
>
> **A1:** Thanks for pointing this out—this is an interesting question. We first clarify that the shift level “0.1” actually corresponds to a large dynamics shift, since the friction in the target environment is only one-tenth of that in the source environment, while 1.0 means exactly the source environment without shift.
>
> Though shift levels are 0.1, it might create different levels of dynamics mismatch for different tasks, given different morphologies/shapes of robots, which might also lead to different performance improvements in different environments. Besides, we believe that this could partially be due to the environment's mechanics themselves. Different environments might have different structures in the transitions, and different dynamics shift leads to different shared knowledge of the transitions. For those environment that has little improvement, there are possibly two reasons: 1) the baseline methods performs well and the assumption of the baseline might be well satisfied, and extra exploration with model-based doesn't birng significant improvement, such as Ant-Friction-0.1, where all methods receive over 50 normalized score or 2) when the shift is large, the shared structure might be hard to learn due to large shift, thus the improvement from MOBODY is small, e.g. Walker2d-Friction-0.1. However, we want to emphasize that these are only a small number of cases, and our method still generally outperforms baselines.
>
>
>
> ---
> **Q2.** Any brief insight or preliminary result on Antmaze or other sparse-reward environments?
>
> **A2:** We did not put the results of sparse reward setting or Antmaze in the paper. But here we can provide some preliminary results we conducted on this environment. In Antmaze, the dynamics shift is created by the different map layout of the target domain, where the target domain has a different layout than the source domain, following the ODRL benchmark. In the following table, we provide some results of the Antmaze. Specifically, we conduct the experiment on the target domain with two different map layouts. MOBODY receives around 35% reaching the goal, and slightly outperforming the DARA. Also, we try some other map layouts, but usually fail to reach the goal as the existing methods did.
>
> | AntMaze-Medium        | MOBODY     | DARA |
> |----------------|------------| ----|
> | shift-level 1  | 35.31 ± 5.13     | 30.49 ± 10.01  |
> | shift-level 2  | 38.95 ± 12.87    | 36.47 ± 17.28  |
>
> Besides Antmaze, which is an off-dynamics environment provided by the ODRL benchmark, we also conducted some experiments here with the Adroit manipulation tasks. Note that in our paper, we also include the Adroit manipulation tasks, but we are using the dense reward setting. Here, we provide additional results on the sparse reward Adroit manipulation task. Following the Adroit API, the environment returns a reward of 10 for environment success (reaching the goal/head of the nail is close to the board) and -0.1 otherwise. In these tasks, the environment and the dynamic shift are created by shrinking the finger of the hand or the broken joint of the hand. We show the experimental results in the following table. We see that our method can receive a positive reward, i.e., success in 24% of the cases, while DARA always fails. These at least demonstrate the effectiveness of our method. But in many cases, both MOBODY and DARA fail, especially in large-shift settings (shift_level = hard), suggesting that sparse-reward setting tasks are much harder than dense-reward settings.
>
> |{Env}-{shift_type}-{shift_level}        | MOBODY     | DARA |
> |----------------|------------| ----|
> | hammer-shrink-finger-medium  | 24.39 ± 7.68     | 0.0  |
> | hammer-broken-joint-medium  |  19.36 ± 4.25   | 0.0  |
>
> Further, we would like to mention that, the difficulties of the sparse reward offline off-dynamics setting come from 1) the successful trajectories are rare 2) when dynamics shift exists, the successful trajectories in the source might not be feasible in the target, making the reward even more sparse and 3) the sparse reward lead to more unstale training (mode collapse, etc), which results in larger variance in the reward.

---

> ### Author Response · Authors · 2025-11-25
>
> **Q3:** How is the target data constructed, and whether the dataset has different data distributions and different levels of supervision for methods like RADT and DARA?
>
> **A3:** For the target domain offline data, we use the medium-level target dataset provided by the ODRL benchmark. Specifically, the medium datasets are collected by an early-stopped SAC that has approximately one third or one half the performance of the expert policy, similar to the convention in the D4RL. The trajectories are rollout until 5k transitions data are collected.
>
> It is true that the RADT [3] constructs the target data at the trajectory level for training, while methods like DARA build the target dataset at the transitions/steps level. We also want to clarify that this different format of training data does not require a different data collection process or a different distribution of the target data. As mentioned, the data are collected through rollout trajectories until reaching 5k transition data. From the offline data, we can construct trajectory-level data from the offline transition data for RADT training, or we can only use the transition-level data for the training of DARA.
>
> Overall, the 5,000 transition dataset is collected at the trajectory level by the medium policy. We will also update the PDF to explain the process in detail.
>
>
> ---
> **Q4.** For A6: Thank you so much for running the additional comparison. Could you share a few more details about the RADT baseline experimental setup? This would help me better understand the comparison.
>
>
> **A4:**
> We reproduce RADT-MV and conduct experiments on the ODRL benchmark. We utilize 1M transitions from the D4RL source dataset and 5,000 transitions from the target environment. We conduct experiments on HalfCheetah and Walker2d with gravity and friction shifts at levels $\{0.1, 0.5, 2.0, 5.0\}$.
>
> Following RADT [3], we first train CQL on both the source and target datasets using OfflineRL-Kit (https://github.com/yihaosun1124/OfflineRL-Kit). We leverage the trained Q-function to augment the returns in the source dataset. Specifically, for each trajectory in the source dataset, we sample 1000 Q-values to estimate the mean and variance for Direct Return Matching. We augment the returns using a clip ratio of $[0.8, 1.2]$.
>
> We then train the Decision Transformer on this augmented source dataset combined with the target dataset for HalfCheetah and Walker2d. During RADT-MV inference, we set the initial returns to 12000 and 6000 for HalfCheetah, and 5000 and 2500 for Walker2d. We report the mean normalized score across three seeds.
>
>
> **Q5.** Revising the PDF and marking the changes in a different color.
>
> **A5:** Thanks for the recommendation. We uploaded the updated PDF, including the new experimental results and clarification on the questions. Due to the time limit, we will update all the baseline results in our camera-ready version.

---

### Author Response · Authors · 2025-12-03
**Summary of the rebuttal**

In summary, we appreciate the reviewer’s positive feedback on our paper: recognizing the key weakness of existing methods, acknowledging that we propose a clear and well-motivated model-based framework for off-dynamics offline RL via model-based exploration to address dynamics mismatch, noting our broad and thorough experimental coverage, highlighting the strong and consistent empirical performance, and commenting that the writing is clear and logically structured.

----

During the rebuttal, we aimed to address the key concerns raised by the reviewers and have updated the submission accordingly. Below, we summarize the main responses:
- (1) We discussed the two baseline methods mentioned by the reviewer, analyzed their weaknesses, and conducted experiments showing that our method outperforms them (R1A6, R2A1). We also discussed and compared with modern systematic offline RL methods as requested (R3A3 (further response)).
- (2) We empirically validated that the performance gains mainly come from model-based rollouts using a better-learned dynamics model, and that the target-Q-weighted BC loss acts as an auxiliary but essential regularizer rather than the primary driver of performance (R2A5, R2A7, R3A7 (further response)).
- (3) We empirically justified the choice of the target-Q-weighted BC loss, showing that it outperforms AWR/IQL-style weighting in MOBODY’s setting (R3Q18, R2Q3, R3A7 (further response)). We further clarified our motivation for using the target-Q-weighted BC loss (R2A6, R3A7).

----

In addition, we addressed several other concerns:
- (1) We clarified multiple points raised by the reviewers, including:
    - training cost/runtime is comparable to the model-based baseline (R1A5);
    - hyperparameter tuning is not complicated (R2A4);
    - how target data is constructed (R1A3 (further response));
    - we focus on deterministic transition settings (R3A2 (further response));
    - why IQL can outperform some off-dynamics baselines (R3A9, R3A4 (further response));
    - how we perform model ensemble uncertainty quantification (R3A11, R3A21);
    - what happens when the target dataset is even smaller (R1A4, R3A20);
    - brief insights and experimental results in sparse-reward settings (R1A1 (further response));
    - analysis of cases where MOBODY’s improvement is modest (R1A2, R1A1 (further response));
    - evidence supporting MOBODY’s performance under large/extreme shifts and justification that the underlying assumptions are reasonable (R3A12, R3A5 (further response));
    - discussion of the relation to recent TD representation learning works (R3A22).
- (2) We conducted additional experiments requested by R3 to further demonstrate the effectiveness and justify the design of our method:
    - replacing the target-Q-weighted BC loss with AWR-style policy regularization, showing that the simpler target-Q-weighted BC loss is preferable in model-based off-dynamics RL settings (R3A18);
    - replacing BOSA’s dynamics learning with MOBODY’s dynamics learning, which improves BOSA and indicates that MOBODY’s dynamics model is stronger;
    - additional experiments on the sensitivity to the number of ensemble models (R3A21);
    - comparison of the generalization ability of MOBODY’s dynamics model versus alternatives, demonstrating the effectiveness of MOBODY’s dynamics learning (R3A2 (further response));
    - experiments that directly support our claim about the main performance drivers of MOBODY (R3A7 (further response)).


---

Update of the PDF:
- (1) We had some minor updates in the introduction and method sections based on the rebuttal, including:
    - include the discussion of baseline SRPO
    - justification of the additive form $z_s+ \psi(z_s,a)$
    - brief introduction on DARA reward regularization, and reward learning
    - empirically justification of the target-Q weighted BC loss
- (2) In the experiment section, we included the following updates:
    - include 2 baseline methods, and update the discussion accordingly
    - provide analysis on the MOBODY modest improvement cases
    - discuss why IQL performs better than off-dynamics RL baselines
    - update ablation studies to show that model-based rollouts and dynamics is the main drivers of the improvement, but target-Q weighted BC loss is also important
    - empirically justify the choice of target-Q weighted BC loss
- (3) In the Appendix, we included additional ablation studies requested by the reviewer that could potentially strengthen the paper:
    - discussion of the related baselines and connection with the previous works
    - comparison of different sizes ofthe  target dataset
    - comparison of using IQL weight and target Q weighted for BC loss in MOBODY, validating the choice of target Q weighted BC


Overall, we thank the reviewer for the constructive feedback and the opportunity to improve our paper.

---

### Meta-Review · Area_Chair_gpsm · 2026-01-07

**Summary:**

Across the three reviews, the paper is generally seen as a solid and promising contribution to off-dynamics offline reinforcement learning, with strong empirical results and a clear, well-motivated problem formulation, but with recurring concerns about clarity of contribution, theoretical grounding, and attribution of performance gains.

All reviewers acknowledge that the paper identifies an important weakness of existing off-dynamics RL methods, namely their conservatism due to reward penalization or data filtering, and agree that MOBODY’s model-based exploration of target dynamics is a meaningful and practically relevant direction. The use of separate action encoders with a shared latent transition model is viewed as intuitive and clever, and the experimental coverage on MuJoCo and Adroit across many types and levels of dynamics shift is consistently praised. The paper is also considered well written and clearly presented.

The main weaknesses raised across reviews center on three themes. First, there is a lack of theoretical analysis: reviewers note the absence of convergence guarantees, error bounds, or stability analysis, and while they understand the paper is primarily empirical, they would prefer at least simplified theoretical insight or clearer discussion of assumptions and failure modes. Second, reviewers repeatedly question whether the core contribution is well isolated. They express difficulty disentangling how much of MOBODY’s performance comes from improved dynamics learning, model-based rollouts, the target-Q-weighted behavior cloning loss, or other design choices, and some reviewers initially suspect that policy extraction techniques similar to AWR or IQL may be driving much of the gains rather than the proposed dynamics modeling. Third, the overall method is seen as complex, combining many standard components (ensembles, MOPO-style rollouts, reward augmentation, BC regularization), which raises concerns about reproducibility, tuning, and whether the novelty lies more in engineering than in a clean conceptual advance.

Specific additional concerns include incomplete baseline coverage in the initial submission, particularly missing comparisons to return-conditioned methods (e.g., Decision Transformer variants) and state-distribution regularization approaches such as SRPO; mixed or marginal gains in some environments that may fall within statistical noise; limited discussion of robustness under extreme assumption violations; reliance on dense-reward benchmarks with limited evidence in sparse-reward settings; and ambiguity around what constitutes “limited” target data, given differences in how target datasets are constructed across papers.

Overall, reviewers find MOBODY empirically strong and practically useful, and after rebuttal many concerns are partially alleviated through additional experiments, ablations, and clarifications. However, they remain somewhat unconvinced at a conceptual level due to the lack of theory, the complexity of the method, and lingering questions about whether the claimed core contributions are sufficiently cleanly justified and isolated.

**Reviewer Concerns:**

Concerns that have been satisfyingly addressed:

- Empirical justification of MOBODY components and performance drivers
  - The rebuttal provided extensive ablation studies showing:
  - Target-Q-weighted BC loss is effective and slightly better than IQL-style weighting.
  - Superior dynamics learning contributes meaningfully to performance.
  - Model-based rollouts using MOBODY dynamics drive most of the performance gains.
  - Individual components’ contributions and their synergy are clarified.
  - Tables and experiments comparing MOBODY dynamics with MOPO and RADT support these claims.
- Baseline coverage and comparisons
  - Additional experiments were provided against SRPO, RADT, and other return-conditioned methods.
  - Results show MOBODY generally outperforms these baselines.
  - Discussion on differences in target dataset construction and trajectory vs. transition-level data is included.
- Runtime and hyperparameter complexity
  - Authors clarified that MOBODY’s runtime is comparable to MOPO, higher than DARA, and acceptable.
  - Hyperparameter tuning is shown to be stable across a range of values; rule-of-thumb guidance is provided.
- Performance in sparse-reward settings
  - Preliminary experiments on AntMaze and sparse-reward Adroit tasks are reported.
  - MOBODY can succeed in some cases, outperforming DARA, though tasks remain difficult.
-  Analysis of modest improvement or underperformance cases
  - Authors explained why some small-shift or high-shift environments show smaller gains.
  - Environment mechanics, baseline performance, and feasibility of high-reward transitions were discussed.
- Construction of target datasets and clarification of “limited” target data
  - Explained that 5k transitions are collected from early-stopped SAC policies and clarify trajectory-level vs. transition-level data.
  - Additional experiments with 500–2k transitions show MOBODY’s robustness.

Concerns that are still outstanding or only partially addressed:
- Theoretical analysis (reviewers wanted at least simplified theoretical results)
  - Rebuttal provides a high-level sketch of potential error decomposition and latent-space bounds.
  - No concrete convergence guarantees, formal error bounds, or stability analysis are presented.
- Clarity of conceptual novelty vs. engineering (conceptual novelty is still somewhat unclear)
  - Reviewers questioned whether MOBODY’s novelty is mainly in engineering multiple components rather than in a clean conceptual advance.
  - Rebuttal explains motivation and synergy but doesn’t fully address whether the overall design constitutes a fundamentally new paradigm beyond existing model-based RL and reward-regularized approaches.
- Sparse-reward off-dynamics RL (highlights a limitation rather than fully resolving it)
  - Preliminary results are shown, but performance remains low, and MOBODY fails on most maps.
  - Reviewers may still be concerned that MOBODY is not generally applicable to sparse-reward settings.
- Sensitivity to highly non-smooth or discontinuous dynamics shifts
  - Authors note that deterministic transitions lead to discontinuities, and extreme shifts were tested.
  - However, formal analysis of performance under pathological dynamics mismatches is still lacking.
- Formal justification of target-Q-weighted BC loss
  - Empirical justification is strong, but reviewers requested more principled derivation or analysis of why exponential weighting and normalization are optimal.

Summary:
Most empirical and practical concerns have been addressed convincingly: baseline coverage, performance analysis, ablations, runtime, hyperparameters, and target data construction. Outstanding concerns mainly revolve around theory, conceptual novelty, sparse-reward generalization, and formal justification of design choices.

**Reviewer Scores:**

Reviewer WLw7: 6->6
Reviewer NauK: 4->5
Reviewer MfFn: 6->7

---

### Decision · Program_Chairs · 2026-01-26

Accept (Poster)